# In-situ grafting of cobalt phthalocyanine on gas diffusion electrodes enables ampere-level CO₂ reduction

Huihui Yan[1,2], Gaoxiang He[2,3], Rongli Fan [1,2], Minyue Zhao[1,2], Huiting Huang [1,2], Zhexing Lin[1,2], Bin Gao[1,2], Jianyong Feng [1,2] ✉, Zhigang Zou [1,2,3] & Zhaosheng Li [1,2] ✉

Immobilizing molecular catalysts onto gas diffusion electrodes (GDEs) through covalent bonds provides a strategy to circumvent their issues of aggregation, detachment and poor conductivity during CO₂ electrolysis. However, this approach has been limited to catalysts equipped with specially designed functional groups, and directly covalent grafting of pristine molecular catalysts onto GDEs remains a formidable challenge. Herein, using pristine cobalt phthalocyanine (CoPc) as a model catalyst, we propose a polypyrrole (ppy) mediated electro-polymerization strategy that creates robust C–C bonds between GDEs and pristine CoPc. In this scheme, ppy acts as both the conductive linker and scaffold for pristine CoPc, and its electron donation effects further enhances the CO₂ electrolysis activity of CoPc centres. Here we show that the assembled CO₂ electrolyzer using CoPc/ppy/GDE electrode achieves stable operation for 120 h at 500 mA cm⁻² and 50 h at 1 A cm⁻² in alkaline media. When coupled with a triple-junction solar cell, the resulting photovoltaic-electrolysis system attains a solar-to-CO efficiency of 19.2%.

Electrocatalytic conversion of CO₂ into value-added chemicals represents a strategy to mitigate the climate crisis caused by greenhouse gases while also facilitating the storage of renewable energy[1–3]. Among various catalytic systems, molecular catalysts, particularly metal phthalocyanines and metal porphyrins, have emerged as promising candidates due to their well-defined atomic active sites and tunable electronic structures[4–6]. However, under operating conditions, especially at industrially relevant current densities (>200 mA cm⁻²), molecular catalysts often suffer from leaching, agglomeration, or structural degradation, which hinders their practical applications[7–12]. Recently, metal phthalocyanine-based molecular catalysts have demonstrated highly selective CO₂ electrolysis at high current densities in flow reactors[13,14]. Typically, these molecular catalysts are generally coated onto the carbon-based gas diffusion electrodes (GDEs) to address

issues of aggregation and poor conductivity. The majority of the previous works have relied on noncovalent anchoring strategies, which often require binder such as Nafion and incorporate scaffolds like carbon nanotube (CNT) and graphene to enhance the dispersion and adhesion of the molecular catalysts through π–π stacking interaction[15–17].

Considering the possible loss of molecular catalysts during operation, particularly at high current densities, anchoring them directly onto carbon-based GDEs through strong covalent bonds presents a desirable strategy[18–20]. Currently, the direct covalent grafting of metal phthalocyanine/porphyrin-based molecular catalysts onto carbon-based GDEs requires the use of materials that are specially designed functional groups (such as amino groups) or polymerizable groups (such as carbazole groups). Until now, due to the inherent

¹National Laboratory of Solid State Microstructures, College of Engineering and Applied Sciences, Nanjing University, Nanjing, China. ²Jiangsu Key Laboratory of Nano Technology, Nanjing University, Nanjing University, Nanjing, China. ³School of Physics, Nanjing University, Nanjing, China. ✉ e-mail: fengjianyong@nju.edu.cn; zsli@nju.edu.cn

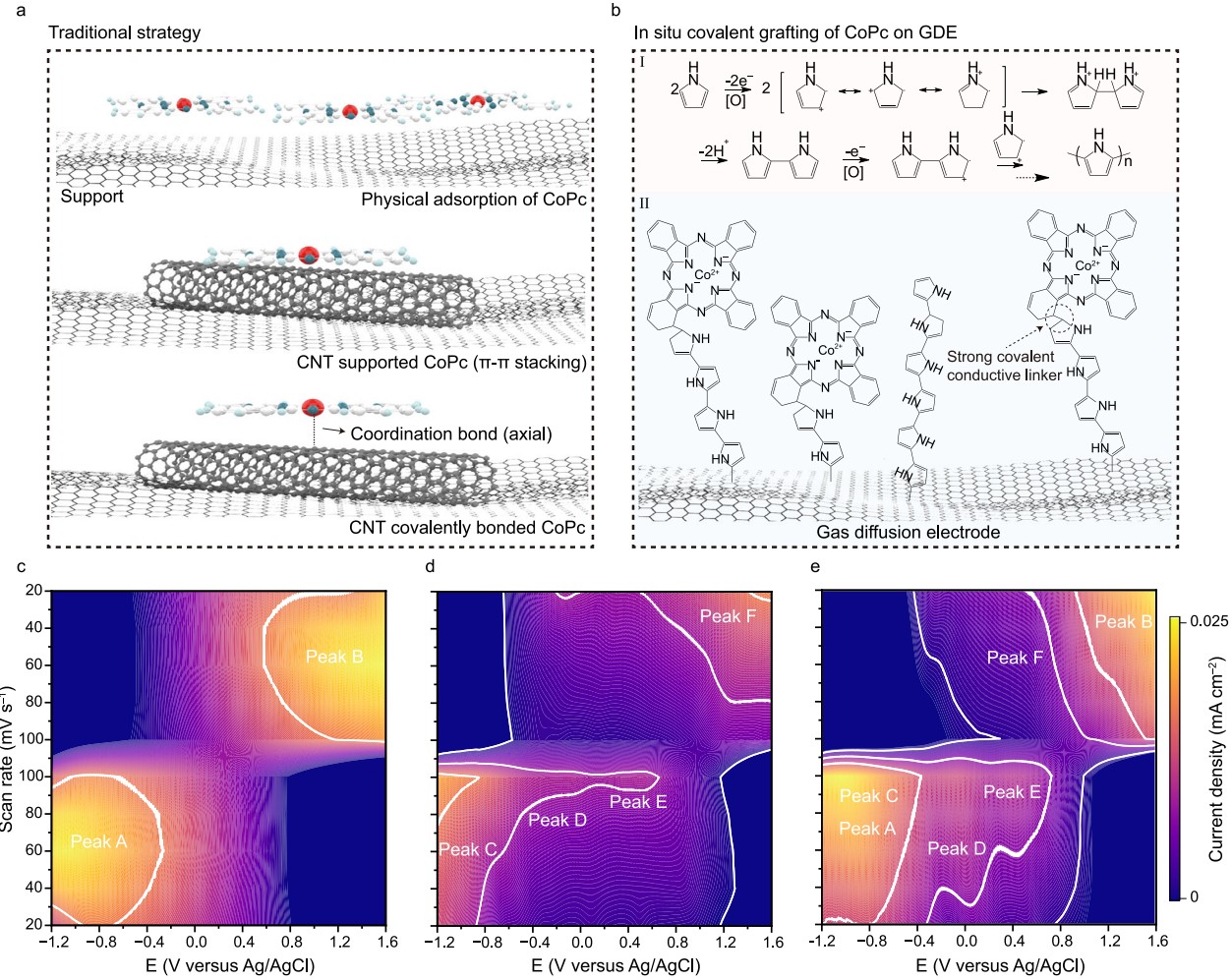

**Fig. 1 | Preparative route of CoPc/ppy/GDE. a** Traditional strategies for immobilizing CoPc on GDEs. **b** In situ covalent grafting of CoPc on GDE. Contour plots of the CV patterns of ppy/GDE (**c**) in 0.01 M 1-butyl-3-methylimidazolium tetrafluoroborate ([BMIM]BF$_4$) with 4 mM pyrole, CoPc/GDE (**d**) in 0.01 M [BMIM]BF$_4$ with 1 mM CoPc and CoPc/ppy/GDE (**e**) in 0.01 M [BMIM]BF$_4$ with 4 mM pyrole and 1 mM CoPc at different scan rates (from 20 to 100 mV s$^{-1}$). All potentials were calibrated to the Ag/AgCl scale, and electrochemical measurements were performed at room temperature (23 ± 2 °C). The resistance value is 15 ± 2 Ω. There is no iR correction for voltages. Source data are provided as a Source Data file.

chemical inertness of pristine metal phthalocyanines/porphyrins, direct electrochemical polymerization or covalent grafting of them onto carbon-based GDEs has been challenging. Furthermore, among the common covalent bonds that can form between metal phthalocyanines/porphyrins (or their derivatives) and carbon-based GDEs, C−C bond offers more inert and robust for extreme reaction conditions, when comparing to those hydrolysable amide and ester bonds[19].

Herein, using pristine CoPc as the example, we report an in situ electro-polymerization strategy for the direct covalent grafting of pristine metal phthalocyanines onto GDEs. This method occurs at room temperature and leverages the conductive polymer ppy as a molecular "glue", forming robust C−C bonds with pristine CoPc. Meanwhile, besides acting as the scaffold, the poly-based matrix also shows electron donation effects that enhance the activity of active Co+ sites. To bridge the gap between material innovation and industrial application, we further engineered a membrane electrode assembly (MEA) electrolyzer incorporating the CoPc/ppy/GDE cathode. This device achieves stable CO production for 50 h at ampere-level current densities (1 A cm$^{-2}$) in alkaline media. The proposed strategy thus addresses the challenge of direct covalent grafting of pristine metal phthalocyanines onto GDEs for efficient and durable CO$_2$ reduction reaction (CO$_2$RR) under industrially relevant current densities (>200 mA cm$^{-2}$).

## Results
### Synthesis of CoPc/ppy/GDE

Figure 1a illustrates the traditional strategies for immobilizing molecular catalysts onto supports (exemplified by CoPc-based catalysts), including physical adsorption, introducing π–π stacking interaction (with CNT or graphene), and forming covalent bonds. For the physically adsorbed or CNT- (or graphene-) supported molecular catalysts, they are still prone to aggregation or detachment, due to their relatively weak interactions with the supports. Although covalently bonded molecular catalysts are robust against detachment and aggregation, this approach generally requires the presence of specially designed functional groups (such as amino and carbazole groups). To functionalize molecular catalysts with these groups, sophisticated synthesis protocols are needed, which prohibit the large-scale application of this covalent bonding strategy in molecular catalysts. In contrast, in our strategy of ppy-mediated electro-polymerization, the oxidative polymerization of pyrrole monomers initiates the C−C coupling processes between GDEs and ppy, ppy and pristine metal phthalocyanines, thus connecting them together through robust C−C bonds[21]. The yielded CoPc/ppy/GDE electrode combines the advantages of robust C−C bonds, binder-free interface, highly conductive ppy linker, appropriate dispersion of Co single-atom sites (Fig. 1b).

Moreover, no pretreatment of carbon-based GDEs is required, which minimizes the amount of oxygen-containing functional groups (such as carboxyl and hydroxyl groups), thereby guaranteeing sufficient hydrophobicity for efficient $CO_2$ gas penetration.

We conducted a series of electrochemical analyses to understand the growth process of CoPc/ppy/GDE electrode. The cyclic voltammetry (CV) profiles of CoPc/ppy/GDE and control samples were recoreded in the potential window of −1.2 to 1.6 V versus Ag/AgCl at various scan rates. As shown in Fig. 1c, ppy/GDE obtained from electro-polymerization of pyrrole exhibits only broad and featureless oxidation and reduction waves between −1.0 and 1.0 V versus Ag/AgCl. Comparatively, three redox couples could be resolved for CoPc-modified GDE (CoPc/GDE), with their locations identified in Fig. 1d. These redox couples are probably associated with the redox transitions of Co centres in CoPc ($Co^{2+} \leftrightarrow Co^+ \leftrightarrow Co^0$). Notably, CoPc/ppy/GDE electrode produced in the presence of both pyrrole and CoPc shows similar redox characteristics to that of CoPc/GDE, with however negatively shifted peak positions and largely enhanced current response in the cathodic scan (Fig. 1d and Supplementary Fig. 1).

Next, we utilized in situ Raman spectroscopy to probe the structural evolution of CoPc/ppy/GDE electrode during the electro-polymerization process. The Raman spectra show a gradual increase in the intensities of peaks that correspond to the characteristic vibrational modes of CoPc. Specifically, the macrocyclic deformation mode at 680 $cm^{-1}$ exhibits an increasing trend with the number of CV electro-polymerization cycles, quantitatively reflecting the progressive deposition of CoPc[22,23]. A shift in Co−N vibration at ~ 755 $cm^{-1}$ was also observed, which may be attributed to interfacial strain-induced changes of Co−N bonds in CoPc during electro-polymerization (Supplementary Fig. 2). Collectively, the progressively increased intensities of CoPc-related peaks with the number of CV cycles provide evidence for the in situ growth and integration of CoPc onto the GDE.

Third, we collected FTIR spectrum of CoPc/ppy/GDE and compared it with those of CoPc and ppy/GDE. To eliminate potential interference of free CoPc (i.e., CoPc adsorbed on GDE or ppy), CoPc/ppy/GDE was thoroughly cleaned by ultrasonication and washed with copious amounts of acetonitrile. As shown in Supplementary Fig. 3, the FTIR spectrum of CoPc/ppy/GDE exhibits combined features of both CoPc and ppy/GDE, confirming that pristine CoPc has been successfully incorporated into ppy/GDE[24].

Based on these observations (and the exceptional operation stability of CoPc/ppy/GDE as discussed below), we infer that during pyrrole electro-polymerization, pristine CoPc is integrated into ppy, with the configuration of CoPc terminated at the ppy chain; electron donation occurs from ppy to CoPc, thus yielding negatively shifted redox peaks relative to bare CoPc. Such inferences are further supported by density functional theory (DFT) calculations, in which strong chemical bonds form in ppy/GDE and CoPc/ppy/GDE; under this condition, ppy serves as an effective linker enabling robust covalent grafting of pristine CoPc onto GDE (Supplementary Figs. 4–6).

## Characterization and properties of CoPc/ppy/GDE

The electrochemical $CO_2RR$ performance of CoPc/ppy/GDE electrodes was systematically optimized by adjusting two key parameters, including the number of CV scans and the feed ratio of pyrrole to CoPc (Supplementary Figs. 7–9). Structural characterization using aberration-corrected transmission electron microscopy (TEM), scanning electron microscopy (SEM), atomic force microscopy, and TEM revealed a hybrid morphology of CoPc/ppy/GDE, which is composed of single-atom Co sites and randomly aggregated spherical structures (Fig. 2a–c, Supplementary Fig. 10). Inductively coupled plasma optical emission spectrometer (ICP-OES) quantification confirmed a higher Co content in CoPc/ppy/GDE (0.0837 wt%) compared to CoPc/GDE (0.0046 wt%), demonstrating the critical role of ppy in immobilizing pristine CoPc (Supplementary Table 1). To probe the chemical states,

electronic structures, and coordination environments of the CoPc/ppy/GDE electrode, X-ray absorption spectroscopy (XAS) and X-ray photoelectron spectroscopy (XPS) were performed. X-ray absorption near-edge structure (XANES) spectrum of CoPc/ppy/GDE at the Co K-edge indicates an average Co oxidation state between metallic Co (0) and CoO (+2); comparing to pristine CoPc of $D_{4h}$ symmetry, its red-shifted absorption edge and weakened $1s{\rightarrow}4p_z$ transition at ~7710 eV suggest reduced symmetry of the Co−$N_4$ coordination (Fig. 2d)[25]. Fourier-transform extended X-ray absorption fine structure analyses identify a dominant Co−N coordination peak at ~1.5 Å for both CoPc/ppy/GDE and pristine CoPc, indicating $Co^{2+}$−N as the primary species in them (Fig. 2e). Wavelet-transforms of EXAFS spectra further confirm this conclusion by revealing prominent intensity maxima at ~5.9 $Å^{-1}$; meanwhile, a minor intensity maximum is exclusively resolved at ~8.2 $Å^{-1}$ over CoPc/ppy/GDE, which is inferred to arise from the $Co^+$−N species (Fig. 2f, g, Supplementary Fig. 11)[26,27]. The XPS measurements further corroborate the modulated electronic structure of CoPc/ppy/GDE, revealing lower binding energies for Co 2p peaks than those of commercial CoPc (780.8 vs. 781.4 eV for Co $2p_{3/2}$; 795.9 vs. 796.5 eV for Co $2p_{1/2}$)[28]. These phenomena, as mentioned above, are consistent with the electron donation effects from ppy to the CoPc domain (Supplementary Fig. 12). Collectively, through the ppy-mediated covalent grafting strategy, improved CoPc dispersion, asymmetric coordination of the Co−$N_4$ centre, and enriched electron density at Co sites are achieved in CoPc/ppy/GDE, which would enhance the $CO_2RR$ performance and selectivity[27,29].

The $CO_2RR$ performance was evaluated in $CO_2$-saturated 0.1 M $KHCO_3$ aqueous solutions (pH = 6.8 ± 0.2) using a standard three-electrode H-type cell configuration. As shown by the linear scanning voltammetry (LSV) curves in Fig. 2h and Supplementary Fig. 13, CoPc/ppy/GDE reaches a cathodic current density of up to 40 mA $cm^{-2}$ at −0.78 V versus reversible hydrogen electrode (RHE), higher than that of ppy/GDE (7.5 mA $cm^{-2}$), CoPc/GDE (2.1 mA $cm^{-2}$), and CoPc/CNT-GDE (9.5 mA $cm^{-2}$). Subsequently, the Faradic efficiency of gaseous and liquid products was analysed via online gas chromatography and nuclear magnetic resonance (NMR) spectroscopy, respectively; only gas-phase CO and $H_2$ were detected, with no liquid products observed over a broad potential range (Supplementary Figs. 14 and 15)[30]. Notably, CoPc/ppy/GDE achieves a CO Faradaic efficiency ($FE_{CO}$) of 96% at −0.8 V vs. RHE, significantly outperforming the control samples of CoPc/GDE (85%), ppy/GDE (10%), and CoPc/CNT-GDE (93%) under identical conditions (Supplementary Fig. 16). To highlight the excellent performance of CoPc/ppy/GDE electrode, a detailed comparison to other systems is provided in Supplementary Table 2.

To clarify the intrinsic activity of these samples, their cobalt loadings were measured by ICP-OES (Supplementary Table 1). Based on these values, the Co loading-normalized specific activity of these samples has been derived (measured at a scan rate of 1 mV $s^{-1}$ to minimize the capacitive contributions), which follows the order of CoPc/ppy/GDE > CoPc/CNT-GDE > CoPc/GDE, demonstrating the unique advantages of ppy-mediated covalent grafting (including binder-free interface, highly conductive ppy linker, electron donation effect of ppy, appropriate dispersion and spatial configuration of Co single-atom sites) in CoPc/ppy/GDE for $CO_2RR$ (Supplementary Fig. 17). Furthermore, the measurements of electrochemical impedance spectroscopy (EIS) revealed a lower charge-transfer resistance (and correspondingly improved reaction kinetics) of CoPc/ppy/GDE than the control sample of CoPc/GDE (Supplementary Fig. 18 and Supplementary Table 3).

Complementary to electrocatalytic performance, long-term stability is a critical criterion for the practical application of catalysts. As an accelerated degradation test, thousands of CV cycles were performed on CoPc/ppy/GDE at a scan rate of 100 mV $s^{-1}$ and negligible performance degradation was observed. Chronoamperometry measurements conducted at −0.75 V vs. RHE further reveal that CoPc/ppy/

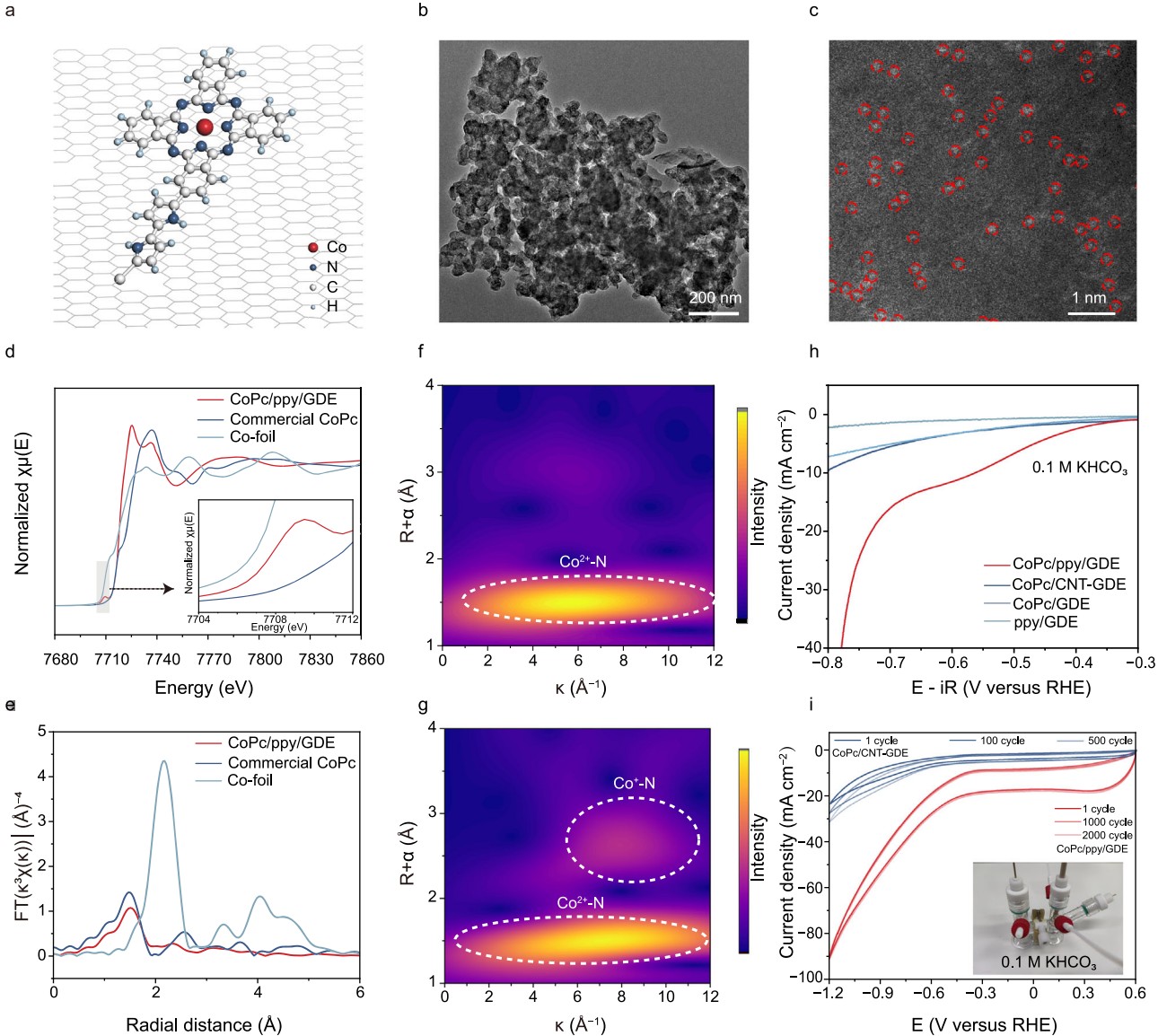

**Fig. 2 | Structural analyses and intrinsic activity of CoPc/ppy/GDE. a** Schematic illustration of CoPc/ppy/GDE. **b** TEM image of CoPc/ppy/GDE. **c** HAADF-STEM image of CoPc/ppy/GDE. The atomically dispersed Co sites are marked by red circles. **d** XAS spectra at Co K-edge of CoPc/ppy/GDE and the reference Co foil and CoPc samples. **e** Co K-edge FT EXAFS spectra in the R space of CoPc/ppy/GDE and reference Co foil and CoPc samples. **f, g** WT-EXAFS plots of CoPc samples and CoPc/ppy/GDE. **h** LSV curves of CoPc/ppy/GDE, CoPc/CNT-GDE, CoPc/GDE and ppy/GDE during $CO_2RR$ in $CO_2$-saturated 0.1 M $KHCO_3$ with an H-type cell (pH = 6.8 ± 0.2). The resistance value of the H-cell is 10 ± 2 Ω. **i** CV curve of CoPc/ppy/GDE during $CO_2RR$ in $CO_2$-saturated 0.1 M $KHCO_3$ with an H-type cell (pH = 6.8 ± 0.2). The gas flow rate of H-cell is 5 mL/min. The resistance value is 15 ± 2 Ω. There is no iR correction for voltages. All electrochemical measurements were performed at room temperature (23 ± 2℃). Source data are provided as a Source Data file.

GDE delivers a stable operating current for 200 h, outperforming that of CoPc/CNT-GDE (Supplementary Fig. 19). Subsequently, the post-reacted CoPc/ppy/GDE was subjected to HAADF-STEM, XAS and XPS analyses. HAADF-STEM measurements confirmed that the aggregated sphere-like morphology of CoPc/ppy/GDE catalyst is retained, with well-preserved atomically dispersed Co sites (Supplementary Fig. 20). Meanwhile, no obvious changes in the valence state and coordination environment of Co could be resolved for post-reacted CoPc/ppy/GDE (Supplementary Fig. 21). In addition, XPS analysis reveals that although the intensities of Co 2*p* peaks in post-reacted CoPc/ppy/GDE have decreased (possibly due to the contamination from electrolytes), their binding energies remain largely unaltered (Supplementary Fig. 22). All these post-reaction analyses together demonstrate that our CoPc/ppy/GDE electrode with the covalent grafting feature is highly resistant to structural collapse/changes under prolonged electrolysis. The negligible demetallation, reduction and aggregation of Co active sites in

CoPc/ppy/GDE guarantee its stability under highly reductive and high current density conditions.

While the aqueous-phase tests convincingly demonstrate the high $CO_2RR$ activity of CoPc/ppy/GDE, the inherent low solubility of $CO_2$ in aqueous electrolytes (~34 mM) imposes severe mass transport constraints. Thus, a flow cell was constructed to facilitate reactant transport and distribution, in which CoPc/ppy/GDE served as the cathode. The LSV curve in Supplementary Fig. 23 clearly shows that CoPc/ppy/GDE requires only −0.68 V vs. RHE to deliver a current density of 300 mA cm$^{-2}$, and the associated $FE_{CO}$ of 97% and turnover frequency (TOF) of ~ 10 s$^{-1}$ further demonstrate the practical application potential of CoPc/ppy/GDE for $CO_2RR$ (Supplementary Figs. 24 and 25).

Motivated by the $CO_2RR$ performance of CoPc/ppy/GDE, NiPc/ppy/GDE, CuPc/ppy/GDE, and FePc/ppy/GDE were further fabricated to extend the applicability of our covalent grafting strategy (Supplementary Fig. 26). Among them, NiPc/ppy/GDE exhibits a CO selectivity

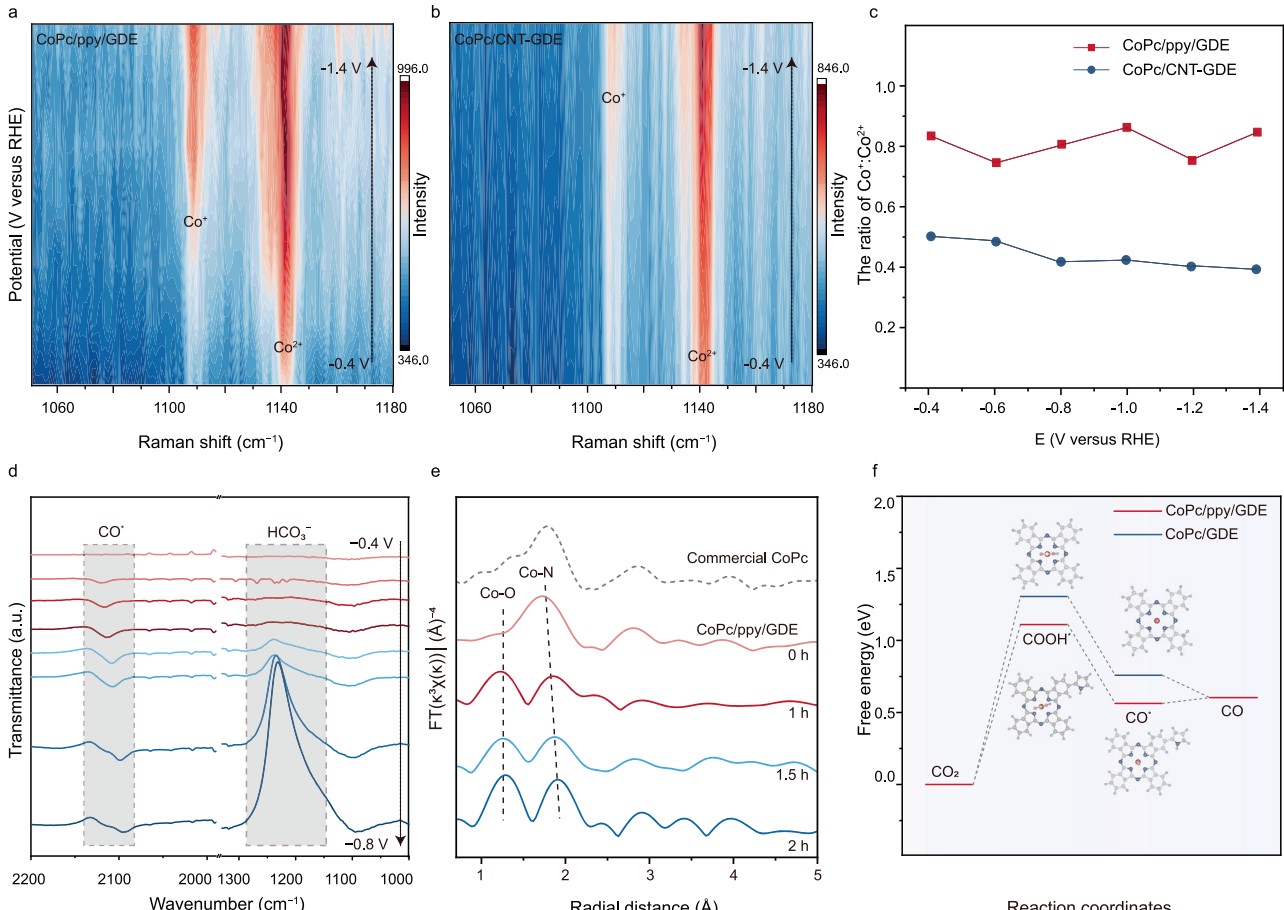

**Fig. 3 | Mechanistic insights into the electrocatalytic CO₂RR on CoPc-based electrodes.** In situ SERS of the **a** CoPc/ppy/GDE and **b** CoPc/CNT-GDE under applied bias ranging from −0.4 to −1.4 V versus RHE in CO₂-saturated 0.1 M KHCO₃. There is no iR correction for voltages. **c** The ratio of Co⁺: Co²⁺ during CO₂RR under applied bias ranging from the −0.4 to −1.4 V versus RHE during CO₂RR. **d** In situ ATR-SEIRAS spectra of CoPc/ppy/GDE collected in CO₂-saturated 0.1 M KHCO₃ with an H-type cell, the potential was applied from −0.4 to −0.8 V versus RHE. There is no iR correction for voltages. **e** EXAFS spectra of CoPc/ppy/GDE polarized at −0. 8 V versus RHE in CO₂-saturated 0.1 M KHCO₃ for different reaction periods; commercial CoPc was used as the reference. There is no iR correction for voltages. Note that these spectra were collected under ex situ conditions, during the CO₂-to-CO conversion the CoPc/ppy/GDE electrode was taken out from the electrochemical cell at designated time and subjected to measurements. Source data are provided as a Source Data file. **f** Calculated free energy diagrams for CO₂RR on CoPc/ppy/ GDE and CoPc/GDE. Raw computational and simulation data are provided as Source Data and Supplementary Data 1.

of 92% with a current density of 14.3 mA cm⁻² at −0.7 V versus RHE, which are slightly inferior to that of CoPc/ppy/GDE; while FePc/ppy/ GDE and CuPc/ppy/GDE demonstrate CO selectivity approaching 90%, accompanied with current densities of 9.3 and 9.8 mA cm⁻², respectively. These results validate the compatibility of our covalent grafting approach with diverse phthalocyanine-based molecular catalysts.

### Probe of active species and mechanistic insights into CO₂-to-CO transformation

Previous studies have established that Co⁺ species in CoPc formed by one-electron reduction of Co²⁺ are the active site for CO₂RR[31,32]. Through the diagnostic Raman signatures of Co⁺ (at ca. 1110 cm⁻¹) and Co²⁺ species (at ca. 1140 cm⁻¹), the identification and quantification of Co⁺ active species have been achieved[33–35]. Following these reports, the evolution of CoPc/ppy/GDE and the distribution of Co⁺ and Co²⁺ species (i.e., the ratio of Co⁺ to Co²⁺) under operating potentials were monitored through in situ surface-enhanced Raman spectroscopy (SERS) in Supplementary Fig. 27. As shown in Fig. 3a, b, characteristic peaks could be resolved at 1113 and 1146 cm⁻¹ at −0.4 V versus RHE for both electrodes of CoPc/ppy/GDE and CoPc/CNT-GDE, which are reported to be associated with the Co⁺ and Co²⁺ species, respectively[35]. The signal intensity of Co⁺ species (at 1113 cm⁻¹) over CoPc/ppy/GDE

becomes prominent at ca. −0.7 V versus RHE, consistent with the redox behavior of Co²⁺/Co⁺ in CoPc; beyond this potential, the spectrum reaches a relatively steady state. In comparison, the formed Co⁺ species over CoPc/CNT-GDE is significantly lower in amount (Supplementary Figs. 28 and 29). Using the Raman peak area, the ratios of Co⁺ to Co²⁺ in CoPc/ppy/GDE and CoPc/CNT-GDE at various applied potentials were quantified. The results in Fig. 3c show that the formed Co⁺ species over CoPc/ppy/GDE are always higher than that of CoPc/CNT-GDE, which aligns with its better catalytic activity for CO generation[36]. It is inferred that the electron donation effects from ppy to CoPc are responsible. Furthermore, the stability of Co⁺ species during CO₂RR was probed by in situ SERS. As shown in Supplementary Fig. 30, with the continuous application of a negative potential of −1.4 V versus RHE, no obvious degradation in the peak intensity of Co⁺ species could be detected on CoPc/ppy/GDE over a reaction period of 30 min, while for CoPc/CNT-GDE, a gradual decrease of Co⁺ species appears, possibly due to the progressive loss of CoPc from CoPc/CNT-GDE electrode.

Having determined that the electron-donating ppy promotes active Co⁺ formation, we employed in situ attenuated total reflection surface enhanced infrared absorption spectroscopy (in situ ATR−SEIRAS) to track the reaction intermediates and gain a molecular-level understanding of CO₂-to-CO conversion over CoPc/ppy/GDE

(Supplementary Fig. 31). For the CoPc/ppy/GDE electrode (Fig. 3d), upon the application of negative potentials, surface-bonded CO (CO$^*$) appears and grows in intensity with increasing negative potential, indicating the gradual transformation of $CO_2$ to CO$^*$[37]. Under the same condition, negligible CO$^*$ signals were observed on ppy/GDE and CoPc/GDE, while CoPc/CNT-GDE prepared according to the previous report showed a delayed onset potential of CO$^*$ formation along with a weaker signal intensity (Supplementary Figs. 32–34). Meanwhile, an inverse absorption band appears at 1230 cm$^{-1}$ and grew in prominence as the applied bias goes more reductive; this phenomenon can be ascribed to the rapid depletion of $HCO_3^-$ at large overpotentials. It is noted that under identical bias, CoPc/ppy/GDE exhibits the largest absorption bands for CO$^*$ and $HCO_3^-$ among tested samples. These observations suggest the improved activity of CoPc/ppy/GDE for $CO_2$RR than the control samples of ppy/GDE, CoPc/GDE and CoPc/CNT-GDE.

To further assess the evolutionary nature of the atomically dispersed Co active sites, ex situ XAS measurements were carried out. Co K-edge XANES spectra of CoPc/ppy/GDE were acquired after reacting in a $CO_2$-saturated 0.1 M $KHCO_3$ solution for different times. Under this condition, oxygen-containing species (OH$^-$, $H_2O$ and reaction intermediates such as $HCO_3^-$, CO$^*$ and COOH$^*$) may adsorb on CoPc/ppy/GDE, and upon removal of the applied potential valence-state increase of the Co centre would occur (Supplementary Fig. 35)[38,39]. Further evidences from the EXAFS analysis reveals a slight elongation of the Co−N bond (from 1.98 to 2.05 Å)[40,41] and importantly the emergence of a Co−O coordination peak at ~1.2 Å (Fig. 3e)[42]. These phenomena are consistent with the coordination of oxygen-containing reactants/intermediates (e.g., OH$^-$, $HCO_3^-$ or COOH$^*$ species) to the Co centre and the correspondingly relaxed structure (elongated Co−N bond) in CoPc/ppy/GDE. In comparison, the blue-shift of absorption edge in the XANES curves and the Co−O coordination in the EXAFS spectra for CoPc/CNT-GDE are not as prominent as those of CoPc/ppy/GDE (Supplementary Fig. 36), indicating weaker adsorption of OH$^-$, $CO_2$, $H_2O$ reactants and less accumulation of oxygen-containing intermediates over CoPc/CNT-GDE than CoPc/ppy/GDE.

To gain more comprehensive mechanistic insights into ppy-enhanced electrocatalytic activity, free energy diagrams for the $CO_2$-to-CO conversion on both CoPc/ppy/GDE and CoPc/GDE were constructed using DFT calculations. As shown in Fig. 3f, the formation of carboxyl intermediate (COOH$^*$) through proton-coupled electron transfer ($CO_2 + H^+ + e^- \rightarrow$ COOH$^*$) is identified as the rate-determining step (RDS) for both electrocatalysts. Crucially, the free energy change for the RDS of CoPc/ppy/GDE (1.11 eV) is decreased by 0.2 eV compared to CoPc/GDE (1.31 eV), indicating the covalent grafting of ppy promotes the formation of COOH$^*$ on CoPc. Charge density difference analysis (Supplementary Figs. 37 and 38) reveal enhanced electron accumulation around COOH$^*$ on CoPc/ppy/GDE, attributed to the electron-donating characteristics of ppy. Thus, this electronic modulation reduces the activation energy of the RDS. Raw simulation data are provided as Supplementary Data 1. Collectively, experimental and theoretical results demonstrate that the incorporated ppy exerts an electron-donating effect on the CoPc centre, which not only facilitates the formation of active Co$^+$ species but also accelerates the reaction kinetics via reducing activation energy for RDS, ultimately boosting the $CO_2$-to-CO conversion activity of CoPc/ppy/GDE.

### $CO_2$RR performance of CoPc/ppy/GDE in an electrolyzer

The direct covalent grafting of pristine molecular catalysts onto GDEs not only maintains the structural advantages of the molecular catalyst but also simplifies the following electrolyzer manufacturing process. To integrate CoPc/ppy/GDE into practical electrochemical devices, an electrolyzer was assembled by using CoPc/ppy/GDE as the cathode and a recently reported CoFe-C$_i$@GQDs/NF as the anode[43]. As depicted by the LSV curve in Fig. 4a, CoPc/ppy/GDE-based MEA (CoFe-C$_i$@GQDs/NF‖CoPc/ppy/GDE) demonstrates excellent activity for

$CO_2$-to-CO conversion, achieving a current density of up to 500 mA cm$^{-2}$ at a cell voltage of ~2.8 V along with a high FE$_{CO}$ of 99%. Importantly, the FE$_{CO}$ values remain above 97% across a wide range of cell voltages. In addition, CoPc/ppy/GDE-based MEA displays operation stability under stringent conditions. As illustrated in Fig. 4b and Supplementary Fig. 39, when operated in 1.0 M KOH, the CoFe-C$_i$@GQDs/NF‖CoPc/ppy/GDE electrolyzer achieves device stability of 50 h at 1 A cm$^{-2}$ and 120 h at 500 mA cm$^{-2}$, corroborating the resistance of CoPc/ppy/GDE electrode against structural collapse/changes under industrially relevant current densities. A comparative analysis with other reported systems is presented in Supplementary Table 4 and Fig. 4c [17,26,44–52], which highlights the unique advantages of CoPc/ppy/GDE for $CO_2$RR. To further assess the industrial applicability of CoPc/ppy/GDE electrode, scalability was executed through electrode area expansion (~9 cm$^2$). As displayed in Supplementary Fig. 40, the scaled-up electrolyzer operated stably at 1 A for 20 h, with a cell voltage of ~2.5 V. Furthermore, a tandem device at the hectowatt scale (180 W) was assembled, which operated at a current density of 1 A cm$^{-2}$ for 300 min with FE$_{CO}$ ranging from 95 to 80% (Supplementary Fig. 41).

The CoFe-C$_i$@GQDs/NF‖CoPc/ppy/GDE electrolyzer was integrated with a triple-junction solar cell (GaInP$_2$/InGaAs/Ge) to assemble an unbiased, solar-driven $CO_2$RR system (Fig. 4d). To synchronize the maximum outputs of solar cell and electrolyzer, the light intensity was incrementally adjusted (Fig. 4e). At the light intensity equivalent to 5 suns, this solar-driven $CO_2$RR device achieves an impressive solar-to-fuel efficiency of 19.2%, with the electrolyzer's operating point (2.23 V and 67 mA cm$^{-2}$) close to solar cell's maximum output (2.31 V and 73 mA cm$^{-2}$). Furthermore, under 5-sun light intensity, this solar-driven device shows essentially stable $CO_2$RR for 50 h, demonstrating the effectiveness and robustness of covalent grafting-derived CoPc/ppy/GDE (Fig. 4f).

## Discussion

Prior to this study, covalent grafting of molecular catalysts onto GDEs relied on those with specially designed functional groups (such as amino and carbazole groups). Towards this challenge, we propose a ppy mediated electro-polymerization strategy to offer robust C−C bonds between GDEs and molecular catalysts, by using pristine CoPc as the example. The resulting CoPc/ppy/GDE electrode possesses combined advantages of robust C−C bonds, binder-free interface, highly conductive ppy linker, appropriate catalyst dispersion, electron donation effects from ppy, and Co single-atom sites with enriched active Co$^+$ species. This unique combination endows the electrode with notable activity and stability for $CO_2$RR compared with previously reported CoPc-based catalysts. By assembling CoPc/ppy/GDE electrode into an electrolyzer, the resulting device achieves the stability of 120 h at 500 mA cm$^{-2}$ and 50 h at 1 A cm$^{-2}$ in alkaline media; further connecting this electrolyzer to a triple-junction solar cell yields a solar-to-CO efficiency of 19.2%. The convenience and robustness of this covalent grafting strategy, along with its good scalability and extendibility, would expand and accelerate the applications of molecular catalysts in various catalytic and energy conversion processes.

## Methods
### Reagents

Iron(III) nitrate nonahydrate (Fe(NO$_3$)$_3$·9H$_2$O, analytical reagent), potassium hydroxide (KOH, analytical reagent) and pyrrole were purchased from Aladdin. 1-butyl-3-methylimidazolium tetrafluoroborate was obtained from TCI. Cobalt nitrate hexahydrate (Co(NO$_3$)$_2$·6H$_2$O) and cobalt phthalocyanine (CoPc) were purchased from Sinopharm Chemical Reagent Co., Ltd. (Shanghai, China). All chemicals were used as received without further purification. A graphene oxide quantum dot (GQD) solution (1 mg mL$^{-1}$) was purchased from Nanjing Xianfeng Nano Co., Ltd. Nickel foam (approximately 1 mm × 1 cm × 2 cm) was carefully cleaned by a specific HCl solution (1 M), ethanol and

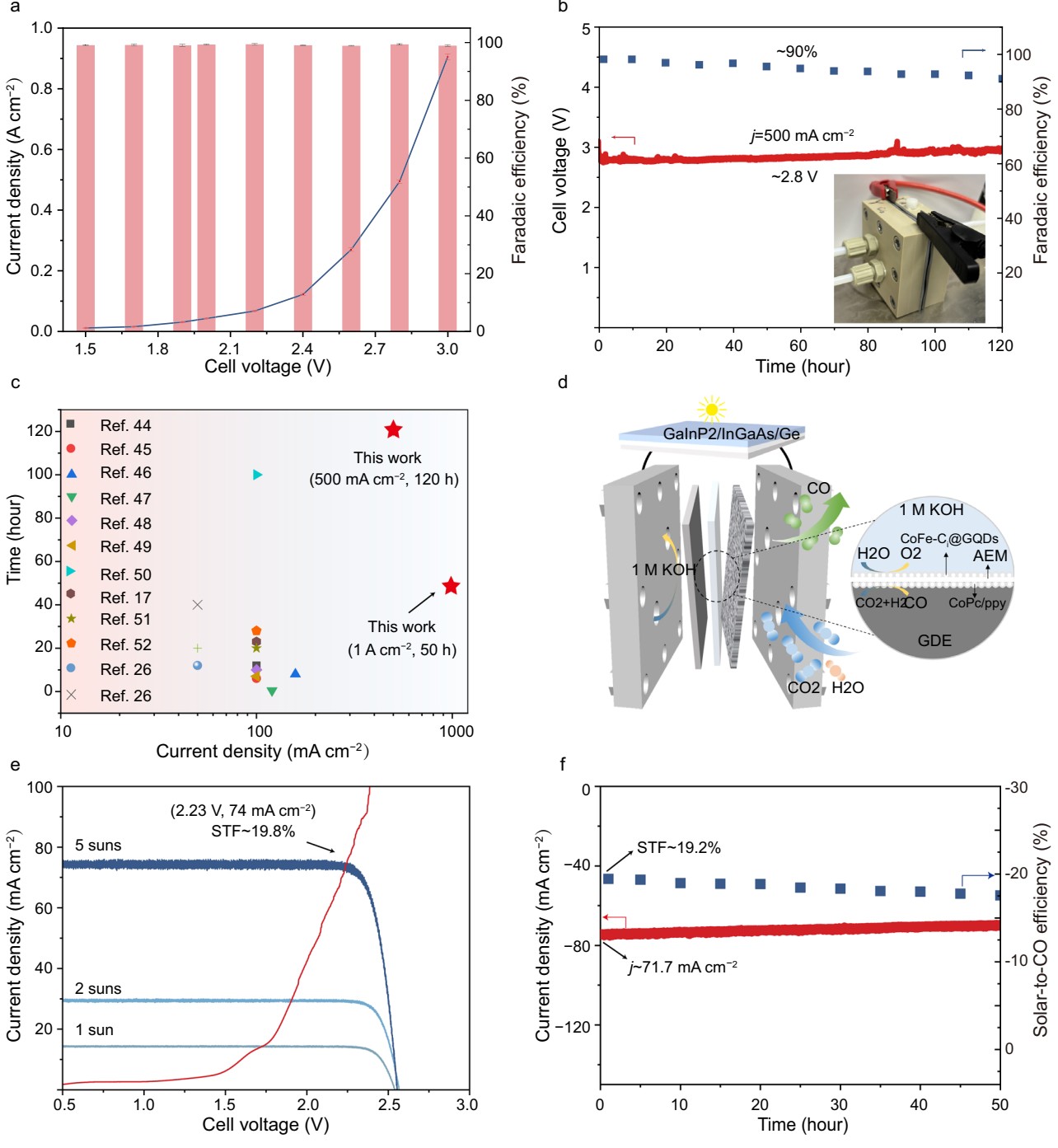

**Fig. 4 | MEA and solar-driven device performance. a** LSV and Faradaic efficiency of $CO_2$-to-CO conversion in the CoFe-$C_i$@GQDs/NF||CoPc/ppy/GDE electrolyzers with 1.0 M KOH electrolyte. **b** Long-term stability test at 500 mA cm⁻² with corresponding Faradaic efficiency of the electrolyzer in 1.0 M KOH. The gas flow rate of MEA is 20 mL/min. **c** A brief summary of electrocatalysts for MEA. Detailed information can be found in Supplementary Table 4. **d** Schematic diagram of the solar-driven device. **e** j-V curves of the triple-junction solar cell (GaInP₂/InGaAs/Ge) and the polarization curve of the CoFe-$C_i$@GQDs/NF||CoPc/ppy/GDE electrolyzers with 1.0 M KOH electrolyte. **f** j-t and corresponding STF efficiency curves of the

CoFe-$C_i$@GQDs/NF||CoPc/ppy/GDE electrolyzers with 1.0 M KOH electrolyte powered by a triple-junction solar cell (GaInP₂/InGaAs/Ge) under 5 suns. The gas flow rate of MEA is 10 mL/min. The illumination area was 1 cm², and the electrode area was 1 cm². The resistance value of MEA is 0.4 ± 0.05 Ω. All measurements were performed at room temperature (23 ± 2 °C). There is no iR correction for voltages. Data are presented as mean values ± standard deviation. The standard deviation is obtained based on three independent samples. Source data are provided as a Source Data file.

deionized water under ultrasonic conditions for 15 min and then dried under vacuum. The gas diffusion electrode (Sigracet 28BC, area: 1 cm × 1 cm) and anion exchange membrane (Sustainion x37-50 Grade RT) were purchased from eChemstore.

## Characterization

Scanning electron microscopy images were obtained with a ZEISS ULTRA55 instrument (accelerating voltage of 20 kV). TEM and high-resolution transmission electron microscopy (HRTEM) images were

obtained on an FEI Tecnai G2 F30. High-angle annular dark field (HAADF) scanning transmission electron microscopy (STEM) images were obtained with a JEOL JEM-ARM200F microscope equipped with a spherical aberration correction system for STEM. XPS measurements were performed using a PHI 5000 Versa Probe (Japan UlVAC-PHI). The binding energies for all the spectra were calibrated with respect to the C 1 s line at 284.6 eV. A LabRAM ARAMIS Raman spectrometer (HORIBA Scientific) was used to measure the composition of the material and the stress state of the chemical bonds. A NEXUS870 Fourier transform infrared (FTIR) spectrometer was used to perform surface chemical analysis in the range of 600–4000 $cm^{-1}$. ICP-OES was performed using an ICP Agilent 7850 (MS). The electrochemical tests were performed with a CS310X instrument (CORRTEST, Wuhan, China) and a M8873 instrument (Maynuo Electronics, Nanjing, China). X-ray absorption spectroscopy was performed via easyXAFS300+ (easyXAFS, USA). The solution-phase products were analysed via Bruker Advance III HD 500 NMR. The gas phase products were analysed by a GC-9720plus (FuLi, Zhejiang, China).

### Synthesis of CoPc/CNT
To prepare CoPc/CNT, 20 mg of CNT were dispersed in 5 mL of pentanol with the assistance of >30 min of sonication. Subsequently, 20 mg of CoPc was added to the homogeneous suspension. The dispersion was subsequently transferred to an autoclave and heated at 180 °C for 2.5 h. The resulting product was collected by centrifugation at 8000 rpm, sequentially washed with ethanol, chloroform, 5% HCl solution, ethanol, and water, and finally vacuum-dried.

### Preparation of the working electrodes of CoPc/ppy/GDE, CoPc/GDE and ppy/GDE
Preparations of CoPc/ppy/GDE: A homogeneous electrolyte was prepared by dissolving 0.010 mol of 1-butyl-3-methylimidazolium tetrafluoroborate in 50 mL of acetonitrile, followed by the addition of pyrrole and CoPc at designated molar ratios. Prior to electro-polymerization, the gas diffusion electrode was pretreated via plasma oxidation under vacuum with controlled oxygen flow using an atomic layer deposition system. Electro-polymerization was carried out in a three-electrode configuration: the pretreated catalyst as the working electrode, a Pt foil counter electrode, and an Ag/AgCl reference electrode (3 M KCl). Cyclic voltammetry was performed between −1.2 V and 1.6 V (versus Ag/AgCl) for 40 cycles at a scan rate of 100 mV s$^{-1}$. The resultant CoPc/ppy/GDE was rinsed thoroughly with acetonitrile and dried under ambient conditions. The active material loading was determined by inductively coupled plasma optical emission spectrometry (ICP-OES) based on cobalt content. Control samples (CoPc/GDE and ppy/GDE) were synthesized under identical electrochemical parameters.

### Preparation of the working electrodes of CoPc/CNT-GDE
A uniform catalyst ink was formulated by dispersing 5 mg CoPc/CNT with 30 μL of 5 wt% Nafion solution in 2 mL of ethanol and vigorously sonicated for 30 min. Then, 200 μL ink was drop cast onto $1 \times 1$ cm$^2$ carbon paper. The loading amount was calculated according to the cobalt content from ICP results.

### Preparation of the working electrodes of CoFe-C$_i$@GQDs/NF
Co(NO$_3$)$_2$·6H$_2$O (2 mmol), Fe(NO$_3$)$_3$·9H$_2$O (2 mmol), Na$_3$C$_6$H$_5$O$_7$·2H$_2$O (0.5 mmol) and urea (5 mmol) were dissolved in 60 mL of deionized water and stirred to form a clear solution. The aqueous solution was transferred to a 100 mL Teflon-lined stainless steel autoclave with NF, maintained at 120 °C for 10 h, and then allowed to cool to room temperature naturally. The samples were rinsed with distilled water and ethanol and then dried at 60 °C under vacuum. The mass loading of the catalyst was approximately 1 mg cm$^{-2}$. One milliliter of GQD solution was diluted with deionized water to 20 mL, transferred to a 50 mL

reactor with dried CoFe-C$_i$/NF, and maintained at 150 °C for 3 h. Afterwards, the obtained CoFe-C$_i$@GQDs/NF was washed and dried.

### CO$_2$RR measurements
The CO$_2$RR performances of the CoPc/ppy/GDE electrode were evaluated and compared to those of the CoPc/CNT-GDE, CoPc/GDE, and ppy/GDE electrodes. The catalytic performances of the electrocatalysts were first evaluated in a customized H-type cell on an electrochemical workstation (CS310M instrument) with two compartments separated by a Nafion 117 proton exchange membrane. The Nafion membrane underwent an immersion treatment in a 5% hydrogen peroxide solution at 80 °C for 30 min, followed by rinsing in deionized water for 10 min, then treatment in 5% sulfuric acid at 80 °C for an hour, and a final deionized water rinse for 30 min. Each compartment contained 10 mL of 0.1 M KHCO$_3$ electrolyte, and the compartment holding the working electrode was prepared to quantify the gaseous products. The electrolyte was purged with CO$_2$ at a flow rate of 5 sccm for 30 min before and during each measurement while stirring. No special storage conditions were needed for the electrolyte beyond room temperature. Ag/AgCl and platinum (Pt) meshes were used as the reference and counter electrodes, respectively. All electrochemical measurements were performed at room temperature (23 ± 2 °C). The flow rate verification was conducted using a bubble flowmeter at the cathode chamber's outlet.

An 85% iR compensation of the solution resistance was applied automatically via the potentiostat. The measured potential was converted to an RHE scale by V (vs. RHE) = V (vs. Ag/AgCl) + 0.197 V + 0.0591 × pH−iR. Where pH ~6.8 ± 0.2 in 0.1 M CO$_2$-saturated KHCO$_3$ solution. The potential of the Ag/AgCl reference electrode was calibrated against a reversible hydrogen electrode (RHE, Gaoss Union, Inc.). Potentials include an 85% iR compensation, with the uncompensated resistance (R$_\Omega$) determined by extrapolating high-frequency impedance data, averaging around 10 ± 2 Ω in 0.1 M CO$_2$-saturated KHCO$_3$.

During constant potential electrolysis, we ran the reaction for at least 30 min before sampling the gaseous products. For the electrochemical measurements in a liquid flow cell, the anolyte and catholyte chambers were both filled with 1.0 M KOH and separated by an anion exchange membrane (Sustainion x37-50 Grade RT) on an electrochemical workstation (CS310M instrument). The Sustainion x37-50 Grade RT was soaked in 1.0 M KOH at room temperature for 24 h to fully convert it to the hydroxide form. CO$_2$ at a constant flow rate of 20 sccm was introduced into the system during each measurement. The gaseous products were quantified via an online GC. Similarly, for MEA testing, the anolyte chamber was filled with 1.0 M KOH electrolyte, while the catholyte chamber was fed with humidified CO$_2$ at a constant flow rate of 20 sccm. The chambers were separated by an anion exchange membrane (Sustainion x37-50 Grade RT) on an electrochemical workstation (CS310M instrument). The MEA testing with 9 cm$^2$ was filled with 1.0 M KOH electrolyte, while the catholyte chamber was fed with humidified CO$_2$ at a constant flow rate of 40 sccm a DC Power Supply (M8873 Maynuo). Besides, the tandem device at the hectowatt scale (180 W) was filled with 1.0 M KOH electrolyte, while the catholyte chamber was fed with humidified CO$_2$ at a constant flow rate of 100 sccm on a DC Power Supply (M8873 Maynuo).

### Product quantification
The Faradaic efficiency (FE) was calculated via the following equation:

$$FE = \frac{zpGFV_{CO}}{JRT} \tag{1}$$

Where $V_{CO}$ was the volume concentration of CO, $J$ was the steady-state total current, $G$ was the CO$_2$ flow rate. The constants were shown as

follows: $z = 2$, $p = 1.013 \times 10^5$ Pa, $T = 298.15$ K, $F = 96485$ C mol$^{-1}$, $R = 8.3145$ J mol$^{-1}$ K$^{-1}$.

The TOF value was calculated via the following equation:

$$TOF = \frac{j \cdot A}{n \cdot F \cdot N_{Co}} \quad (2)$$

where $j$ was the current density (A cm$^{-2}$), $A$ was the geometric electrode area (1 cm$^2$), $n$ represented the electron transfer number for $CO_2$-to-CO conversion ($n = 2$), FF was Faraday's constant (96,485 C mol$^{-1}$), and $N_{Co}$ was the total number of Co sites quantified by ICP-OES. Actual Co loading of the optimized CoPc/ppy/GDE catalyst (obtained with 40 CV cycles) was measured to be 0.0837 wt% with ICP-OES. For our CoPc/ppy/GDE catalyst, we assumed all Co sites participate in the $CO_2$RR reaction. Therefore, the TOF value at 40 mA cm$^{-2}$ was calculated as follows:

$$TOF = \frac{j\left[\frac{mA}{cm^2}\right] \times 10^{-3} \times A[cm^2]}{2 \times 96485 \times N_{Co}}$$

$$TOF = \frac{40 \times \left[\frac{mA}{cm^2}\right] \times 10^{-3} \times 1 \text{ cm}^2}{2 \times 96485 \frac{C}{mol} \times \frac{0.0837\% \times 0.01g}{59\left[\frac{g}{mol}\right]}} = 1.46 \text{s}^{-1}$$

The solar-to-fuel efficiency (STF) was calculated via the following equation:

$$STF = \frac{\Delta E \times j \times FE\%}{P} \quad (3)$$

where $\Delta E$ is the equilibrium potential (1.34 V) for $CO_2$-CO, $j$ is the current density, and $P$ is the incident solar power. A triple junction gallium arsenide battery was purchased from Hasunopto, and the effective area of the chip was 1 cm$^2$.

### Electrochemical characterization

In situ attenuated total reflectance-surface enhanced infrared absorption spectroscopy (ATR-SEIRAS) measurements.

In situ ATR-SEIRAS measurements were determined on a Thermo Scientific Nicolet iS50 instrument (USA) equipped with a liquid nitrogen-cooled HgCdTe (MCT) detector and an in situ attenuated total reflection-FTIR device (Hefei In situ Technology Co., Ltd.). A silicon (Si) prism coated with gold particles (particle size <100 nm) and a catalyst was fixed in a commercial three-electrode H-type cell with an Ag/AgCl reference electrode, a Pt-wire counter electrode, and $CO_2$-saturated 0.1 M KHCO$_3$. The electrolyte was 0.1 M KHCO$_3$ with continuous $CO_2$ bubbling. All electrochemical tests were controlled using a CS310M instrument. There is no iR correction for voltages. All IR spectra were obtained via averaging 32 scans at a spectral resolution of 8 cm$^{-1}$.

### Operando surface-enhanced Raman spectroscopy (SERS) measurements

A LabRAM ARAMIS Raman spectrometer (HORIBA Scientific) equipped with an excitation of 532 nm laser was used to measure the composition of the material and the stress state of the chemical bonds. Operando measurements were conducted in a three-electrode cell, assembled with a catalyst-loaded working electrode, a Pt wire counter electrode, an Ag/AgCl reference electrode, and $CO_2$-saturated 0.1 M KHCO$_3$. Before the tests, all working electrodes were sputtered with a layer of nano-gold particles (particle size <100 nm). All electrochemical tests were controlled using a CS310M instrument. There is no iR correction for voltages. The Raman frequencies were calibrated by a Si wafer.

### Computational methods

All the theoretical calculations are performed with a plane-wave basis set as implemented in the Vienna ab initio simulation package (VASP)[53]. The general gradient approximation with Perdew–Burke–Ernzerhof[54] was employed to describe the exchange-correlation potential[55]. DFT + U was chosen for this study, in which the empirical value of U was set to 3.32 eV. A 30 Å vacuum layer was constructed to prevent interactions between layers[56,57]. The cut-off energy for the plane wave was set to 500 eV, and the Brillouin zone used a $1 \times 1 \times 1$ mesh of special k points[58].

The Gibbs free energy (G) of all the reaction steps was calculated via the following equation:

$$G = E_{DFT} + E_{ZPE} + \int C_p dT - TS \quad (4)$$

where $E_{DFT}$, $E_{ZPE}$, $\int C_p dT$, and TS are the free energies calculated via DFT, zero-point energy, enthalpic temperature correction, and entropic temperature correction, respectively.

## Data availability

The data that support the findings of this study are available within the article and its Supplementary Information and from the corresponding author upon reasonable request. Source data are provided with this paper.

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

## Acknowledgements

We thank the National Science Fund for Distinguished Young Scholars (22025202, Z.S.L.), the National Natural Science Foundation of China (22379065, J.Y.F.), the Natural Science Foundation of Jiangsu Province of China (BK20232021, Z.G.Z.), the National Key Research and Development Program of China (2021YFA1502100, J.Y.F.) and the Fundamental Research Funds for the Central Universities (KG202505) for financial support. We are grateful to the High-Performance Computing Center of Nanjing University for performing the numerical calculations in this paper on its blade cluster system.

## Author contributions

Z.S.L. and J.Y.F. conceived the idea and directed the project. H.H.Y. carried out the synthesis and characterization of the samples and the $CO_2$-to-CO conversion experiments. G.X.H. carried out the theoretical calculations, and R.L.F. provided the anode catalyst. M.Y.Z. and Z.X.L. performed the Raman characterization. H.T.H. and B.G. tested the solar cell. H.H.Y., J.Y.F. and Z.S.L. analysed the data. H.H.Y. wrote the manuscript. J.Y.F. and Z.S.L. revised the manuscript. Z.G.Z. provided advice.

## Competing interests

The authors declare no competing interests.
