## [Transparent Peer Review file · Nature Communications]

In situ covalent grafting of pristine cobalt phthalocyanine on gas diffusion electrodes enables CO₂ reduction at ampere-level current density

Corresponding Author: Professor Zhaosheng Li

Version 0:

Reviewer comments:

Reviewer #1

(Remarks to the Author)

The manuscript reports an in situ growth method for immobilizing molecular catalysts on gas diffusion electrodes (GDE) through electrochemical covalent grafting at room temperature, with the goal of enhancing the stability and catalytic performance of cobalt phthalocyanine (CoPc). This approach demonstrates promising CO₂-to-CO conversion efficiency and claims a record solar-to-fuel efficiency when integrated with a photovoltaic-electrolysis device. While this work suggests a potentially valuable contribution to CO₂ reduction technology, it lacks sufficient structural characterization and fails to convincingly establish the novelty required for publication in Nature Communications. The main concerns are as follows:

1. The manuscript lacks direct evidence supporting the formation of covalent bonds between CoPc and the polypyrrole (ppy) network. The FTIR spectra provided merely confirm the presence of CoPc and ppy/GDE individually but do not substantiate the claimed covalent interactions between CoPc molecules and the ppy matrix. To address this, the authors should incorporate additional spectroscopic techniques to directly confirm and visualize the proposed covalent bonding. Without such evidence, the structural integrity and uniqueness of the CoPc/ppy@GDE system remain speculative.

2. The authors suggest that the lower frequency of the CO* peak on CoPc/ppy@GDE, compared to CoPc/CNT, reflects stronger CO* adsorption and greater CO coverage. However, it is unclear why the Co site would not undergo reduction to a zero-valence state under electrochemical reduction conditions. This assumption appears inconsistent with the Co K-edge spectra, which indicate an increase in the Co valence state during the reaction. The manuscript would benefit from a more detailed mechanistic explanation or additional experimental data to reconcile this apparent discrepancy in the behavior of the Co(I) site during CO₂RR.

3. Although the study reports promising catalytic stability at high current densities, further structural analysis of the CoPc/ppy@GDE catalyst before and after extended catalytic testing is essential to validate its durability.

4. The electropolymerization approach for anchoring ppy on carbon supports is well-established and widely applied in CO₂ reduction reactions. The manuscript does not sufficiently explain what differentiates CoPc/ppy@GDE from similar electropolymerized systems reported in the literature. The authors are encouraged to provide a clear comparative analysis to highlight any unique structural characteristics or performance advantages of CoPc/ppy@GDE.

Reviewer #2

(Remarks to the Author)

The authors address the immobilization limitations of phthalocyanine catalysts on electrode towards industrial-scale CO₂ reduction, and graft CoPc onto gas diffusion electrode through polypyrrole, which was applied in membrane electrode assembly electrolyser for CO₂ to CO. The present work was well organized with large characterization including XAS, in situ technology, and DFT calculation. However, there are still problems need further revision.

1. How to know the molecular weight? Whether it influence the performance of CoPc/ppy@GDE catalyst? And how to ascertain the best polypyrrole?
2. Line 136-139. Since the electron density of Co is enhanced, the binding energy should blueshift, while not redshift. This part needs reconsideration. The result is not consistent with XAS analysis.
3. The fig.2a, it should be labelled for different color for better understanding. The Fig. 2f.g caption was not correct.
4. Line 153, the pH needs further tested, it cannot be 8.3.
5. How TOF was calculated according to the all Co sites? i.e. how to ascertain the Co sites?
6. Line 241, from the calculation, we can only know that ppy alters the catalytic thermodynamics while not kinetic.
7. Fig. 4f in present is not quite convincing.
8. Since former work has published with CoPc cross-linked ppy(ChemSusChem 2022,15(23):e202201455), CoPc incorporated with ppy(ELECTROCHEMICAL AND SOLID STATE LETTERS 2009,12(8):E17-E19), what's the novelty of present work?

Reviewer #3

(Remarks to the Author)

The authors presented a study on molecular catalyst immobilization through direct covalent grafting, demonstrating that CoPc anchored via pyrrole linkers (CoPc/ppy@GDE) exhibits enhanced stability and catalytic performance. The evidence from XAFS and XPS provides interesting insights into the electronic effects of pyrrole grafting on the Co center, while in-situ SERS and SEIRAS offer valuable mechanistic information about Co oxidation states and their role in the reaction pathway. I believe the fundamental observation of enhanced stability through pyrrole-based grafting is well-supported; However, I have a number of significant scientific and technical concerns that should be addressed in a major revision. I think the work could be suitable for publication after the authors strengthen their mechanistic understanding and provide more comprehensive evidence. Here are some detailed comments:

1. The manuscript lacks a direct comparison of electrochemical impedance between CoPc@GDE and CoPc/ppy@GDE systems. To better understand the electron transfer characteristics of these catalysts, please provide Nyquist plots along with the corresponding serial resistance values for both systems.
2. In lines 130-133, the notation system using Co(I)-N and Co(II)-N creates confusion in interpreting the oxidation states of cobalt. For example, Co(I)-N in CoPc@GDE can be interpreted specifically refers to Co(+1) coordinated with nitrogen. Please provide clearer terminology to distinguish the different cobalt species.
3. There appears to be a typographical error in line 135 where "XAENS" should be corrected to "XANES"
4. While Figure 2h demonstrates performance comparisons between CoPc/ppy@GDE, CoPc-GDE, and CoPc-CNT, the data appears to be collected using different cobalt loadings. Additionally, there is a discrepancy in the Co content values reported in the main text versus Supplementary Table 1. To make a fair assessment of catalyst performance, please provide comparisons with normalized cobalt content across all catalyst systems.
5. The manuscript attributes the redshifted and broadened CO* bands observed on CoPc/ppy@GDE to surface coverage effects. However, two important questions need to be addressed: First, could dynamic dipole coupling of CO* produce similar spectral changes in single-atom catalysts? Second, the cited reference 37 indicates that high surface coverage typically results in blue-shifted CO* bands, which contradicts the current explanation. Please clarify these apparent discrepancies.
6. The time-dependent blue shift in absorption edges shown in Figure 3d indicates a gradual increase in cobalt valence state during the reaction. This observation seems to contradict the proposed mechanism where catalyst stability and activity are attributed to maintained Co⁺ active species. To resolve this inconsistency, please provide time-resolved in-situ SERS data of CoPc/ppy@GDE at fixed potential.
7. The authors showed the changes of CoPc/ppy@GDE at the fixed potential over the time and discuss the changes of the XANES and EXAFS spectra. However, control sample CoPc@GDE is also required to be measured over the same period of time to investigate whether the changes of the Co oxidation states are associated with ppy introduction.
8. The author mentioned that "The absorption edges blueshifted with reaction time, indicating that the valence state of Co gradually increased during the reaction." Is it reversible? If so, after the reaction, (post-analysis) showed the similar states of Co or does it go back to the initial states. If not, the authors want to discuss how the ppy helps the charge transfer. The changes of the oxidation states can be an indication of the degradation of the CoPc. Show the post-analysis, (TEM, EXAFS, XANES, XPS) can provide the information whether the catalyst maintained the initial CoPc/ppy structure.
9. Can it be applicable to other Pc such as NiPc and FePc for electrochemical CO₂RR?
10. How the current density affected by the loading amount of the catalyst? That can be the useful information.

Reviewer #4

(Remarks to the Author)

This manuscript reported the synthesis of CoPc for the efficient conversion of CO₂ to CO product. A photovoltaic-electrolysis device is also constructed and achieved a record solar-to-fuel efficiency of 19.2%. Despite this work provides a complete study, the novelty of the electrocatalytic CO₂ to CO is not enough. Moreover, the electrolysis in alkaline electrolytes under high current density do not provide new insight. Whether this catalyst is suitable for MEA testing under pure water conditions. Thus, the novelty and depth of academic significance does not reach the level of Nature Communications.

Version 1:

Reviewer comments:

Reviewer #1

(Remarks to the Author)

After reviewing the manuscript, I acknowledge the authors' efforts in conducting additional experiments and expanding the discussion. Although similar systems have been reported in the literature, this work presents some performance advantages. Given the sound methodology and the relevance of the results, I recommend the manuscript for publication.

Reviewer #2

(Remarks to the Author)

The authors have revised the ms according to the comments and can be accepted. Still, there are some grammar and words needing further revision, e.g. times or time?

Reviewer #4

(Remarks to the Author)

Although the authors have addressed some of the previous comments, the overall quality of the manuscript still does not meet the high publication standards of Nature Communications. Therefore, I recommend submitting this work to a more specialized journal. Specific suggestions for improvement include:

1. While the revised manuscript provides additional data, the fundamental mechanism underlying the enhanced activity and stability of the CoPc/ppy/GDE system remains insufficiently elucidated. For instance: the role of polypyrrole in modulating the electronic structure of CoPc is discussed, but direct experimental evidence to confirm the proposed electron-donation effect is still lacking.
2. The claim that C–C covalent bonds are formed between CoPc and ppy relies heavily on indirect evidence. More rigorous characterization is needed to unambiguously verify the bonding configuration.
3. The manuscript emphasizes the novelty of covalent grafting for pristine CoPc, but the performance metrics do not significantly surpass recent reports using functionalized molecular catalysts. The stability data is commendable but could be further contextualized against industrial benchmarks.
4. The post-reaction characterization suggests structural retention, but the origin of eventual performance decay is not investigated. Long-term stability tests under intermittent operation would strengthen the practical relevance.
5. The DFT analysis of CO₂ reduction energetics is oversimplified. Key aspects are overlooked: How does the ppy linker influence the local pH or CO₂ concentration at the catalyst surface? Are the proposed active sites (Co⁺) stable under alkaline conditions, given prior reports of CoPc demetallation at high pH?

Version 2:

Reviewer comments:

Reviewer #4

(Remarks to the Author)

I thank the authors for their thorough revisions in response to the feedback. They have addressed the majority of the concerns raised. I have no further issues and believe the manuscript is now suitable for publication.

Responses to Editor and Reviewers' Comments

(manuscript ID: NCOMMS-24-65311-T)

General response:

We express sincerely thank for the editor, editorial staff and all reviewers for your valuable feedbacks that we have based on to further enhance the quality and completeness of our manuscript. The manuscript has been modified point-by-point after addressing all the suggestions as listed below. (Our response is given in blue and the corrections in the revised manuscript are highlighted with a yellow background).

Response to the Editor Office:

We agree with the importance of validating our findings under practical conditions and will conduct additional activity and stability tests in a scaled-up MEA with an electrode area of 9 cm² (Figure R1). These new experiments will be analyzed, and integrated into the revised manuscript to strengthen the applicability of our conclusions.

Figure R1. **a**, LSV and Faradaic efficiency of CO₂-to-CO conversion in the CoFe-C_i@GQDs/NF||CoPc/ppy@GDE electrolyzers with electrode area of 9 cm² in 1.0 M KOH. **b**, Stability test at 1 A with corresponding Faradaic efficiency of the electrolyzer in 1.0 M KOH.

Reviewer #1:

Comment: The manuscript reports an in situ growth method for immobilizing molecular catalysts on gas diffusion electrodes (GDE) through electrochemical covalent grafting at room temperature, with the goal of enhancing the stability and catalytic performance of cobalt phthalocyanine (CoPc). This approach demonstrates promising CO₂-to-CO conversion efficiency and claims a record solar-to-fuel efficiency when integrated with a photovoltaic-electrolysis device. While this work suggests a potentially valuable contribution to CO₂ reduction technology, it lacks sufficient structural characterization and fails to convincingly establish the novelty required for publication in Nature Communications. The main concerns are as follows:

Comment 1: The manuscript lacks direct evidence supporting the formation of covalent bonds between CoPc and the polypyrrole (ppy) network. The FTIR spectra provided merely confirm the presence of CoPc and ppy/GDE individually but do not substantiate the claimed covalent interactions between CoPc molecules and the ppy matrix. To address this, the authors should incorporate additional spectroscopic techniques to directly confirm and visualize the proposed covalent bonding. Without such evidence, the structural integrity and uniqueness of the CoPc/ppy@GDE system remain speculative.

Reply to comment 1: Thank you for your valuable comments on our manuscript. We understand your concerns regarding the direct evidence for the formation of covalent bonds between CoPc and the ppy network. We have taken your suggestions seriously and have conducted additional experiments and analyses to address this issue.

First, we utilized in situ Raman spectroscopy to monitor the structural evolution of CoPc during the electro-polymerization on GDE. As shown in Figure R2, the Raman spectra reveal gradually increased peaks that correspond to the characteristic vibrational modes of CoPc. Specifically, the macrocyclic deformation mode at 680 cm⁻¹ exhibits an increase trend with the number of CV polymerization cycles, quantitatively correlating with the progressive deposition of CoPc. We also observed a shift of Co–N vibration at ca. 755 cm⁻¹, which is possibly associated with the interfacial strain-induced changes of Co–N bonds in CoPc during polymerization¹. Collectively, the progressively increased intensities of CoPc-related peaks with the number of CV cycles², provide evidence for the in situ growth and integration of CoPc onto the GDE.

Second, to provide further evidence for the covalent integration of CoPc with ppy on the GDE, CV cycles were systematically performed within under controlled potential windows. As shown in Figure R3, ppy generated from electrochemical polymerization of pyrrole shows only broad and featureless oxidation and reduction waves in the potential range of –1.0 to 1.0 V vs. Ag/AgCl. Comparatively, three redox couples could be resolved for CoPc-modified GDE (CoPc/GDE), with their locations presented in Table R1. These redox couples are probably associated with the redox behaviours of Co centres of CoPc (Co²⁺ ↔ Co⁺ ↔ Co⁰). Interestingly, CoPc/ppy/GDE electrode produced in the presence of both pyrrole and CoPc shows similar redox characteristics to that of CoPc/GDE, with however negatively shifted peak positions and largely enhanced current response in the cathodic scan. Based on this observation, we infer that during pyrrole polymerization CoPc

is integrated into ppy, with the configuration of CoPc terminated at the ppy chain; electron donation occurs from ppy to CoPc, thus yielding negatively shifted redox peaks relative to bare CoPc.

Figure R2. In situ Raman spectra of CoPc/ppy/GDE collected at different CV cycles in 0.1 M [BMIM]BF₄ with 4 mM pyrrole and 1 mM CoPc.

Third, we used F-doped tin oxide (FTO) as the conductive substrate to study the grow process of CoPc/ppy, which would exclude the influence of carbon-based substrates. However, the weak bonding between FTO and CoPc/ppy led to easy detachment of product. We also tried to monitor the deposition process of CoPc/ppy through the microcrystal quartz balance, but no available signals could be collected. After these attempts, we collected FTIR spectrum of CoPc/ppy/GDE and compared to those of CoPc and ppy/GDE. To avoid the possible contamination of free CoPc (i.e., CoPc adsorbed on GDE or ppy), CoPc/ppy/GDE was thoroughly cleaned by ultrasonication and washed by copious amounts of acetonitrile. As shown in Figure R4, the FTIR spectrum of CoPc/ppy/GDE shows a combined feature of CoPc and ppy/GDE, which indicates that CoPc had been successfully incorporated into ppy/GDE.

Fourth, CoPc/ppy/GDE electrode exhibits an exceptional 200 hours stability during chronoamperometry measurements at -0.78 V versus RHE, while the control sample of CoPc/CNT-GDE shows only a considerably shorter operational lifespan of 12 hours. After 200 hours stability test, no obvious valence state, structure and coordination environment changes were detected over CoPc/ppy/GDE by XPS, HAADF-STEM and XAS. Furthermore, the as-assembled CoFe-Ci@GQDs/NF||CoPc/ppy/GDE electrolyzer achieves record device stability of 1000 mA cm^{-2} and 120 hours at 500 mA cm^{-2} .

All these experimental observations point to the formation of covalent bonds between CoPc and the ppy network, based on which improved activity (due to electron transfer from ppy to CoPc) and robust operational stability are simultaneously achieved.

Figure R3. Comparative CV patterns of ppy/GDE in 0.01 M [BMIM]BF₄ with 4 mM pyrole, CoPc/GDE in 0.01 M [BMIM]BF₄ with 1 mM CoPc and CoPc/ppy/GDE in [BMIM]BF₄ with 4 mM pyrole and 1 mM CoPc at 100 mV s⁻¹.

Figure R4. FTIR spectra of CoPc/GDE, CoPc/ppy/GDE and ppy/GDE.

Table R1. The location of the CV peaks

Catalysts	Reduction 1	Reduction 2	Reduction 3
CoPc/ppy/GDE	0.52	-0.04	-0.98
CoPc/GDE	0.61	0.03	-0.88
	Oxidation 1	Oxidation 2	Oxidation 3
CoPc/ppy/GDE	-0.27	0.36	1.18
CoPc/GDE	-0.15	0.48	1.32

We have incorporated these findings into the revised manuscript and have provided detailed explanations and references to support our conclusions. We hope that these additional analyses

would address your concerns and strengthen the scientific validity of our work.

“We performed following experiments and analyses to understand the growth process of CoPc/ppy/GDE electrode. The cyclic voltammetry (CV) characteristics of CoPc/ppy/GDE and control samples were monitored in the potential window of -1.2 to 1.6 V versus Ag/AgCl at various scan rates. As shown in Fig. 1c, ppy/GDE generated from electro-polymerization of pyrrole exhibits only broad and featureless oxidation and reduction waves in the potential range of -1.0 to 1.0 V versus Ag/AgCl. Comparatively, three redox couples could be resolved for CoPc-modified GDE (CoPc/GDE), with their locations presented in Fig. 1d. These redox couples are probably associated with the redox behaviours of Co centres in CoPc ($\text{Co}^{2+} \leftrightarrow \text{Co}^+ \leftrightarrow \text{Co}^0$). Interestingly, CoPc/ppy/GDE electrode produced in the presence of both pyrrole and CoPc shows similar redox characteristics to that of CoPc/GDE, with however negatively shifted peak positions and largely enhanced current response in the cathodic scan (Fig. 1d and Supplementary Fig. 1).

Next, we utilized in situ Raman spectroscopy to monitor the structural evolution of CoPc/ppy/GDE during the electro-polymerization process. The Raman spectra reveal gradually increased peaks that correspond to the characteristic vibrational modes of CoPc. Specifically, the macrocyclic deformation mode at 680 cm^{-1} exhibits an increase trend with the number of CV electro-polymerization cycles, quantitatively correlating with the progressive deposition of CoPc^{22,23}. We also observed a shift of Co–N vibration at $\sim 755\text{ cm}^{-1}$, which is possibly associated with the interfacial strain-induced changes of Co–N bonds in CoPc during electro-polymerization (Supplementary Fig. 2). Collectively, the progressively increased intensities of CoPc-related peaks with the number of CV cycles, provide evidences for the in situ growth and integration of CoPc onto the GDE.

Third, we collected FTIR spectrum of CoPc/ppy/GDE and compared to those of CoPc and ppy/GDE. To avoid the possible influence of free CoPc (i.e., CoPc adsorbed on GDE or ppy), CoPc/ppy/GDE was thoroughly cleaned by ultrasonication and washed by copious amounts of acetonitrile. As shown in Supplementary Fig. 3, the FTIR spectrum of CoPc/ppy/GDE possesses a combined feature of CoPc and ppy/GDE, which indicates that pristine CoPc has been successfully incorporated into ppy/GDE²⁴.

Based on these observations (and the exceptional operation stability of CoPc/ppy/GDE as discussed below), we infer that during pyrrole electro-polymerization pristine CoPc is integrated into ppy, with the configuration of CoPc terminated at the ppy chain; electron donation occurs from ppy to CoPc, thus yielding negatively shifted redox peaks relative to bare CoPc. Such inferences are further supported by density functional theory (DFT) calculations, in which strong chemical bonds form in ppy/GDE and CoPc/ppy/GDE; under this condition, ppy serves as an effective linker to realize robust covalent grafting of pristine CoPc onto GDE (Supplementary Figs. 4–6).”

References

[1] Szybowicz, M. et al. Micro-Raman spectroscopic investigations of cobaltphthalocyanine thin

films deposited on quartz and diamond substrates. *Cryst. Res. Technol.* **45**, 1265–1271 (2010).

[2] Chen, Y. T. et al. Charge transfer and electromagnetic enhancement processes revealed in the SERS and TERS of a CoPc thin film. *Nanophotonics*, **8**, 1533–1546 (2019).

Comment 2: The authors suggest that the lower frequency of the CO* peak on CoPc/ppy@GDE, compared to CoPc/CNT, reflects stronger CO* adsorption and greater CO coverage. However, it is unclear why the Co site would not undergo reduction to a zero-valence state under electrochemical reduction conditions. This assumption appears inconsistent with the Co K-edge spectra, which indicate an increase in the Co valence state during the reaction. The manuscript would benefit from a more detailed mechanistic explanation or additional experimental data to reconcile this apparent discrepancy in the behavior of the Co(I) site during CO₂RR.

Reply to comment 2: We appreciate your insightful comments regarding the CO* adsorption behaviour and the valence state of Co in CoPc/ppy/GDE during CO₂RR. According to your suggestions, we have conducted additional experiments and analyses to provide a more comprehensive understanding of the underlying mechanisms.

We employed in situ ATR-SEIRAS to monitor the CO₂ evolution pathway over CoPc/ppy/GDE. As shown in Figure R5, upon the application of negative potentials, surface-bonded CO (CO*) appears and grows in intensity with increasing negative potential, indicating the gradual transformation of CO₂ to CO. Under the same condition, negligible CO* signals were observed on ppy/GDE and CoPc/GDE, while CoPc/CNT-GDE prepared according to the previous report showed a delayed onset potential of CO* formation along with a weaker signal intensity. These observations suggest the improved activity of CoPc/ppy/GDE for CO₂RR than the control samples of ppy/GDE, CoPc/GDE and CoPc/CNT-GDE (Figure R6). Regarding the red-shift of CO* vibrational frequency than those of control samples, we have deleted these descriptions, because after careful checking the data, we found that no direct relationship could be established between the shift degree of CO* frequency and its formation rate.

Figure R5. In situ ATR-SEIRAS spectra of CoPc/ppy/GDE collected in CO₂-saturated 0.1 M KHCO₃ with an H-type cell, the potential was applied from -0.4 to -0.8 V versus RHE.

Figure R6. In situ ATR-SEIRAS spectra of (a) CoPc/CNT-GDE (b) CoPc/GDE, and (c) ppy/GDE collected in CO₂-saturated 0.1 M KHCO₃ with an H-type cell, the potential was applied from -0.4 to -0.8 V versus RHE.

Then, according to your suggestion, we tried to understand the unusual increase of Co valence state during reaction. Indeed, under highly reductive conditions, Co²⁺ sites are expected to undergo a reduction event, leading to the formation of Co⁺ or even Co⁰. After revisiting the measurement procedures, we rationalize the absence of Co⁰ centre and unusual increase of Co valence state during reaction to the following reasons.

First, we should make it clear that our XAS measurements were performed under ex situ conditions on an easyXAFS setup (Figure R7), due to the low intensity of collected signals with a conventional electrochemical cell; the CoPc/ppy/GDE electrode was polarized at -0.8 V versus RHE, and at different reaction times the CoPc/ppy/GDE electrode was taken out from the electrochemical cell and subjected to XAS measurements. Under this condition, oxygen-containing species (OH⁻, H₂O and reaction intermediates such as HCO₃⁻ and COOH*) may adsorb on CoPc/ppy/GDE, and upon removal of the applied potential valence-state increase of the Co centre would occur (Figure R8a)¹. More evidences come from the EXAFS analysis, which reveals a slight elongation of the Co–N bond (from 1.98 to 2.05 Å) and importantly the emergence of a Co–O coordination peak at ~ 1.2 Å (Figure R8b).

These phenomena are consistent with the coordination of oxygen-containing reactants/intermediates (e.g., OH⁻, HCO₃⁻ or HCOO* species) to the Co centre and the correspondingly relaxed structure (elongated Co–N bond) in CoPc/ppy/GDE. In comparison, the blue-shift of absorption edge in the XANES curves and the Co–O coordination in the EXAFS spectra for CoPc/CNT-GDE are not as prominent as those of CoPc/ppy/GDE (Figure R9), indicating weaker adsorption of OH⁻, CO₂, H₂O reactants and less accumulation of oxygen-containing intermediates over CoPc/CNT-GDE than CoPc/ppy/GDE. DFT calculations provide additional evidences, which reveal that Co sites in CoPc/ppy/GDE exhibit a strong interaction with intermediates, such a feature is crucial for accelerating CO₂ conversion (Figure R10). We attribute the enhanced stabilization of reaction intermediates on CoPc/ppy/GDE than CoPc/CNT-GDE to the electron donation effects from ppy to CoPc.

Figure R7. Picture of the easyXAFS/XES setup.

Figure R8. Co K-edge (a) XANES and (b) EXAFS spectra of CoPc/ppy/GDE polarized at -0.8 V versus RHE for different reaction periods; commercial CoPc was used as the reference. Note that these spectra were collected under ex situ conditions, during the CO_2 -to-CO conversion the CoPc/ppy/GDE electrode was taken out from the electrochemical cell at designated times and subjected to measurements.

Figure R9. Co K-edge (a) XANES and (b) EXAFS spectra of CoPc/CNT-GDE polarized at -0.8 V versus RHE for different reaction periods; commercial CoPc was used as the reference. Note that these spectra were collected under ex situ conditions, during the CO_2 -to- CO conversion the CoPc/CNT-GDE electrode was taken out from the electrochemical cell at designated times and subjected to measurements.

Figure R10. Schematic diagram of differential charge density of CoPc/ppy and Bader charge analysis. The cyan (yellow) region shows electron loss (gain).

Second, researchers have claimed that Co^+ species in CoPc formed by one-electron reduction of Co^{2+} , is the active site for CO_2RR ^{2,3}. Through the diagnostic Raman signatures of Co^+ (at ca. 1110 cm^{-1}) and Co^{2+} species (at ca. 1140 cm^{-1}), the identification and quantification of Co^+ active species have been achieved⁴. Following these reports, we have also monitored the evolution of CoPc/ppy/GDE and the distribution of Co^+ and Co^{2+} species (i.e. the ratio of Co^+ to Co^{2+}) with applied potentials through in situ surface-enhanced Raman spectroscopy (SERS). As shown in Figure R11, at -0.4 V versus RHE, characteristic peaks could be resolved at 1110 and 1140 cm^{-1} on both electrodes of CoPc/ppy/GDE and CoPc/CNT-GDE, which are reported to be associated with the Co^+ and Co^{2+} species, respectively^{5,6}. The signal intensity of Co^+ species (at 1110 cm^{-1}) over CoPc/ppy/GDE becomes prominent at ca. -0.7 V versus RHE, consistent with the redox behaviour of $\text{Co}^{2+}/\text{Co}^+$ in CoPc; beyond this potential, the spectrum reaches a relatively steady state. In

comparison, the formed Co^+ species over CoPc/CNT-GDE is significantly lower in amount. Using the Raman peak area, we calculated the ratios of Co^+ to Co^{2+} in CoPc/ppy/GDE and CoPc/CNT-GDE at various applied potentials. The results in Figure R12 show that the formed Co^+ species over CoPc/ppy/GDE are always higher than that of CoPc/CNT-GDE, for which we infer that the electron donation effects from ppy to CoPc is responsible⁷.

Figure R11. a, b, In situ SERS of the CoPc/ppy/GDE and CoPc/CNT-GDE during CO_2RR under applied bias ranging from -0.4 to -1.4 V versus RHE.

Figure R12. The ratio of $\text{Co}^+:\text{Co}^{2+}$ during CO_2RR under applied bias ranging from -0.4 to -1.4 V versus RHE in CO_2 -saturated 0.1 M KHCO_3 .

In the previous study, Co^+ species in CoPc is reported to be stable up to the potential of -2.2 V versus RHE, as the percentage of Co^+ species reaches a steady state starting from ca. -1.0 V versus RHE⁴. Our results under operando conditions in Figure R11 also indicate that no obvious reduction of Co^+ species occurs up to -1.4 V versus RHE. For these phenomena, we infer that the presence of oxygen-containing species (OH^- , H_2O and reaction intermediates such as HCO_3^- and HCOO^*) may stabilize the Co centre in the Co^+ form under operational conditions. In addition, several recent studies propose that high-valence metal single-atom sites (e.g., Fe^{3+} , Sn^{4+}) with specific

coordination environments can be stable and dominate the CO₂RR activity. For instance, in Fe³⁺ single atom catalyst, Fe³⁺ is stabilized on N-doped carbon carriers via pyrrole-nitrogen coordination⁸. It maintains a +3 oxidation state during CO₂ reduction, with its activity attributed to the optimized adsorption of COOH* intermediate at Fe³⁺. Similarly, Sn⁴⁺ single atom catalysts synthesized in a CuO matrix could achieve 98% CO selectivity, due to the Sn⁴⁺-O₃-Cu⁺ coordination promoted high-valence state of Sn and correspondingly accelerated CO desorption⁹. We have also evaluated the stability of Co⁺ species during CO₂RR using in situ SERS. As shown in Figure R13, with the continuous application of a negative potential of -1.4 V versus RHE, no obvious degradation in the peak intensity of Co⁺ species could be detected on CoPc/ppy/GDE over a reaction period of 30 min; while for CoPc/CNT-GDE, a gradual decrease of Co⁺ species appears, possibly due to the progressive loss of CoPc from CoPc/CNT-GDE electrode.

Figure R13. Time-dependent in situ SERS of the (a) CoPc/ppy/GDE (b) CoPc/CNT-GDE during CO₂RR from 0 to 30 min at -1.4 V versus RHE in CO₂-saturated 0.1 M KHCO₃.

Third, we collected the XAS and XPS spectra before and after the stability test in the Co 2p region. As shown in Figures R14 and 15, after 200 hours reaction no sign for the formation of zero-valence Co could be detected.

Figure R14. **a**, XAS spectra at Co K-edge of CoPc/ppy/GDE and **b**, Co K-edge FT EXAFS spectra in the R space of CoPc/ppy/GDE of initial state and after 200 hours in CO₂-saturated 0.1 M KHCO₃ with an H-type cell at -0.75 V versus RHE.

Figure R15. Comparison of the Co 2p XPS spectra of CoPc/ppy/GDE of initial state and after 200 hours in CO₂-saturated 0.1 M KHCO₃ with an H-type cell at -0.75 V versus RHE.

Combined all the above experiments and analyses together, we propose that Co⁺ could be the active species in CoPc/ppy/GDE for CO₂RR. Co⁺ species are possibly stabilized by the microenvironment and ligands, forming a dynamically stable active centre; upon the removal of applied potential, oxygen-containing species (OH⁻, H₂O and reaction intermediates such as HCO₃⁻ and COOH*) may push the oxidation state of Co to a higher value. However, we agree with you that zero-valence Co may appear during CO₂RR, especially at highly negative potentials and high current densities, the presence and significance of which would require further investigation in the future.

We have included these additional experimental data and detailed mechanistic explanations in the revised manuscript. We hope that these additional analyses would provide a clearer understanding on the behaviour of CoPc/ppy/GDE during CO₂RR. Thank you once again for your valuable feedback.

References

- [1] Wang, S. F. et al. Manipulating C-C coupling pathway in electrochemical CO₂ reduction for selective ethylene and ethanol production over single-atom alloy catalyst. *Nat. Commun.* **15**, 10247 (2024).
- [2] Corbin, N. et al. Heterogeneous molecular catalysts for electrocatalytic CO₂ reduction. *Nano Res.* **12**, 2093–2125 (2019).
- [3] Hu, X. M. et al. Enhanced catalytic activity of cobalt porphyrin in CO₂ electroreduction upon immobilization on carbon materials. *Angew. Chem. Int. Ed.* **56**, 6468–6472 (2017).
- [4] Ren, S. et al. Catalyst aggregation matters for immobilized molecular CO₂RR electrocatalysts. *J. Am. Chem. Soc.* **145**, 4414–4420 (2023).
- [5] Jiang, S. et al. Investigation of cobalt phthalocyanine at the solid/liquid interface by electrochemical tip-enhanced raman spectroscopy. *J. Phys. Chem. C* **123**, 9852–9859 (2019).
- [6] Chen, X. et al. Operando observation of molecular-scale manipulation using electrochemical tip-enhanced raman spectroscopy. *J. Phys. Chem. C* **123**, 24329–24333 (2018).
- [7] Zhu, M. et al. Elucidating the reactivity and mechanism of CO₂ electroreduction at highly dispersed cobalt phthalocyanine. *ACS Energy Lett.* **3**, 1381–1386 (2018).
- [8] Gu, J. et al., Atomically dispersed Fe³⁺ sites catalyze efficient CO₂ electroreduction to CO. *Science* **364**, 1091–1094 (2019).
- [9] Chen, R. R. et al. Operando Mössbauer spectroscopic tracking the metastable state of atomically dispersed tin in copper oxide for selective CO₂ electroreduction. *J. Am. Chem. Soc.* **145**, 20683–20691 (2023).

“Previous studies have claimed that Co⁺ species in CoPc formed by one-electron reduction of Co²⁺, are the active site for CO₂RR^{31,32}. Through the diagnostic Raman signatures of Co⁺ (at ca. 1110 cm⁻¹) and Co²⁺ species (at ca. 1140 cm⁻¹), the identification and quantification of Co⁺ active species have been achieved^{33–35}. Following these reports, the evolution of CoPc/ppy/GDE and the distribution of Co⁺ and Co²⁺ species (i.e. the ratio of Co⁺ to Co²⁺) with applied potentials were monitored through in situ surface-enhanced Raman spectroscopy (SERS). As shown in Figs. 3a and b, characteristic peaks could be resolved at 1113 and 1146 cm⁻¹ at -0.4 V versus RHE for both electrodes of CoPc/ppy/GDE and CoPc/CNT-GDE, which are reported to be associated with the Co⁺ and Co²⁺ species, respectively³⁵. The signal intensity of Co⁺ species (at 1113 cm⁻¹) over CoPc/ppy/GDE becomes prominent at ca. -0.7 V versus RHE, consistent with the redox behaviour of Co²⁺/Co⁺ in CoPc; beyond this potential, the spectrum reaches a relatively steady state. In comparison, the formed Co⁺ species over CoPc/CNT-GDE is significantly lower in amount (Supplementary Figs. 25 and 26). Using the Raman peak area, the ratios of Co⁺ to Co²⁺ in CoPc/ppy/GDE and CoPc/CNT-GDE at various applied potentials were calculated. The results in Fig. 3c show that the formed Co⁺ species over CoPc/ppy/GDE are always higher than that of CoPc/CNT-GDE, aligned with its better catalytic activity for CO generation³⁶. It is inferred that the electron donation effects from ppy to CoPc is responsible. Furthermore, the stability of Co⁺ species

during CO₂RR was evaluated by in situ SERS. As shown in Supplementary Fig. 27, with the continuous application of a negative potential of -1.4 V versus RHE, no obvious degradation in the peak intensity of Co⁺ species could be detected on CoPc/ppy/GDE over a reaction period of 30 min; while for CoPc/CNT-GDE, a gradual decrease of Co⁺ species appears, possibly due to the progressive loss of CoPc from CoPc/CNT-GDE electrode.”

“Having determined that the electron-donating ppy promotes active Co⁺ formation, we employed in situ attenuated total reflection surface enhanced infrared absorption spectroscopy (in situ ATR-SEIRAS) to track the reaction intermediates and gain a molecular-level understanding of CO₂-to-CO conversion over CoPc/ppy/GDE. For the CoPc/ppy/GDE electrode (Fig. 3d), upon the application of negative potentials, surface-bonded CO (CO*) appears and grows in intensity with increasing negative potential, indicating the gradual transformation of CO₂ to CO³⁷. Under the same condition, negligible CO* signals were observed on ppy/GDE and CoPc/GDE, while CoPc/CNT-GDE prepared according to the previous report showed a delayed onset potential of CO* formation along with a weaker signal intensity (Supplementary Figs. 28–30). Meanwhile, an inverse absorption band appears at 1230 cm⁻¹ and becomes more prominent as the applied bias goes more reductive; this phenomenon can be ascribed to the rapid consumption of HCO₃⁻ at large overpotentials. It is noted that under identical bias, CoPc/ppy/GDE exhibits the largest absorption bands for CO* and HCO₃⁻ among tested samples. These observations suggest the improved activity of CoPc/ppy/GDE for CO₂RR than the control samples of ppy/GDE, CoPc/GDE and CoPc/CNT-GDE.”

“To further assess the evolutionary nature of the atomically dispersed Co active sites, ex situ XAS measurements were carried out. Co K-edge XANES spectra of CoPc/ppy/GDE were collected after reacting in a CO₂-saturated 0.1 M KHCO₃ solution for different times. Under this condition, oxygen-containing species (OH⁻, H₂O and reaction intermediates such as HCO₃⁻, CO* and COOH*) may adsorb on CoPc/ppy/GDE, and upon removal of the applied potential valence-state increase of the Co centre would occur (Supplementary Fig. 31)^{38,39}. More evidences come from the EXAFS analysis, which reveals a slight elongation of the Co–N bond (from 1.98 to 2.05 Å)^{40,41} and importantly the emergence of a Co–O coordination peak at ~ 1.2 Å (Fig. 3e)⁴². These phenomena are consistent with the coordination of oxygen-containing reactants/intermediates (e.g., OH⁻, HCO₃⁻ or COOH* species) to the Co centre and the correspondingly relaxed structure (elongated Co–N bond) in CoPc/ppy/GDE. In comparison, the blue-shift of absorption edge in the XANES curves and the Co–O coordination in the EXAFS spectra for CoPc/CNT-GDE are not as prominent as those of CoPc/ppy/GDE (Supplementary Fig. 32), indicating weaker adsorption of OH⁻, CO₂, H₂O reactants and less accumulation of oxygen-containing intermediates over CoPc/CNT-GDE than CoPc/ppy/GDE.”

Comment 3: Although the study reports promising catalytic stability at high current densities, further structural analysis of the CoPc/ppy@GDE catalyst before and after extended catalytic testing

is essential to validate its durability.

Reply to comment 3: We appreciate your suggestion regarding the importance of post reaction analysis of the CoPc/ppy/GDE catalyst to validate its durability. To address this concern, we have conducted additional experiments to examine the structural integrity of the CoPc/ppy/GDE catalyst before and after extended catalytic testing.

The CoPc/ppy/GDE electrode after testing at -0.78 V vs. RHE for 200 h was subjected to XAS, XPS and HAADF-STEM analyses. As shown in Figure R16, no obvious changes in the valence state and coordination environment of Co could be resolved for post-reacted CoPc/ppy/GDE. XPS analysis reveals that although the intensities of Co 2p peaks in post-reacted CoPc/ppy/GDE have decreased (possibly due to the contamination from electrolyte), their positions remain largely unaltered (Figure R17). In addition, HAADF-STEM measurements suggest that the morphology of post-reacted CoPc/ppy/GDE catalyst is retained, with well-preserved atomically dispersed Co sites (Figure R18). All these post-reaction analyses together demonstrate that our CoPc/ppy/GDE electrode is highly resistant to structural collapse/changes under prolonged electrolysis, the negligible demetallation, reduction and aggregation of Co active sites in CoPc/ppy/GDE guarantee its promising stability under highly reductive and high current density conditions.

Figure R16. **a**, XAS spectra at Co K-edge of CoPc/ppy/GDE and **b**, Co K-edge FT EXAFS spectra in the R space of CoPc/ppy/GDE of initial state and after 200 hours in CO_2 -saturated 0.1 M KHCO_3 with an H-type cell at -0.75 V versus RHE.

Figure R17. Comparison of the Co 2p XPS spectra of CoPc/ppy/GDE of initial state and after 200 hours in CO₂-saturated 0.1 M KHCO₃ with an H-type cell at -0.75 V versus RHE.

Figure R18. **a**, TEM image of post-reacted CoPc/ppy/GDE after 200 hours. **b**, HAADF-STEM image of post-reacted CoPc/ppy/GDE after 200 hours.

In summary, our structural analyses on post-reacted CoPc/ppy/GDE provide strong evidences for its high durability under prolonged CO₂RR conditions. We have incorporated these findings into the revised manuscript to further validate the performance and stability of this catalyst.

“Subsequently, the post-reacted CoPc/ppy/GDE was subjected to HAADF-STEM, XAS and XPS analyses. HAADF-STEM measurements suggest that the aggregated sphere-like morphology of CoPc/ppy/GDE catalyst is retained, with well-preserved atomically dispersed Co sites (Supplementary Fig. 19). Meanwhile, no obvious changes in the valence state and coordination environment of Co could be resolved for post-reacted CoPc/ppy/GDE (Supplementary Fig. 20). In addition, XPS analysis reveals that although the intensities of Co 2p peaks in post-reacted CoPc/ppy/GDE have decreased (possibly due to the contamination from electrolytes), their positions remain largely unaltered (Supplementary Fig. 21). All these post-reaction analyses together demonstrate that our CoPc/ppy/GDE electrode with the covalent grafting feature is highly resistant to structural collapse/changes under prolonged electrolysis, the negligible demetallation,

reduction and aggregation of Co active sites in CoPc/ppy/GDE guarantee its promising stability under highly reductive and high current density conditions.”

Comment 4: The electropolymerization approach for anchoring ppy on carbon supports is well-established and widely applied in CO₂ reduction reactions. The manuscript does not sufficiently explain what differentiates CoPc/ppy@GDE from similar electropolymerized systems reported in the literature. The authors are encouraged to provide a clear comparative analysis to highlight any unique structural characteristics or performance advantages of CoPc/ppy@GDE.

Reply to comment 4: We thank the reviewer for raising this critical point, an explanation and a comparative analysis are indeed needed to show the uniqueness and advantages of the CoPc/ppy/GDE electrode for CO₂RR.

Indeed, electropolymerization of ppy on carbon supports has been widely explored, while the yielded ppy/carbon catalysts are generally less efficient for CO₂RR due to the lack of metal-based active centres. Incorporating molecular catalysts with atomically defined active sites, tunable electronic structures, into carbon-based gas diffusion electrodes (GDEs) is a popular strategy to circumvent their limitations of aggregation and poor conductivity, through which significant progresses in CO₂RR activity and product selectivity have been achieved.

In previous studies, as mentioned in Comment 8 from Reviewer 2, similar CoPc-ppy incorporated carbon electrodes have been prepared and applied for CO₂RR. However, significant differences exist between these electrodes and our CoPc/ppy@GDE. Specifically, in the paper of “Cobalt phthalocyanine cross-linked polypyrrole for efficient electroreduction of low concentration CO₂ to CO” (*Chemsuschem* 2022, **15**, e202201455), CoPc modified with four sulphyloxy groups (defined as CoPcS₄) was employed as both the dopant and gelator to crosslink ppy; the chemical bonds formed between CoPcS₄ and ppy are ionic bonds instead of C–C covalent bonds in our CoPc/ppy/GDE electrode. (Figure R19). Most importantly, the yielded CoPcS₄-ppy-CC (carbon cloth) electrode exhibits much lower CO₂RR activity and stability than our CoPc/ppy/GDE under similar conditions (ca. –6.5 mA cm⁻² vs. ca. –60 mA cm⁻² at –1 V_{RHE} in an H-cell; –2.1 mA cm⁻² for 3500 s vs. –20 mA cm⁻² for 200 h). While in the paper of “Electrocatalytic reduction of carbon dioxide by cobalt-phthalocyanine-incorporated polypyrrole” (*Electrochem. Solid St.* 2009, **12**, E17), CoPc-ppy modified glassy carbon is obtained by electrochemical polymerization of ppy at glassy carbon followed by drop-casting of CoPc solution.

Figure R19. Synthesis of CoPcS₄-PPy.

Such a catalyst loading manner of drop-casting has been widely applied for monomers of CoPc or CoPc derivatives (Liang yongye, Wang hailiang), CoPc-derived/containing covalent organic frameworks or metal organic frameworks (Peng tianyou, *Adv. Mater.* 2022, 2203139; Zhang liming, *J. Am. Chem. Soc.* 2022, 144, 21502), with carbon-based gas diffusion electrodes; while ultrasonic spray could offer a more uniform catalyst dispersion. For these CoPc-based powder catalysts, binders like Nafion are usually required, scaffolds such as CNT (carbon nanotube) and graphene are also included to enhance their dispersion and adhesion strength through π - π stacking interaction (*Adv. Mater.* 2023, 35, 2303179).

Considering the possible loss of CoPc centres during operation (especially at high current densities), anchoring these molecular catalysts directly onto carbon-based GDEs through strong covalent bonds is a desirable strategy. Presently, direct covalent grafting of CoPc (and other metal phthalocyanine/porphyrin) based molecular catalysts onto carbon-based GDEs would require the use of CoPc with specially designed functional groups (such as amino group) or polymerizable groups (such as carbazole). Among common covalent bonds that could form between CoPc (or its derivatives) and carbon-based GDEs, C–C bond is more inert and robust for extreme reaction conditions, when comparing to those hydrolysable amide and ester bonds.

In our study, to offer robust C–C bonds between carbon-based GDEs and pristine CoPc, we introduce ppy as the linker which initiates the C–C coupling process between GDEs and ppy, ppy and pristine CoPc thereby connecting them together via C–C bonds. During this covalent grafting process, pristine CoPc without polymerizable groups is applied; meanwhile, no pretreatment of carbon-based GDEs is required, which minimizes the amount of oxygen-containing functional groups (such as carboxyl and hydroxyl groups) thereby guaranteeing sufficient hydrophobicity for efficient CO₂ gas penetration. As far as we know, such a convenient and efficient approach has not been explored for direct grafting of pristine CoPc (rather than CoPc derivatives) on carbon-based GDEs, the employed ppy not only acts as the highly conductive and robust linker between CoPc and GDEs, but also shows electron donation effects to promote the CO₂RR activity of CoPc centers (negative shift of Co 2p peaks by ~0.6 eV Figure R20). Importantly, this covalent grafting strategy allows facile control over catalyst loadings (e.g., through adjusting the polymerization parameters

of CV number, potential range), and could be well extended to other pristine metal phthalocyanines or porphyrins (Figure R21).

Figure R20. Comparison of the Co 2p XPS spectra of CoPc/ppy/GDE and commercial CoPc.

Figure R21. a, LSV curves of CoPc/ppy/GDE, FePc/ppy/GDE, CuPc/ppy/GDE and NiPc/ppy/GDE during CO₂RR in CO₂-saturated 0.1 M KHCO₃ with an H-type cell. **b,** Faradaic efficiency and current density of CoPc/ppy/GDE, FePc/ppy/GDE, CuPc/ppy/GDE and NiPc/ppy/GDE during CO₂RR in CO₂-saturated 0.1 M KHCO₃ with an H-type cell at -0.7 V versus RHE.

Based on the combined advantages of robust C–C bonds, binder-free interface, highly conductive ppy linker, appropriate catalyst dispersion, and active Co single-atom sites, our in situ covalent grafting derived CoPc/ppy/GDE establishes promising activity and stability for CO₂RR. Specifically, CoPc/ppy/GDE electrode exhibits a high current density of -40 mA cm⁻² at -0.78 V versus RHE, better than previously reported CoPc-based catalysts (Table R2 and R3); an exceptional 200 hours stability at -0.78 V versus RHE is demonstrated on CoPc/ppy/GDE, and at the end of stability test no obvious valence state, structure and coordination environment changes of Co are detected by XPS, HAADF-STEM and XAS (Figure R22–R24). Furthermore, the as-assembled CoFe-Ci@GQDs/NF||CoPc/ppy/GDE electrolyzer achieves a record device stability of 50 hours at

1000 mA cm⁻² and 120 hours at 500 mA cm⁻², which strongly demonstrate that our CoPc/ppy/GDE electrode is highly resistant to structural collapse/changes under industrially relevant current densities.

Table R2. Comparison of the CO₂-to-CO conversion of different electrocatalysts in H-type cells.

Catalysts	Electrolyte	FE _{CO} (%)	Potential (V vs. RHE)	J (mA cm ⁻²)	Stability (h)	Ref.
CoPc/ppy/GDE	0.1 M KHCO ₃	96	-0.78	-40	200	This work
EP-CoP	0.5 M KHCO ₃	95	-0.62	-8.5	42	17
CoTPP/CNT	0.5 M KHCO ₃	91	-0.62	-3.2	4	S8
FePGF/CFP	0.1 M KHCO ₃	98.7	-0.54	-1.68	10	S9
Co-PMOF	0.5 M KHCO ₃	98.7	-0.8	-18.3	36	S10
STPyP-Co	0.5 M KHCO ₃	96	-0.62	-6.6	48	S11
Fe-PB	0.5 M KHCO ₃	100	-0.63	~-0.6	24	S12
Co-TTCOF	0.5 M KHCO ₃	91.3	-0.7	-2.02	40	S13
COF-366-(OMe) ₂ - Co@CNT	0.5 M KHCO ₃	93.6	-0.68	~-6	12	S14
COF-367-Co	0.5 M KHCO ₃	91	-0.55	-3.3	24	S15
CoPc-TFPN COF	0.5 M KHCO ₃	99.8	-0.9	-14.1	60	S16
TT-Por(Co)-COF	0.5 M KHCO ₃	91.4	-0.6	~-1.5	10	S17
CoPc-PI-COF-1	0.5 M KHCO ₃	90	-0.7	-10	40	S18
MWCNT-Por- COF-Co	0.5 M KHCO ₃	88	-0.7	-6	50	S19
CoPc/CNT	0.1 M KHCO ₃	92	-0.63	-10	10	S20

CoTMAPc@CNT	0.5 M KHCO ₃	95	-0.62	~-12	12	S21
CoPPc/CNT	0.5 M NaHCO ₃	90	-0.54	-12	24	S22
CoPc-2H2Por	0.5 M KHCO ₃	95	-0.6	-5	70	S23

Table R3. Comparison of the CO₂-to-CO conversion of different electrocatalysts in MEA.

Catalysts	Electrolyte	FE _{CO}	Cell voltage (V)	J (mA cm ⁻²)	Stability (h)	References
CoPc/ppy/GDE	1.0 M KOH	99%	~2.8	500	120	This work
CoPc/ppy/GDE	1.0 M KOH	98%	~3.2	1000	50	This work
Co-CNTs MW	1.0 M KOH	98.3%	~2	100	12	44
CoPc	1.0 M KOH	88%	~2.52	200	6	45
MWNT/ PyPBI/Au	2.0 M KOH	85%	~2.25	158	8	46
Ag/C	1.0 M KOH	83%	~2.75	120	0.417	47
Ag nanoparticles	1.0 M KOH+0.33 M Urea	98%	~2.16	100	10	48
Zn ₂ P ₂ O ₇	1.0 M KOH	93.9%		100	7	49
Au ₂₄	0.1 M KOH	90.0%	~3	100	100	50
TC- CoPc/MWCNTs	0.5 M KHCO ₃	97%	~2.6	50	60	17
CoPc/Mg(OH) ₂	0.1 M KHCO ₃	95%	3.4	100	20	51
β-CoPc/CP	1.0 M KOH	92%	2	100	29.3	52
AG- CoPc/MWCNTs	0.1 M KHCO ₃	80%	3	50	12	26
MD- CoPc/MWCNTs	0.1 M KHCO ₃	80%	3.2	50	40	26

Figure R22. **a**, XAS spectra at Co K-edge of CoPc/ppy/GDE and **b**, Co K-edge FT EXAFS spectra in the R space of CoPc/ppy/GDE of initial state and after 200 hours in CO₂-saturated 0.1 M KHCO₃ with an H-type cell at -0.75 V versus RHE.

Figure R23. Comparison of the Co 2p XPS spectra of CoPc/ppy/GDE of initial state and after 200 hours in CO₂-saturated 0.1 M KHCO₃ with an H-type cell at -0.75 V versus RHE.

Figure R24. **a**, TEM image of post-reacted CoPc/ppy/GDE after 200 hours. **b**, HAADF-STEM image of post-reacted CoPc/ppy/GDE after 200 hours.

In summary, our in situ electropolymerization yields CoPc/ppy/GDE electrode with robust C–C covalent bonds between CoPc and ppy, ppy and GDEs. Performance evaluations including activity and stability tests, as well as the structural and spectroscopic analyses demonstrate the unique advantages of CoPc/ppy/GDE for CO₂RR, which presents a significant improvement in the design of molecular catalysts for sustainable CO₂-to-fuel technologies. To highlight the merits of our electropolymerization-derived CoPc/ppy/GDE electrode, the introduction section has been revised to offer a more explicit comparison with conventional synthesis methods.

Reference

- [1] Chen, J. M. et al. Cobalt phthalocyanine cross-linked polypyrrole for efficient electroreduction of low concentration CO₂ to CO. *Chemsuschem* **15**, e202201455 (2022).
- [2] Zhang, A. J. et al. Electrocatalytic reduction of carbon dioxide by cobalt-phthalocyanine-incorporated polypyrrole. *Electrochem. Solid St.* **12**, E17–E19 (2009).

“Electrocatalytic conversion of CO₂ into value-added chemicals can mitigate the climate crisis induced by greenhouse gases while simultaneously enabling renewable energy storage^{1–3}. Among various catalytic systems, molecular catalysts, particularly metal phthalocyanines and metal porphyrins, have emerged as promising candidates due to their atomically defined active sites and tunable electronic structures^{4–6}. However, under operating conditions, especially those with industrially relevant current densities (>200 mA cm⁻²), molecular catalysts often suffer from leaching, agglomeration, or even structural degradation, which hinder their practical applications^{7–12}. Recently, metal phthalocyanine-based molecular catalysts have demonstrated highly selective CO₂ electrolysis at high current densities in flow reactors^{13,14}. To this end, these molecular catalysts are generally coated onto the carbon-based GDEs to circumvent their limitations of aggregation and poor conductivity, with the majority of the previous works utilizing the noncovalent anchoring strategies. In these strategies, binders like Nafion are usually required, scaffolds such as carbon nanotube (CNT) and graphene are also included to enhance the dispersion and adhesion strength of molecular catalysts through π - π stacking interaction^{15–17}.

Considering the possible loss of molecular catalysts during operation (especially at high current densities), anchoring them directly onto carbon-based GDEs through strong covalent bonds is a desirable strategy^{18–20}. Presently, direct covalent grafting of metal phthalocyanine/porphyrin based molecular catalysts onto carbon-based GDEs would require the use of these materials with specially designed functional groups (such as amino group) or polymerizable groups (such as carbazole group). Until now, due to the inherent chemical inertness of pristine metal phthalocyanines/porphyrins, direct electrochemical polymerization or covalent grafting of them onto the carbon-based GDEs is challenging. On the other hand, among common covalent bonds that could form between metal phthalocyanines/porphyrins (or their derivatives) and carbon-based GDEs, C–C bond is more inert and robust for extreme reaction conditions, when comparing to those hydrolysable amide and ester bonds¹⁹.

Herein, using pristine CoPc as the example, we report an in situ electro-polymerization

strategy for the direct covalent grafting of pristine metal phthalocyanines onto GDEs. This method occurs at room temperature and leverages the conductive polymer ppy as a molecular “glue”, forming robust C–C bonds with pristine CoPc; meanwhile, besides as the scaffold, ppy-based matrix also shows electron donation effects to promote the activity of active Co⁺ sites. Finally, to bridge the gap between material innovation and industrial application, we further engineered a membrane electrode assembly (MEA) electrolyzer incorporating the CoPc/ppy/GDE cathode. This device achieves 50 hours of stable CO production at ampere-level current densities (1 A cm⁻²) in alkaline media. The proposed strategy here effectively addresses the challenge of direct covalent grafting of pristine metal phthalocyanines onto GDEs for efficient and durable CO₂ reduction reaction (CO₂RR) under industrially relevant current densities (>200 mA cm⁻²).”

Reviewer #2:

Comment: The authors address the immobilization limitations of phthalocyanine catalysts on electrode towards industrial-scale CO₂ reduction, and graft CoPc onto gas diffusion electrode through polypyrrole, which was applied in membrane electrode assembly electrolyser for CO₂ to CO. The present work was well organized with large characterization including XAS, in situ technology, and DFT calculation. However, there are still problems need further revision.

Comment 1: How to know the molecular weight? Whether it influence the performance of CoPc/ppy@GDE catalyst? And how to ascertain the best polypyrrole?

Reply to comment 1: We appreciate the reviewer's insightful question regarding the molecular weight control in the CoPc/ppy/GDE catalyst. Due to the inherent chemical inertness of pristine CoPc (without specially designed functional groups or polymerizable groups), direct electrochemical polymerization or grafting of it onto the carbon-based GDEs is challenging. To address this issue, we propose a ppy-mediated grafting strategy to offer robust C–C bonds between GDEs and pristine CoPc, in which ppy serves as both the conductive linker and molecular scaffold.

In this approach, precise control over the molecular weight or chain length of ppy is difficult because the electropolymerization process is dynamically influenced by various steps including monomer diffusion, radical coupling, and CoPc incorporation. We have tried to use F-doped tin oxide (FTO) as the conductive substrate to study the grow process and molecular weight of CoPc/ppy, however, the weak bonding between FTO and CoPc/ppy has led to easy detachment of product. We have also tried to monitor the deposition process of CoPc/ppy through the microcrystal quartz balance, but no available signals could be collected.

Despite the lack of precise control over the ppy molecular weight, our CoPc/ppy/GDE catalysts show controllable and reproducible electrochemical behaviors towards CO₂RR, when the electropolymerization conditions are well organized. Specifically, the electrochemical CO₂RR performance of CoPc/ppy/GDE electrodes can be systematically and finely optimized through tuning two key parameters including the number of CV scans and the feed ratio of pyrrole to CoPc, based on which the chain length of ppy and CoPc loading/incorporating could be controlled. Figure R25 illustrates the effects of CV number during electropolymerization, the current density that corresponds to the electrochemical growth of CoPc and ppy on GDE initially increases with cycle number but exhibits a saturation trend beyond 40 cycles. We infer that mass transport or ppy resistance issues gradually dominate and cease the continuous growth of CoPc and ppy on GDE in the applied CV potential range. Correspondingly, the CO₂RR performance of yielded CoPc/ppy/GDE electrodes begins to stabilize from 40 CV cycles (Figure R26). Beyond 40 CV cycles, the gain in current density becomes limited, along with the consideration of synthesis time, the CV number for CoPc/ppy/GDE fabrication is set at 40.

Next, considering that the CO₂RR performance of CoPc/ppy/GDE electrodes is closely correlated with the loading/incorporating amount of CoPc, we optimized the synthetic procedure by varying the pyrrole-to-CoPc feed ratio (1:6 to 6:1). As shown in Figure R27, only small activity differences could be observed for these CoPc/ppy/GDE electrodes obtained in the presence of both

pyrrole and CoPc, and the best performing one is obtained with a high pyrrole-to-CoPc ratio of 4:1; when only CoPc is applied, the yielded CoPc/GDE electrode shows largely suppressed CO₂RR activity. Based on these results, we infer that the electropolymerization step of ppy dominates the whole formation process of CoPc/ppy/GDE electrodes, as pristine CoPc is chemically less active towards electropolymerization; under this condition, ppy initiates the C–C coupling process between GDEs and ppy, ppy and pristine CoPc, thereby realizing covalent grafting of pristine CoPc onto carbon-based GDEs via robust C–C bonds.

Collectively, through utilizing ppy as the electroactive "glue", covalent grafting of pristine CoPc onto carbon-based GDEs via robust C–C bonds has been achieved. The optimal conditions for CoPc/ppy/GDE electrode fabrication are determined to be 40 of CV cycles and 4:1 of pyrrole-to-CoPc ratio that balance performance and cost efficiency. Although the precise control over the ppy molecular weight remains a challenge and is worth of future studies, the yielded CoPc/ppy/GDE electrodes show reproducible CO₂RR activity and promising stability under high current densities, suggesting the effectiveness of our in situ electropolymerization strategy for covalent grafting of pristine molecular catalysts onto carbon-based GDEs. We have added these optimizations and discussion in the revised manuscript. Thank the reviewer again!

Figure R25. **a**, The different deposition cycles of CoPc/ppy/GDE during the electro-polymerization in 0.01 M [BMIM]BF₄ with 4 mM pyrrole and 1 mM CoPc at 100 mV s⁻¹. The potentials were recorded with the Ag/AgCl scale. **b**, The variation of current density with the number of deposition cycles.

Figure R26. **a**, The LSV of CoPc/ppy/GDE during CO₂RR in CO₂-saturated 0.1 M KHCO₃ with an H-type cell.

Figure R27. a, The CV patterns of the growth of CoPc/ppy on GDE in 0.1 M [BMIM]BF₄ with different ratios of pyrole and CoPc. **b**, The Faradaic efficiency and current density of CoPc/ppy/GDE synthesized by different precursors of pyrole and CoPc at -0.8 V versus RHE during CO₂RR in CO₂-saturated 0.1 M KHCO₃ with an H-type cell.

Comment 2: Line 136-139. Since the electron density of Co is enhanced, the binding energy should blueshift, while not redshift. This part needs reconsideration. The result is not consistent with XAS analysis.

Reply to comment 2: We sincerely appreciate the reviewer's insightful question regarding the shift of Co binding energy in our XPS data. Upon re-examining the data, we found that the charge correction step had been ignored, which led to an erroneous interpretation of the Co 2p peak shifts. We have recalibrated the XPS spectra, and the revised data in Figure R28 now show reduced binding energies of Co 2p peaks (780.8 and 795.9 eV) in CoPc/ppy/GDE than those of commercial CoPc (781.4 and 796.5 eV), consistent with the XAS data (absorption edge of CoPc/ppy/GDE shifts towards the lower energy direction compared to commercial CoPc in Figure R29) and the enhanced electron density at Co sites in CoPc/ppy/GDE due to the electron donation effects of ppy. We appreciate your attention to this mistake, thanks again!

Figure R28. Comparison of the Co 2p XPS spectra of CoPc/ppy/GDE and commercial CoPc.

Figure R29. **a**, XAS spectra at Co K-edge of CoPc/ppy/GDE and the reference Co foil and CoPc samples. **b**, Co K-edge FT EXAFS spectra in the R space of CoPc/ppy/GDE and reference Co foil and CoPc samples.

“The XPS measurements further corroborate the modulated electronic structure of CoPc/ppy/GDE, which shows lower binding energies for Co 2p peaks than those of commercial CoPc (780.8 vs. 781.4 eV for Co 2p_{3/2}; 795.9 vs. 796.5 eV for Co 2p_{1/2})²⁸. These phenomena, as mentioned above, are attributed to the electron donation effects from ppy to the CoPc domain (Supplementary Fig. 12).”

Comment 3: The fig.2a, it should be labelled for different color for better understanding. The Fig. 2f.g caption was not correct.

Reply to comment 3: Thank you for your suggestions regarding the clarity and accuracy of the figure and figure caption in our manuscript. We have revised these figure and figure caption to clearly and correctly show the information we want to express (Figure R30).

Figure R30. Structural analyses and intrinsic activity of CoPc/ppy/GDE. **a**, Schematic illustration of CoPc/ppy/GDE. **b**, TEM image of CoPc/ppy/GDE. **c**, HAADF-STEM image of CoPc/ppy/GDE. The atomically dispersed Co sites are marked by red circles. **d**, XAS spectra at Co K-edge of CoPc/ppy/GDE and the reference Co foil and CoPc samples. **e**, Co K-edge FT EXAFS spectra in the R space of CoPc/ppy/GDE and reference Co foil and CoPc samples. **f**, **g**, WT-EXAFS plots of CoPc samples and CoPc/ppy/GDE. **h**, LSV curves of CoPc/ppy/GDE, CoPc/CNT-GDE, CoPc/GDE and ppy/GDE during CO₂RR in an H-cell with CO₂-saturated 0.1 M KHCO₃ electrolyte. **i**, CV curve of CoPc/ppy/GDE during CO₂RR in an H-cell with CO₂-saturated 0.1 M KHCO₃ electrolyte.

Comment 4: Line 153, the pH needs further tested, it cannot be 8.3.

Reply to comment 4: Thank you for your comment regarding the pH value of the 0.1 M KHCO₃ solution. To ensure the accuracy of the pH values reported in our manuscript, we have re-conducted pH measurements using a calibrated pH meter, following the standard procedures. Our experimental results show that the pH of the 0.1 M KHCO₃ solution is around 8.3 and 0.1 M CO₂ saturated KHCO₃ solution is around 6.8. We have corrected the error and thank you once again for your valuable comments.

Comment 5: How TOF was calculated according to the all Co sites? i.e. how to ascertain the Co sites?

Reply to comment 5: We appreciate the reviewer's question regarding the TOF calculation based

on all cobalt sites.

TOF estimation methods face limitations when dealing with single-atom systems. In our CoPc/ppy/GDE electrode, due to the influence of carbon-based porous GDE, and the lack of capacitance value of atomically dispersed Co-N₄ site, obtaining the amount of Co sites through electrochemical double-layer capacitance is challenging. In-situ spectroscopic quantification (e.g., operando XAS) of active sites, while theoretically ideal, suffers from insufficient sensitivity. Precise determination of Co amount in our CoPc/ppy/GDE electrode through spectroscopic methods is further complicated by the ultra-low loading of CoPc in the single atom configuration.

Therefore, in our study we have adopted the ICP-OES to quantify the Co concentration in CoPc/ppy/GDE. The obtained Co amount in CoPc/ppy/GDE is comparable to those of recently reported M-N-C catalysts^{1,2}.

The TOF value was then determined using the following formula:

$$TOF = \frac{j \cdot A}{n \cdot F \cdot N_{Co}}$$

where j was the current density ($A\text{ cm}^{-2}$), A was the geometric electrode area (1 cm^2), n represented the electron transfer number for CO₂-to-CO conversion ($n=2$), F was Faraday's constant (96485 C mol^{-1}), and N_{Co} was the total number of Co sites quantified by ICP-OES. Actual Co loading of the optimized CoPc/ppy/GDE catalyst (obtained with 40 CV cycles) was measured to be 0.0837 wt% with ICP-OES. For our CoPc/ppy/GDE catalyst, we assumed all Co sites participate in the CO₂RR reaction^{1,2}. Therefore, the TOF value at 40 mA cm^{-2} was calculated as follows:

$$TOF = \frac{j \left[\frac{\text{mA}}{\text{cm}^2} \right] \times 10^{-3} \times A \left[\text{cm}^2 \right]}{2 \times 96485 \times N_{Co}}$$

$$TOF = \frac{40 \times \left[\frac{\text{mA}}{\text{cm}^2} \right] \times 10^{-3} \times 1\text{ cm}^2}{2 \times 96485 \frac{\text{C}}{\text{mol}} \times \frac{0.0837\% \times 0.01\text{ g}}{59 \left[\frac{\text{g}}{\text{mol}} \right]}} = 1.46\text{ s}^{-1}$$

“The TOF value was determined using the formula:

$$TOF = \frac{j \cdot A}{n \cdot F \cdot N_{Co}}$$

where j was the current density ($A\text{ cm}^{-2}$), A was the geometric electrode area (1 cm^2), n represented the electron transfer number for CO₂-to-CO conversion ($n=2$), F was Faraday's constant (96485 C mol^{-1}), and N_{Co} was the total number of Co sites quantified by ICP-OES. Actual Co loading of

the optimized CoPc/ppy/GDE catalyst (obtained with 40 CV cycles) was measured to be 0.0837 wt% with ICP-MS. For our CoPc/ppy/GDE catalyst, we assumed all Co sites participate in the CO₂RR reaction^{1,2}. Therefore, the TOF value at 40 mA cm⁻² was calculated as follows:

$$TOF = \frac{j \left[\frac{\text{mA}}{\text{cm}^2} \right] \times 10^{-3} \times A \left[\text{cm}^2 \right]}{2 \times 96485 \times N_{\text{Co}}}$$

$$TOF = \frac{40 \times \left[\frac{\text{mA}}{\text{cm}^2} \right] \times 10^{-3} \times 1 \text{ cm}^2}{2 \times 96485 \frac{\text{C}}{\text{mol}} \times \frac{0.0837\% \times 0.01 \text{ g}}{59 \left[\frac{\text{g}}{\text{mol}} \right]}} = 1.46 \text{ s}^{-1} \text{ ,}$$

References

- [1] Zhou, S. et al. Amphiphilic cobalt phthalocyanine boosts carbon dioxide reduction. *Adv. Mater.* **35**, 2300923 (2023).
- [2] Zhang, H., et al. Mechanistic insights into CO₂ conversion chemistry of copper bis-(terpyridine) molecular electrocatalyst using accessible operando spectrochemistry. *Nat. Commun.* **13**, 6029 (2022).

Comment 6: Line 241, from the calculation, we can only know that ppy alters the catalytic thermodynamics while not kinetic.

Reply to comment 6: Thank you for your insightful comment regarding the promoting effects of ppy on the catalytic performance of CoPc/ppy/GDE electrode. We agree with you that our theoretical calculations can only reach a conclusion that ppy could affect the overall reaction thermodynamics of CoPc/ppy/GDE electrode through altering the interaction strength of CO₂ and COOH intermediate at the Co active sites. We have corrected the corresponding discussion in the revised manuscript. Thank the reviewer again!

“To gain more comprehensive mechanistic insights into ppy-enhanced electrocatalytic activity, free energy diagrams considering thermodynamics of CO₂-to-CO conversion on both CoPc/ppy/GDE and CoPc/GDE were constructed using DFT calculations.”

Comment 7: Fig. 4f in present is not quite convincing.

Reply to comment 7: Thank you for your comment regarding the strength of Fig. 4f in supporting our conclusion. We have revised Fig. 4f as supplementary Fig. 6 (Figure R31), and added corresponding discussion as follows.

Figure R31. Schematic diagram of differential charge density of CoPc/ppy and Bader charge analysis. The cyan (yellow) region shows electron loss (gain).

“Bader charge analysis suggest that ppy has a tendency to transfer electrons to CoPc (Supplementary Fig. 6). The ppy acts as a linker to anchor CoPc onto the GDE through covalent bonds, while simultaneously modulating the electronic structure of CoPc for improved catalytic performance.” Thank the reviewer again!

Comment 8: Since former work has published with CoPc cross-linked ppy(ChemSusChem 2022,15(23):e202201455), CoPc incorporated with ppy (ELECTROCHEMICAL AND SOLID STATE LETTERS 2009,12(8):E17-E19), what’s the novelty of present work?

Reply to comment 8: We thank the reviewer for raising this critical point, an explanation and a comparative analysis are indeed needed to show the uniqueness and advantages of the CoPc/ppy/GDE electrode for CO₂RR.

Indeed, electropolymerization of ppy on carbon supports has been widely explored, while the yielded ppy/carbon catalysts are generally less efficient for CO₂RR due to the lack of metal-based active centres. Incorporating molecular catalysts with atomically defined active sites, tunable electronic structures, into carbon-based gas diffusion electrodes (GDEs) is a popular strategy to circumvent their limitations of aggregation and poor conductivity, through which significant progresses in CO₂RR activity and product selectivity have been achieved.

In previous studies, as mentioned in Comment 8 from Reviewer 2, similar CoPc-ppy incorporated carbon electrodes have been prepared and applied for CO₂RR. However, significant differences exist between these electrodes and our CoPc/ppy@GDE. Specifically, in the paper of “Cobalt phthalocyanine cross-linked polypyrrole for efficient electroreduction of low concentration CO₂ to CO” (ChemSuschem 2022, **15**, e202201455), CoPc modified with four sulphyloxy groups (defined as CoPcS₄) was employed as both the dopant and gelator to crosslink ppy; the chemical bonds formed between CoPcS₄ and ppy are ionic bonds instead of C–C covalent bonds in our CoPc/ppy/GDE electrode. (Figure R32). Most importantly, the yielded CoPcS₄-ppy-CC (carbon cloth) electrode exhibits much lower CO₂RR activity and stability than our CoPc/ppy/GDE under

similar conditions (ca. -6.5 mA cm^{-2} vs. ca. -60 mA cm^{-2} at $-1 \text{ V}_{\text{RHE}}$ in an H-cell; -2.1 mA cm^{-2} for 3500 s vs. -20 mA cm^{-2} for 200 h). While in the paper of “Electrocatalytic reduction of carbon dioxide by cobalt-phthalocyanine-incorporated polypyrrole” (*Electrochem. Solid St.* 2009, 12, E17), CoPc-ppy modified glassy carbon is obtained by electrochemical polymerization of ppy at glassy carbon followed by drop-casting of CoPc solution.

Figure R32. Synthesis of CoPcS₄-PPy.

Such a catalyst loading manner of drop-casting has been widely applied for monomers of CoPc or CoPc derivatives (Liang yongye, Wang hailiang), CoPc-derived/containing covalent organic frameworks or metal organic frameworks (Peng tianyou, *Adv. Mater.* 2022, 2203139; Zhang liming, *J. Am. Chem. Soc.* 2022, 144, 21502), with carbon-based gas diffusion electrodes; while ultrasonic spray could offer a more uniform catalyst dispersion. For these CoPc-based powder catalysts, binders like Nafion are usually required, scaffolds such as CNT (carbon nanotube) and graphene are also included to enhance their dispersion and adhesion strength through π - π stacking interaction (*Adv. Mater.* 2023, 35, 2303179).

Considering the possible loss of CoPc centres during operation (especially at high current densities), anchoring these molecular catalysts directly onto carbon-based GDEs through strong covalent bonds is a desirable strategy. Presently, direct covalent grafting of CoPc (and other metal phthalocyanine/porphyrin) based molecular catalysts onto carbon-based GDEs would require the use of CoPc with specially designed functional groups (such as amino group) or polymerizable groups (such as carbazole). Among common covalent bonds that could form between CoPc (or its derivatives) and carbon-based GDEs, C–C bond is more inert and robust for extreme reaction conditions, when comparing to those hydrolysable amide and ester bonds.

In our study, to offer robust C–C bonds between carbon-based GDEs and pristine CoPc, we introduce ppy as the linker which initiates the C–C coupling process between GDEs and ppy, ppy and pristine CoPc thereby connecting them together via C–C bonds. During this covalent grafting process, pristine CoPc without polymerizable groups is applied; meanwhile, no pretreatment of carbon-based GDEs is required, which minimizes the amount of oxygen-containing functional groups (such as carboxyl and hydroxyl groups) thereby guaranteeing sufficient hydrophobicity for

efficient CO₂ gas penetration. As far as we know, such a convenient and efficient approach has not been explored for direct grafting of pristine CoPc (rather than CoPc derivatives) on carbon-based GDEs, the employed ppy not only acts as the highly conductive and robust linker between CoPc and GDEs, but also shows electron donation effects to promote the CO₂RR activity of CoPc centers (negative shift of Co 2p peaks by ~0.6 eV Figure R33). Importantly, this covalent grafting strategy allows facile control over catalyst loadings (e.g., through adjusting the polymerization parameters of CV number, potential range), and could be well extended to other pristine metal phthalocyanines or porphyrins (Figure R34).

Figure R33. Comparison of the Co 2p XPS spectra of CoPc/ppy/GDE and commercial CoPc.

Figure R34. a, LSV curves of CoPc/ppy/GDE, FePc/ppy/GDE, CuPc/ppy/GDE and NiPc/ppy/GDE during CO₂RR in CO₂-saturated 0.1 M KHCO₃ with an H-type cell. **b,** Faradaic efficiency and current density of CoPc/ppy/GDE, FePc/ppy/GDE, CuPc/ppy/GDE and NiPc/ppy/GDE during CO₂RR in CO₂-saturated 0.1 M KHCO₃ with an H-type cell at -0.7 V versus RHE.

Based on the combined advantages of robust C–C bonds, binder-free interface, highly conductive ppy linker, appropriate catalyst dispersion, and active Co single-atom sites, our in situ covalent grafting derived CoPc/ppy/GDE establishes promising activity and stability for CO₂RR.

Specifically, CoPc/ppy/GDE electrode exhibits a high current density of -40 mA cm^{-2} at -0.78 V versus RHE, better than previously reported CoPc-based catalysts (Table R4 and R5); an exceptional 200 hours stability at -0.78 V versus RHE is demonstrated on CoPc/ppy/GDE, and at the end of stability test no obvious valence state, structure and coordination environment changes of Co are detected by XAS, XPS and HAADF-STEM (Figure R35–R37). Furthermore, the as-assembled CoFe-Ci@GQDs/NF||CoPc/ppy/GDE electrolyzer achieves a record device stability of 50 hours at 1000 mA cm^{-2} and 120 hours at 500 mA cm^{-2} , which strongly demonstrate that our CoPc/ppy/GDE electrode is highly resistant to structural collapse/changes under industrially relevant current densities.

Table R4. Comparison of the CO₂-to-CO conversion of different electrocatalysts in H-type cells.

Catalysts	Electrolyte	FE _{CO} (%)	Potential (V vs. RHE)	J (mA cm ⁻²)	Stability (h)	Ref.
CoPc/ppy/GDE	0.1 M KHCO ₃	96	-0.78	-40	200	This work
EP-CoP	0.5 M KHCO ₃	95	-0.62	-8.5	42	17
CoTPP/CNT	0.5 M KHCO ₃	91	-0.62	-3.2	4	S8
FePGF/CFP	0.1 M KHCO ₃	98.7	-0.54	-1.68	10	S9
Co-PMOF	0.5 M KHCO ₃	98.7	-0.8	-18.3	36	S10
STPyP-Co	0.5 M KHCO ₃	96	-0.62	-6.6	48	S11
Fe-PB	0.5 M KHCO ₃	100	-0.63	~-0.6	24	S12
Co-TTCOF	0.5 M KHCO ₃	91.3	-0.7	-2.02	40	S13
COF-366- (OMe) ₂ - Co@CNT	0.5 M KHCO ₃	93.6	-0.68	~-6	12	S14
COF-367-Co	0.5 M KHCO ₃	91	-0.55	-3.3	24	S15
CoPc-TFPN COF	0.5 M KHCO ₃	99.8	-0.9	-14.1	60	S16
TT-Por(Co)- COF	0.5 M KHCO ₃	91.4	-0.6	~-1.5	10	S17
CoPc-PI-COF-1	0.5 M KHCO ₃	90	-0.7	-10	40	S18

MWCNT-Por-COF-Co	0.5 M KHCO ₃	88	-0.7	-6	50	S19
CoPc/CNT	0.1 M KHCO ₃	92	-0.63	-10	10	S20
CoTMAPc@CNT	0.5 M KHCO ₃	95	-0.62	~-12	12	S21
CoPPc/CNT	0.5 M NaHCO ₃	90	-0.54	-12	24	S22
CoPc-2H2Por	0.5 M KHCO ₃	95	-0.6	-5	70	S23

Table R5. Comparison of the CO₂-to-CO conversion of different electrocatalysts in MEA.

Catalysts	Electrolyte	FE _{CO}	Cell voltage (V)	J (mA cm ⁻²)	Stability (h)	References
CoPc/ppy/GDE	1.0 M KOH	99%	~2.8	500	120	This work
CoPc/ppy/GDE	1.0 M KOH	98%	~3.2	1000	50	This work
Co-CNTs MW	1.0 M KOH	98.3%	~2	100	12	44
CoPc	1.0 M KOH	88%	~2.52	200	6	45
MWNT/PyPBI/Au	2.0 M KOH	85%	~2.25	158	8	46
Ag/C	1.0 M KOH	83%	~2.75	120	0.417	47
Ag nanoparticles	1.0 M KOH+0.33 M Urea	98%	~2.16	100	10	48
Zn ₂ P ₂ O ₇	1.0 M KOH	93.9%		100	7	49
Au ₂₄	0.1 M KOH	90.0%	~3	100	100	50
TC-CoPc/MWCNTs	0.5 M KHCO ₃	97%	~2.6	50	60	17
CoPc/Mg(OH) ₂	0.1 M KHCO ₃	95%	3.4	100	20	51
β-CoPc/CP	1.0 M KOH	92%	2	100	29.3	52

AG- CoPc/MWCNTs	0.1 M KHCO ₃	80%	3	50	12	26
MD- CoPc/MWCNTs	0.1 M KHCO ₃	80%	3.2	50	40	26

Figure R35. a, XAS spectra at Co K-edge of CoPc/ppy/GDE and **b**, Co K-edge FT EXAFS spectra in the R space of CoPc/ppy/GDE of initial state and after 200 hours in CO₂-saturated 0.1 M KHCO₃ with an H-type cell at -0.75 V versus RHE.

Figure R36. Comparison of the Co 2p XPS spectra of CoPc/ppy/GDE of initial state and after 200 hours in CO₂-saturated 0.1 M KHCO₃ with an H-type cell at -0.75 V versus RHE.

Figure R37. **a**, TEM image of post-reacted CoPc/ppy/GDE after 200 hours. **b**, HAADF-STEM image of post-reacted CoPc/ppy/GDE after 200 hours.

In summary, our in situ electropolymerization yields CoPc/ppy/GDE electrode with robust C–C covalent bonds between CoPc and ppy, ppy and GDEs. Performance evaluations including activity and stability tests, as well as the structural and spectroscopic analyses demonstrate the unique advantages of CoPc/ppy/GDE for CO₂RR, which presents a significant improvement in the design of molecular catalysts for sustainable CO₂-to-fuel technologies., To highlight the merits of our electropolymerization-derived CoPc/ppy/GDE electrode, the introduction section has been revised to offer a more explicit comparison with conventional synthesis methods.

Reference

- [1] Chen, J. M. et al. Cobalt phthalocyanine cross-linked polypyrrole for efficient electroreduction of low concentration CO₂ to CO. *Chemsuschem* **15**, e202201455 (2022).
- [2] Zhang, A. J. et al. Electrocatalytic reduction of carbon dioxide by cobalt-phthalocyanine-incorporated polypyrrole. *Electrochem. Solid St.* **12**, E17–E19 (2009).

“Electrocatalytic conversion of CO₂ into value-added chemicals can mitigate the climate crisis induced by greenhouse gases while simultaneously enabling renewable energy storage^{1–3}. Among various catalytic systems, molecular catalysts, particularly metal phthalocyanines and metal porphyrins, have emerged as promising candidates due to their atomically defined active sites and tunable electronic structures^{4–6}. However, under operating conditions, especially those with industrially relevant current densities (>200 mA cm⁻²), molecular catalysts often suffer from leaching, agglomeration, or even structural degradation, which hinder their practical applications^{7–12}. Recently, metal phthalocyanine-based molecular catalysts have demonstrated highly selective CO₂ electrolysis at high current densities in flow reactors^{13,14}. To this end, these molecular catalysts are generally coated onto the carbon-based GDEs to circumvent their limitations of aggregation and poor conductivity, with the majority of the previous works utilizing the noncovalent anchoring strategies. In these strategies, binders like Nafion are usually required, scaffolds such as carbon nanotube (CNT) and graphene are also included to enhance the dispersion

and adhesion strength of molecular catalysts through π - π stacking interaction¹⁵⁻¹⁷.

Considering the possible loss of molecular catalysts during operation (especially at high current densities), anchoring them directly onto carbon-based GDEs through strong covalent bonds is a desirable strategy¹⁸⁻²⁰. Presently, direct covalent grafting of metal phthalocyanine/porphyrin based molecular catalysts onto carbon-based GDEs would require the use of these materials with specially designed functional groups (such as amino group) or polymerizable groups (such as carbazole group). Until now, due to the inherent chemical inertness of pristine metal phthalocyanines/porphyrins, direct electrochemical polymerization or covalent grafting of them onto the carbon-based GDEs is challenging. On the other hand, among common covalent bonds that could form between metal phthalocyanines/porphyrins (or their derivatives) and carbon-based GDEs, C-C bond is more inert and robust for extreme reaction conditions, when comparing to those hydrolysable amide and ester bonds¹⁹.

Herein, using pristine CoPc as the example, we report an in situ electro-polymerization strategy for the direct covalent grafting of pristine metal phthalocyanines onto GDEs. This method occurs at room temperature and leverages the conductive polymer ppy as a molecular “glue”, forming robust C-C bonds with pristine CoPc; meanwhile, besides as the scaffold, ppy-based matrix also shows electron donation effects to promote the activity of active Co⁺ sites. Finally, to bridge the gap between material innovation and industrial application, we further engineered a membrane electrode assembly (MEA) electrolyzer incorporating the CoPc/ppy/GDE cathode. This device achieves 50 hours of stable CO production at ampere-level current densities (1 A cm⁻²) in alkaline media. The proposed strategy here effectively addresses the challenge of direct covalent grafting of pristine metal phthalocyanines onto GDEs for efficient and durable CO₂ reduction reaction (CO₂RR) under industrially relevant current densities (>200 mA cm⁻²).”

Reviewer #3:

Comment: The authors presented a study on molecular catalyst immobilization through direct covalent grafting, demonstrating that CoPc anchored via pyrrole linkers (CoPc/ppy@GDE) exhibits enhanced stability and catalytic performance. The evidence from XAFS and XPS provides interesting insights into the electronic effects of pyrrole grafting on the Co center, while in-situ SERS and SEIRAS offer valuable mechanistic information about Co oxidation states and their role in the reaction pathway.

I believe the fundamental observation of enhanced stability through pyrrole-based grafting is well-supported; However, I have a number of significant scientific and technical concerns that should be addressed in a major revision. I think the work could be suitable for publication after the authors strengthen their mechanistic understanding and provide more comprehensive evidence. Here are some detailed comments:

Comment 1: The manuscript lacks a direct comparison of electrochemical impedance between CoPc@GDE and CoPc/ppy@GDE systems. To better understand the electron transfer characteristics of these catalysts, please provide Nyquist plots along with the corresponding serial resistance values for both systems.

Reply to comment 1: Thank you for your suggestion regarding the comparison of electrochemical impedance between the CoPc/GDE and CoPc/ppy/GDE systems. We agree that this comparison is essential for understanding the electron transfer characteristics of these catalysts. To address your comment, we have conducted electrochemical impedance spectroscopy (EIS) measurements at -0.1 V vs. RHE. As shown in Figure R38, CoPc/ppy/GDE and CoPc/GDE exhibit comparable solution resistances R_1 of 10.59Ω and 9.81Ω , respectively. Equivalent circuit modeling yields a much lower electrode resistance R_2 of 4.53Ω for CoPc/ppy/GDE than that of CoPc/GDE (9.68Ω), as well as an almost halved interfacial charge transfer resistance R_3 (38.47Ω versus 69.20Ω ; Table R6). The measurements of EIS further confirm the lower charge-transfer resistance (and correspondingly improved reaction kinetics) of CoPc/ppy/GDE than the control sample of CoPc/GDE.

Figure R38. The EIS of CoPc/ppy/GDE and CoPc/GDE in 0.1 M CO_2 -saturated KHCO_3 electrolyte at -0.1 V versus RHE.

Table R6. The values derived from EIS Nyquist plot fitting.

	Rs	R1	R2
CoPc/ppy/GDE	9.81	4.532	38.47
CoPc/GDE	10.59	9.678	69.20

“Besides, the measurements of electrochemical impedance spectroscopy (EIS) further confirm the lower charge-transfer resistance (and correspondingly improved reaction kinetics) of CoPc/ppy/GDE than the control sample of CoPc/GDE (Supplementary Fig. 17 and Supplementary Table 3).”

Comment 2: In lines 130-133, the notation system using Co(I)-N and Co(II)-N creates confusion in interpreting the oxidation states of cobalt. For example, Co(I)-N in CoPc@GDE can be interpreted specifically refers to Co(+1) coordinated with nitrogen. Please provide clearer terminology to distinguish the different cobalt species.

Reply to comment 2: Thank you for your insightful suggestions regarding the notation of cobalt oxidation states in the CoPc/ppy/GDE catalyst. We understand the potential confusion caused by the notation and would like to take the opportunity to clarify this aspect.

To avoid confusion and ensure clarity, we have revised the terminology used to describe the cobalt species in our catalyst. Specifically, Fourier-transform extended X-ray absorption fine structure (FT-EXAFS) analyses identify a dominant Co–N coordination peak at $\sim 1.5 \text{ \AA}$ for both CoPc/ppy/GDE and pristine CoPc, indicating $\text{Co}^{2+}\text{-N}$ as the primary species in them (Figure R39e). Wavelet-transforms of EXAFS spectra further confirm this conclusion by revealing prominent intensity maxima at $\sim 5.9 \text{ \AA}^{-1}$; meanwhile, a minor intensity maximum is exclusively resolved at $\sim 8.2 \text{ \AA}^{-1}$ over CoPc/ppy/GDE, which is inferred to arise from the $\text{Co}^+\text{-N}$ species (Fig. 39f–g). The corresponding change in the revised manuscript is as follows:

Figure R39. Structural analyses and intrinsic activity of CoPc/ppy/GDE. **a**, Schematic illustration of CoPc/ppy/GDE. **b**, TEM image of CoPc/ppy/GDE. **c**, HAADF-STEM image of CoPc/ppy/GDE. The atomically dispersed Co sites are marked by red circles. **d**, XAS spectra at Co K-edge of CoPc/ppy/GDE and the reference Co foil and CoPc samples. **e**, Co K-edge FT EXAFS spectra in the R space of CoPc/ppy/GDE and reference Co foil and CoPc samples. **f**, **g**, WT-EXAFS plots of CoPc samples and CoPc/ppy/GDE. **h**, LSV curves of CoPc/ppy/GDE, CoPc/CNT-GDE, CoPc/GDE and ppy/GDE during CO₂RR in an H-cell with CO₂-saturated 0.1 M KHCO₃ electrolyte. **i**, CV curve of CoPc/ppy/GDE during CO₂RR in an H-cell with CO₂-saturated 0.1 M KHCO₃ electrolyte.

“Fourier-transform extended X-ray absorption fine structure (FT-EXAFS) analyses identify a dominant Co–N coordination peak at ~ 1.5 Å for both CoPc/ppy/GDE and pristine CoPc, indicating Co²⁺–N as the primary species in them (Fig. 2e). Wavelet-transforms of EXAFS (WT-EXAFS) spectra further confirm this conclusion by revealing prominent intensity maxima at ~ 5.9 Å⁻¹; meanwhile, a minor intensity maximum is exclusively resolved at ~ 8.2 Å⁻¹ over CoPc/ppy/GDE, which is inferred to arise from the Co⁺–N species (Fig. 2f–g, Supplementary Fig. 11)^{26,27}.”

Comment 3: There appears to be a typographical error in line 135 where "XAENS" should be corrected to "XANES"

Reply to comment 3: Thank you for pointing out the typographical error in line 135. We have

revised this error and reviewed the manuscript to make the necessary correction.

We appreciate your help in ensuring the accuracy of our manuscript.

Comment 4: While Figure 2h demonstrates performance comparisons between CoPc/ppy@GDE, CoPc-GDE, and CoPc-CNT, the data appears to be collected using different cobalt loadings. Additionally, there is a discrepancy in the Co content values reported in the main text versus Supplementary Table 1. To make a fair assessment of catalyst performance, please provide comparisons with normalized cobalt content across all catalyst systems.

Reply to comment 4: Thank you for your insightful comments regarding the cobalt loading values for performance comparisons and discrepancies in Figure R40 and Table R7.

Figure R40. LSV curves normalized by ECSA of CoPc/ppy/GDE, CoPc-GDE, and CoPc/CNT-GDE during CO₂RR in an H-cell with CO₂-saturated 0.1 M KHCO₃ electrolyte with the scan rate of 1 mV s⁻¹.

We have checked the cobalt loadings for various samples, and the Co content values for CoPc/ppy/GDE, CoPc/GDE, and CoPc/CNT-GDE samples are measured to be 0.0837 wt%, 0.0046 wt% and 6.4097 wt%, respectively (Table R7). Based on these values, we have derived the cobalt loading-normalized LSV curves eliminating the influence from GDE for CoPc/ppy/GDE, CoPc/GDE, and CoPc/CNT-GDE samples. As shown in Figure 40, the intrinsic activity (Co loading-normalized specific activities) follows the order of CoPc/ppy/GDE > CoPc/CNT-GDE > CoPc/GDE, demonstrating the unique advantages of ppy-mediated covalent grafting (including binder-free interface, highly conductive ppy linker, electron donation effect of ppy, appropriate dispersion and spatial configuration of Co single-atom sites) in CoPc/ppy/GDE for CO₂RR. We have corrected the corresponding discussion in the revised manuscript. Thank the reviewer again!

Table R7. Co content of CoPc/ppy/GDE, CoPc/CNT-GDE and CoPc/GDE

Catalyst	Content (wt. %)
CoPc/ppy/GDE	0.0837%
CoPc/CNT-GDE	6.4097%
CoPc/GDE	0.0046%

“To clarify the intrinsic activity of these samples, their cobalt loadings were measured by ICP-OES (Supplementary Table 1). Based on these values, the Co loading-normalized specific activity of these samples has been derived (measured at a scan rate of 1 mV s⁻¹ to minimize the capacitive contributions), which follows the order of CoPc/ppy/GDE > CoPc/CNT-GDE > CoPc/GDE, demonstrating the unique advantages of ppy-mediated covalent grafting (including binder-free interface, highly conductive ppy linker, electron donation effect of ppy, appropriate dispersion and spatial configuration of Co single-atom sites) in CoPc/ppy/GDE for CO₂RR (Supplementary Fig. 16).”

Comment 5: The manuscript attributes the redshifted and broadened CO* bands observed on CoPc/ppy@GDE to surface coverage effects. However, two important questions need to be addressed: First, could dynamic dipole coupling of CO* produce similar spectral changes in single-atom catalysts? Second, the cited reference 37 indicates that high surface coverage typically results in blue-shifted CO* bands, which contradicts the current explanation. Please clarify these apparent discrepancies.

Reply to comment 5: We sincerely appreciate your valuable comments. We would like to offer the following clarifications regarding the points you raised. First, dynamic dipole coupling typically results in a blue shift of the CO* vibrational bands on high-coverage metal surfaces (as observed on copper surfaces in Reference 37), due to enhanced dipole interactions between adsorbed CO molecules. However, in single-atom catalysts such as CoPc/ppy/GDE, the active sites are atomically dispersed (e.g., isolated Co single atoms), different from the status of bulk metals or nanoparticle systems. Under this condition, the CO* adsorption sites of Co are spatially isolated, leading to little chance for CO*–CO* interactions. Therefore, using the dynamic dipole coupling to explain the red-shifted CO* bands over CoPc/ppy/GDE is wrong.

Second, as discussed above, minimal dynamic dipole coupling effects occur for CoPc/ppy/GDE, as contrary to the metallic copper surfaces with high CO* coverage mentioned in Reference 37. Regarding the observed red-shifted and broadened CO* bands, we infer that the unique electronic structure of CoPc/ppy/GDE, due to the electron donation effects from ppy, may be responsible for this spectral change of CO* (Figure R41 and R42). After careful checking the data, we found that no direct relationship could be established between the shift degree of CO* frequency and its formation rate. Therefore, we have deleted these descriptions to avoid misinterpretation, and

have made the corresponding revision in the manuscript. Collectively, according to your comments and after carefully checking the requirements for dynamic dipole coupling, we have realized that the spatial separation of Co single atoms significantly weakens the dynamic dipole coupling effects in CoPc/ppy/GDE, which contribute minimally to the shift of CO^* vibrational frequency. We have revised the manuscript to avoid misinterpretation, as shown below. Once again, we thank you for your insightful comments on our manuscript.

Figure R41. In situ ATR-SEIRAS spectra of the CoPc/ppy/GDE during CO_2RR under applied bias ranging from -0.4 to -0.8 V versus RHE in CO_2 -saturated 0.1 M KHCO_3 .

Figure R42. In situ ATR-SEIRAS spectra of (a) CoPc/CNT-GDE (b) CoPc/GDE (c) ppy/GDE during CO_2RR under applied bias ranging from -0.4 to -0.8 V versus RHE in CO_2 -saturated 0.1 M KHCO_3 .

“For the CoPc/ppy/GDE electrode (Fig. 3d), upon the application of negative potentials, surface-bonded CO (CO^) appears and grows in intensity with increasing negative potential, indicating the gradual transformation of CO_2 to CO^* . Under the same condition, negligible CO^* signals were observed on ppy/GDE and CoPc/GDE, while CoPc/CNT-GDE prepared according to the previous report showed a delayed onset potential of CO^* formation along with a weaker signal intensity (Supplementary Figs. 28–30). Meanwhile, an inverse absorption band appears at 1230 cm^{-1} and becomes more prominent as the applied bias goes more reductive; this phenomenon can*

be ascribed to the rapid consumption of HCO_3^- at large overpotentials. It is noted that under identical bias, CoPc/ppy/GDE exhibits the largest absorption bands for CO^ and HCO_3^- among tested samples. These observations suggest the improved activity of CoPc/ppy/GDE for CO_2 RR than the control samples of ppy/GDE, CoPc/GDE and CoPc/CNT-GDE.”*

Comment 6: The time-dependent blue shift in absorption edges shown in Figure 3d indicates a gradual increase in cobalt valence state during the reaction. This observation seems to contradict the proposed mechanism where catalyst stability and activity are attributed to maintained Co^+ active species. To resolve this inconsistency, please provide time-resolved in-situ SERS data of CoPc/ppy@GDE at fixed potential.

Reply to comment 6: We appreciate your insightful comments regarding the blue shifts of absorption edge in the XANES spectra and the change behaviour of Co valence state in CoPc/ppy/GDE during reaction. According to your comments and suggestions, we have conducted additional experiments and analyses to provide a more comprehensive understanding of the underlying mechanisms.

We should make it clear that our XAS measurements were performed under ex situ conditions on an easyXAFS setup (Figure R43), due to the low intensity of collected signals with a conventional electrochemical cell; the CoPc/ppy/GDE electrode was polarized at -0.8 V versus RHE, and at different reaction times the CoPc/ppy/GDE electrode was taken out from the electrochemical cell and subjected to XAS measurements. Under this condition, oxygen-containing species (OH^- , H_2O and reaction intermediates such as HCO_3^- and COOH^*) may adsorb on CoPc/ppy/GDE, and upon removal of the applied potential valence-state increase of the Co centre would occur (Figure R44a)¹. More evidences come from the EXAFS analysis, which reveals a slight elongation of the Co–N bond (from 1.98 to 2.05 Å) and importantly the emergence of a Co–O coordination peak at ~ 1.2 Å (Figure R44b).

These phenomena are consistent with the coordination of oxygen-containing reactants/intermediates (e.g., OH^- , HCO_3^- or HCOO^* species) to the Co centre and the correspondingly relaxed structure (elongated Co–N bond) in CoPc/ppy/GDE. In comparison, the blue-shift of absorption edge in the XANES curves and the Co–O coordination in the EXAFS spectra for CoPc/CNT-GDE are not as prominent as those of CoPc/ppy/GDE (Figure R45), indicating weaker adsorption of OH^- , CO_2 , H_2O reactants and less accumulation of oxygen-containing intermediates over CoPc/CNT-GDE than CoPc/ppy/GDE. DFT calculations provide additional evidences, which reveal that Co sites in CoPc/ppy/GDE exhibit a strong interaction with intermediates; such a feature is crucial for accelerating CO_2 conversion (Figure R46). We attribute the enhanced stabilization of reaction intermediates on CoPc/ppy/GDE than CoPc/CNT-GDE to the electron donation effects from ppy to CoPc. Regarding the change behaviour of Co^+ species under operando conditions, we fixed the potential at -1.4 V versus RHE and monitored the evolution of Co^+ and Co^{2+} species in CoPc/ppy/GDE through in situ surface-enhanced Raman spectroscopy

(SERS). As shown in Figure R47, with the continuous application of a negative potential of -1.4 V versus RHE, no obvious degradation in the peak intensity of Co^+ species (at ca. 1110 cm^{-1}) could be detected on CoPc/ppy/GDE over a reaction period of 30 min; while for CoPc/CNT-GDE, a gradual decrease of Co^+ species appears, possibly due to the progressive loss of CoPc from CoPc/CNT-GDE electrode.

In terms of further reduction of Co^+ species to zero-valence Co, a previous study demonstrated that Co^+ species in CoPc is stable up to the potential of -2.2 V versus RHE, as the percentage of Co^+ species reaches a steady state from ca. -1.0 V versus RHE². If a further reduction of Co^+ species occurs, a decreased Co^+ percentage is expected. Consistent with the previous study, our results under varied potentials reveal that the signal intensity of Co^+ species (at 1110 cm^{-1}) over CoPc/ppy/GDE becomes prominent at ca. -0.7 V versus RHE, and beyond this potential the spectrum reaches a relatively steady state (Figure R48).

Figure R43. Picture of easyXAFS/XES.

Figure R44. a, XAFS spectra at the Co K-edge of CoPc/ppy/GDE during CO_2RR at -0.8 V versus

RHE from 0 to 2 h in CO₂-saturated 0.1 M KHCO₃. **b**, Co K-edge EXAFS spectra of CoPc/ppy/GDE during CO₂RR at -0.8 V versus RHE from 0 to 2 h in CO₂-saturated 0.1 M KHCO₃.

Figure R45. a, XAFS spectra at the Co K-edge of CoPc/CNT-GDE during CO₂RR at -0.8 V versus RHE from 0 to 2 h in CO₂-saturated 0.1 M KHCO₃. **b**, Co K-edge EXAFS spectra of CoPc/CNT-GDE during CO₂RR at -0.8 V versus RHE from 0 to 2 h in CO₂-saturated 0.1 M KHCO₃.

Figure R46. Schematic diagram of differential charge density of CoPc/ppy and Bader charge analysis. The cyan (yellow) region shows electron loss (gain).

Figure R47. Time-dependent in situ surface-enhanced Raman spectra of the (a) CoPc/ppy/GDE (b) CoPc/CNT-GDE during CO₂RR from 0 to 30 min at -1.4 V versus RHE in CO₂-saturated 0.1 M KHCO₃.

Figure R48. The ratio of Co⁺:Co²⁺ during CO₂RR under applied bias ranging from -0.4 to -1.4 V versus RHE in CO₂-saturated 0.1 M KHCO₃.

Finally, the CoPc/ppy/GDE electrode after testing for 200 h was subjected to XAS, XPS and HAADF-STEM analyses. As shown in Figure R49, no obvious changes in the valence state and coordination environment of Co could be resolved for post-reacted CoPc/ppy/GDE. XPS analysis reveals that although the intensities of Co 2p peaks in post-reacted CoPc/ppy/GDE have decreased (possibly due to the contamination from electrolyte), their positions remain largely unaltered (Figure R50). In addition, HAADF-STEM measurements suggest that the morphology of post-reacted CoPc/ppy/GDE catalyst is retained, with well-preserved atomically dispersed Co sites (Figure R51). All these post-reaction analyses together demonstrate that our CoPc/ppy/GDE electrode is highly resistant to structural collapse/changes under prolonged electrolysis, the negligible demetallation, reduction and aggregation of Co active sites in CoPc/ppy/GDE guarantee its promising stability under highly reductive and high current density conditions.

Figure R49. **a**, XAS spectra at Co K-edge of CoPc/ppy/GDE and **b**, Co K-edge FT EXAFS spectra in the R space of CoPc/ppy/GDE of initial state and after 200 hours in CO₂-saturated 0.1 M KHCO₃ with an H-type cell at -0.75 V versus RHE.

Figure R50. Comparison of the Co 2p XPS spectra of CoPc/ppy/GDE of initial state and after 200 hours in CO₂-saturated 0.1 M KHCO₃ with an H-type cell at -0.75 V versus RHE.

Figure R51. **a**, TEM image of post-reacted CoPc/ppy/GDE after 200 hours. **b**, HAADF-STEM image of post-reacted CoPc/ppy/GDE after 200 hours.

Combined all the above experiments and analyses together, we propose that Co⁺ could be the active species in CoPc/ppy/GDE for CO₂RR. Co⁺ species are possibly stabilized by the microenvironment and ligands, forming a dynamically stable active centre; upon the removal of applied potential, oxygen-containing species (OH⁻, H₂O and reaction intermediates such as HCO₃⁻ and COOH*) may push the oxidation state of Co to a higher value.

We have included these additional experimental data and detailed mechanistic explanations in the revised manuscript. We hope that these additional analyses would provide a clearer understanding on the behaviour of CoPc/ppy/GDE during CO₂RR. Thank you once again for your valuable feedback.

References

[1] Wang, S. F. et al. Manipulating C-C coupling pathway in electrochemical CO₂ reduction for selective ethylene and ethanol production over single-atom alloy catalyst. *Nat. Commun.* **15**, 10247

(2024).

[2] Ren, S. et al. Catalyst aggregation matters for immobilized molecular CO₂RR electrocatalysts. *J. Am. Chem. Soc.* **145**, 4414–4420 (2023).

Comment 7: The authors showed the changes of CoPc/ppy@GDE at the fixed potential over the time and discuss the changes of the XANES and EXAFS spectra. However, control sample CoPc@GDE is also required to be measured over the same period of time to investigate whether the changes of the Co oxidation states are associated with ppy introduction.

Reply to comment 7: Thank you for your suggestion regarding the measurements of control sample to understand the change behaviour of Co in CoPc/ppy/GDE and the impact of introduced ppy. As the loading amount of CoPc in CoPc/GDE is significantly low (below the detection limit of the instrument), CoPc/CNT-GDE (without ppy) was employed as the reference to decouple the contribution of ppy in CoPc/ppy/GDE.

As mentioned above, our XAS measurements were performed under ex situ conditions on an easyXAFS setup, due to the low intensity of collected signals with a conventional electrochemical cell. At different reaction times, the CoPc/ppy/GDE was taken out from the electrochemical cell and subjected to XAS measurements. Under this condition, oxygen-containing species (OH⁻, H₂O and reaction intermediates such as HCO₃⁻ and COOH*) may adsorb on CoPc/ppy/GDE, and upon removal of the applied potential valence-state increase of the Co centre would occur (XANES spectra in Figure R52a)¹. More evidences come from the EXAFS analysis, which reveals a slight elongation of the Co–N bond (from 1.98 to 2.05 Å) and importantly the emergence of a Co–O coordination peak at ~1.2 Å (Figure R52b). These phenomena are consistent with the coordination of oxygen-containing reactants/intermediates (e.g., OH⁻, HCO₃⁻ or HCOO* species) to the Co centre and the correspondingly relaxed structure (elongated Co–N bond) in CoPc/ppy/GDE. In comparison, the blue-shift of absorption edge in the XANES curves and the Co–O coordination in the EXAFS spectra for CoPc/CNT-GDE are not as prominent as those of CoPc/ppy/GDE (Figure R53), indicating weaker adsorption of OH⁻, CO₂, H₂O reactants and less accumulation of oxygen-containing intermediates over CoPc/CNT-GDE than CoPc/ppy/GDE. DFT calculations provide additional evidences, which reveal that Co sites in CoPc/ppy/GDE exhibit a strong interaction with intermediates; such a feature is crucial for accelerating CO₂ conversion (Figure R54). We attribute the enhanced stabilization of reaction intermediates on CoPc/ppy/GDE than CoPc/CNT-GDE to the electron donation effects from ppy to CoPc.

We have added these data and discussion in the revised manuscript. Thanks again for your valuable feedback.

Figure R52. **a**, XAFS spectra at the Co K-edge of CoPc/ppy/GDE during CO₂RR at -0.8 V versus RHE from 0 to 2 h in CO₂-saturated 0.1 M KHCO₃. **b**, Co K-edge EXAFS spectra of CoPc/ppy/GDE during CO₂RR at -0.8 V versus RHE from 0 to 2 h in CO₂-saturated 0.1 M KHCO₃.

Figure R53. **a**, XAFS spectra at the Co K-edge of CoPc/CNT-GDE during CO₂RR at -0.8 V versus RHE from 0 to 2 h in CO₂-saturated 0.1 M KHCO₃. **b**, Co K-edge EXAFS spectra of CoPc/CNT-GDE during CO₂RR at -0.8 V versus RHE from 0 to 2 h in CO₂-saturated 0.1 M KHCO₃.

Figure R54. Schematic diagram of differential charge density of CoPc/ppy and Bader charge analysis. The cyan (yellow) region shows electron loss (gain).

Reference

[1] Wang, S. F. et al. Manipulating C-C coupling pathway in electrochemical CO₂ reduction for selective ethylene and ethanol production over single-atom alloy catalyst. *Nat. Commun.* **15**, 10247 (2024).

Comment 8: The author mentioned that “The absorption edges blueshifted with reaction time, indicating that the valence state of Co gradually increased during the reaction.” Is it reversible? If so, after the reaction, (post-analysis) showed the similar states of Co or does it go back to the initial states. If not, the authors want to discuss how the ppy helps the charge transfer. The changes of the oxidation states can be an indication of the degradation of the CoPc. Show the post-analysis, (TEM, EXAFS, XANES, XPS) can provide the information whether the catalyst maintained the initial CoPc/ppy structure.

Reply to comment 8: We appreciate your comments regarding the oxidation states of Co in CoPc/ppy@GDE catalyst.

As mentioned in above replies, our XAS measurements were performed under ex situ conditions. Upon the removal of applied potentials, coordination of oxygen-containing reactants/intermediates (e.g., OH⁻, HCO₃⁻ or HCOO* species) to the Co centre occurs and leads to valence-state increase of the Co centre. Therefore, we can observe the blue-shift of absorption edge in the XANES curves and the emergence of a Co–O coordination peak at ~1.2 Å in the EXAFS spectra for CoPc/CNT-GDE. As long as the negative potentials are applied (or as long as the reaction proceeds without stop), this increase of Co valence state in CoPc/CNT-GDE is reversible.

In the previous study, Co⁺ is considered as the active species for CO₂RR and through the diagnostic Raman signatures of Co⁺ (at ca. 1110 cm⁻¹) and Co²⁺ species (at ca. 1140 cm⁻¹), their identification and quantification have been achieved². To provide a real-time monitor of the Co valence state (change of Co⁺ and Co²⁺ species) in CoPc/CNT-GDE, we also performed surface-enhanced Raman spectroscopy (SERS) measurements under operando conditions. We fixed the potential at -1.4 V versus RHE and monitored the evolution of Co⁺ and Co²⁺ species in CoPc/ppy/GDE. As shown in Figure R55, with the continuous application of a negative potential of -1.4 V versus RHE, no obvious degradation in the peak intensity of Co⁺ species (at ca. 1110 cm⁻¹) could be detected on CoPc/ppy/GDE over a reaction period of 30 min; while for CoPc/CNT-GDE, a gradual decrease of Co⁺ species appears, possibly due to the progressive loss of CoPc from CoPc/CNT-GDE electrode (Figure R56).

In terms of further reduction of Co⁺ species to zero-valence Co, a previous study demonstrated that Co⁺ species in CoPc is stable up to the potential of -2.2 V versus RHE, as the percentage of Co⁺ species reaches a steady state from ca. -1.0 V versus RHE². If a further reduction of Co⁺ species occurs, a decreased Co⁺ percentage is expected. Consistent with the previous study, our results under varied potentials reveal that the signal intensity of Co⁺ species (at 1110 cm⁻¹) over CoPc/ppy/GDE becomes prominent at ca. -0.7 V versus RHE, and beyond this potential the spectrum reaches a relatively steady state (Figure R57).

Figure R55. a, b, In situ SERS of the CoPc/ppy/GDE and CoPc/CNT-GDE under applied bias ranging from -0.4 to -1.4 V versus RHE.

Figure R56. Time-dependent in situ surface-enhanced Raman spectra of the (a) CoPc/ppy/GDE (b) CoPc/CNT-GDE during CO_2RR from 0 to 30 min at -1.4 V versus RHE in CO_2 -saturated 0.1 M KHCO_3 .

Figure R57. The ratio of $\text{Co}^+ : \text{Co}^{2+}$ during CO_2RR under applied bias ranging from -0.4 to -1.4 V versus RHE in CO_2 -saturated 0.1 M KHCO_3 .

Finally, according to your suggestion, we performed detailed structural characterizations using XAS, XPS and HAADF-STEM on the post-reacted CoPc/ppy/GDE catalyst (polarized at -0.75 V versus RHE for 200 hours). As shown in Figure R58, no obvious changes in the valence state and coordination environment of Co could be resolved for post-reacted CoPc/ppy/GDE. XPS analysis reveals that although the intensities of Co 2p peaks in post-reacted CoPc/ppy/GDE have decreased (possibly due to the contamination from electrolyte), their positions remain largely unaltered (Figure R59). In addition, HAADF-STEM measurements suggest that the morphology of post-reacted CoPc/ppy/GDE catalyst is retained, with well-preserved atomically dispersed Co sites (Figure R60). All these post-reaction analyses together demonstrate that our CoPc/ppy/GDE electrode is highly resistant to structural collapse/changes under prolonged electrolysis, the negligible demetallation, reduction and aggregation of Co active sites in CoPc/ppy/GDE guarantee its promising stability under highly reductive and high current density conditions.

Figure R58. **a**, XAS spectra at Co K-edge of CoPc/ppy/GDE and **b**, Co K-edge FT EXAFS spectra in the R space of CoPc/ppy/GDE of initial state and after 200 hours in CO_2 -saturated 0.1 M KHCO_3 with an H-type cell at -0.75 V versus RHE.

Figure R59. Comparison of the Co 2p XPS spectra of CoPc/ppy/GDE of initial state and after 200 hours in CO_2 -saturated 0.1 M KHCO_3 with an H-type cell at -0.75 V versus RHE.

Figure R60. **a**, TEM image of post-reacted CoPc/ppy/GDE after 200 hours. **b**, HAADF-STEM image of post-reacted CoPc/ppy/GDE after 200 hours.

Combined all the above experiments and analyses together, we propose that Co^+ could be the active species in CoPc/ppy/GDE for CO_2RR . Co^+ species are possibly stabilized by the microenvironment and ligands, forming a dynamically stable active centre; upon the removal of applied potential, oxygen-containing species (OH^- , H_2O and reaction intermediates such as HCO_3^- and COOH^*) may push the oxidation state of Co to a higher value.

We have included these additional experimental data and detailed mechanistic explanations in the revised manuscript. We hope that these additional analyses would provide a clearer understanding on the behaviour of CoPc/ppy/GDE during CO_2RR . Thank you once again for your valuable feedback.

References

- [1] Wang, S. F. et al. Manipulating C-C coupling pathway in electrochemical CO_2 reduction for selective ethylene and ethanol production over single-atom alloy catalyst. *Nat. Commun.* **15**, 10247 (2024).
- [2] Ren, S. et al. Catalyst aggregation matters for immobilized molecular CO_2RR electrocatalysts. *J. Am. Chem. Soc.* **145**, 4414–4420 (2023).

“Previous studies have claimed that Co^+ species in CoPc formed by one-electron reduction of Co^{2+} , are the active site for CO_2RR ^{31,32}. Through the diagnostic Raman signatures of Co^+ (at ca. 1110 cm^{-1}) and Co^{2+} species (at ca. 1140 cm^{-1}), the identification and quantification of Co^+ active species have been achieved^{33–35}. Following these reports, the evolution of CoPc/ppy/GDE and the distribution of Co^+ and Co^{2+} species (i.e. the ratio of Co^+ to Co^{2+}) with applied potentials were monitored through in situ surface-enhanced Raman spectroscopy (SERS). As shown in Figs. 3a and b, characteristic peaks could be resolved at 1113 and 1146 cm^{-1} at -0.4 V versus RHE for both electrodes of CoPc/ppy/GDE and CoPc/CNT-GDE, which are reported to be associated with the Co^+ and Co^{2+} species, respectively³⁵. The signal intensity of Co^+ species (at 1113 cm^{-1}) over CoPc/ppy/GDE becomes prominent at ca. -0.7 V versus RHE, consistent with the redox behaviour

of $\text{Co}^{2+}/\text{Co}^+$ in CoPc; beyond this potential, the spectrum reaches a relatively steady state. In comparison, the formed Co^+ species over CoPc/CNT-GDE is significantly lower in amount (Supplementary Figs. 25 and 26). Using the Raman peak area, the ratios of Co^+ to Co^{2+} in CoPc/ppy/GDE and CoPc/CNT-GDE at various applied potentials were calculated. The results in Fig. 3c show that the formed Co^+ species over CoPc/ppy/GDE are always higher than that of CoPc/CNT-GDE, aligned with its better catalytic activity for CO generation³⁶. It is inferred that the electron donation effects from ppy to CoPc is responsible. Furthermore, the stability of Co^+ species during CO_2RR was evaluated by in situ SERS. As shown in Supplementary Fig. 27, with the continuous application of a negative potential of -1.4 V versus RHE, no obvious degradation in the peak intensity of Co^+ species could be detected on CoPc/ppy/GDE over a reaction period of 30 min; while for CoPc/CNT-GDE, a gradual decrease of Co^+ species appears, possibly due to the progressive loss of CoPc from CoPc/CNT-GDE electrode.”

“To further assess the evolutionary nature of the atomically dispersed Co active sites, ex situ XAS measurements were carried out. Co K-edge XANES spectra of CoPc/ppy/GDE were collected after reacting in a CO_2 -saturated 0.1 M KHCO_3 solution for different times. Under this condition, oxygen-containing species (OH^- , H_2O and reaction intermediates such as HCO_3^- , CO^* and COOH^*) may adsorb on CoPc/ppy/GDE, and upon removal of the applied potential valence-state increase of the Co centre would occur (Supplementary Fig. 31)^{38,39}. More evidences come from the EXAFS analysis, which reveals a slight elongation of the Co–N bond (from 1.98 to 2.05 \AA)^{40,41} and importantly the emergence of a Co–O coordination peak at $\sim 1.2\text{ \AA}$ (Fig. 3e)⁴². These phenomena are consistent with the coordination of oxygen-containing reactants/intermediates (e.g., OH^- , HCO_3^- or COOH^* species) to the Co centre and the correspondingly relaxed structure (elongated Co–N bond) in CoPc/ppy/GDE. In comparison, the blue-shift of absorption edge in the XANES curves and the Co–O coordination in the EXAFS spectra for CoPc/CNT-GDE are not as prominent as those of CoPc/ppy/GDE (Supplementary Fig. 32), indicating weaker adsorption of OH^- , CO_2 , H_2O reactants and less accumulation of oxygen-containing intermediates over CoPc/CNT-GDE than CoPc/ppy/GDE.”

Comment 9: Can it be applicable to other Pc such as NiPc and FePc for electrochemical CO_2RR ?

Reply to comment 9: Thank you for your question regarding the applicability of our strategy to other Pc-based catalysts, such as NiPc, FePc and CuPc, for the electrochemical CO_2RR . We have conducted these experiments, the results show that FePc, NiPc and CuPc can be successfully incorporated into the ppy matrix and act as effective electrocatalysts for CO_2RR (Figure R61). Among them, NiPc/ppy/GDE exhibits a CO selectivity of 92% with a current density of 14.3 mA cm^{-2} at -0.7 V versus RHE, which are slightly inferior to that of CoPc/ppy/GDE. While FePc/ppy/GDE and CuPc/ppy/GDE demonstrate CO selectivities approaching 90%, accompanied with current densities of 9.3 and 9.8 mA cm^{-2} , respectively. These results conclusively validate the compatibility of our methodology with diverse phthalocyanine-based catalytic systems. We have included these additional experimental data and discussion in the revised manuscript.

Figure R61. **a**, LSV curves of CoPc/ppy/GDE, FePc/ppy/GDE, CuPc/ppy/GDE and NiPc/ppy/GDE during CO₂RR in an H-cell with CO₂-saturated 0.1 M KHCO₃ electrolyte. **b**, Faradaic efficiency of CoPc/ppy/GDE, FePc/ppy/GDE, CuPc/ppy/GDE and NiPc/ppy/GDE during CO₂RR at -0.7 V versus RHE in an H-cell with CO₂-saturated 0.1 M KHCO₃ electrolyte.

“Motivated by the excellent CO₂RR performance of CoPc/ppy/GDE, NiPc/ppy/GDE, CuPc/ppy/GDE, and FePc/ppy/GDE were further fabricated to extend the applicability of our covalent grafting strategy (Supplementary Fig. 24). Among them, NiPc/ppy/GDE exhibits a CO selectivity of 92% with a current density of 14.3 mA cm⁻² at -0.7 V versus RHE, which are slightly inferior to that of CoPc/ppy/GDE; while FePc/ppy/GDE and CuPc/ppy/GDE demonstrate CO selectivity approaching 90%, accompanied with current densities of 9.3 and 9.8 mA cm⁻², respectively. These results validate the compatibility of our covalent grafting strategy with diverse phthalocyanine-based molecular catalysts.”

Comment 10: How the current density affected by the loading amount of the catalyst? That can be the useful information.

Reply to comment 10: Thank you for raising this critical question. Despite the lack of precise control over the ppy molecular weight, our CoPc/ppy/GDE catalysts show controllable and reproducible electrochemical behaviors towards CO₂RR, when the electropolymerization conditions are well organized. Specifically, the electrochemical CO₂RR performance of CoPc/ppy/GDE electrodes can be systematically and finely optimized through tuning two key parameters including the number of CV scans and the feed ratio of pyrrole to CoPc, based on which the chain length of ppy and CoPc loading/incorporating could be controlled.

Figure R62a illustrates the effects of CV number during electropolymerization, the current density that corresponds to the electrochemical growth of CoPc and ppy on GDE initially increases with cycle number but exhibits a saturation trend beyond 40 cycles. We infer that mass transport or ppy resistance issues gradually dominate and cease the continuous growth of CoPc and ppy on GDE in the applied CV potential range. Correspondingly, the CO₂RR performance of yielded CoPc/ppy/GDE electrodes begins to stabilize from 40 CV cycles (Figure R62b).

Next, considering that the CO₂RR performance of CoPc/ppy/GDE electrodes is closely

correlated with the loading/incorporating amount of CoPc, we optimized the synthetic procedure by varying the pyrrole-to-CoPc feed ratio (1:6 to 6:1). As shown in Figure R63, only small activity differences could be observed for these CoPc/ppy/GDE electrodes obtained in the presence of both pyrrole and CoPc, and the best performing one is obtained with a high pyrrole-to-CoPc ratio of 4:1; when only CoPc is applied, the yielded CoPc/GDE electrode shows largely suppressed CO₂RR activity.

Based on these results, we infer that the electropolymerization step of ppy dominates the whole formation process of CoPc/ppy/GDE electrodes, as pristine CoPc is chemically less active towards electropolymerization; under this condition, ppy initiates the C–C coupling process between GDEs and ppy, ppy and pristine CoPc, thereby realizing covalent grafting of pristine CoPc onto carbon-based GDEs via robust C–C bonds.

Collectively, through utilizing ppy as the electroactive "glue", covalent grafting of pristine CoPc onto carbon-based GDEs via robust C–C bonds has been achieved. The optimal conditions for CoPc/ppy/GDE electrode fabrication are determined to be 40 of CV cycles and 4:1 of pyrrole-to-CoPc ratio that balance performance and cost efficiency. Although the precise control over the ppy molecular weight remains a challenge and is worth of future studies, the yielded CoPc/ppy/GDE electrodes show reproducible CO₂RR activity and promising stability under high current densities, suggesting the effectiveness of us in situ electropolymerization strategy for covalent grafting of pristine molecular catalysts onto carbon-based GDEs. We have added these optimizations and discussion in the revised manuscript. Thank you again!

Figure R62. a, The different deposition cycles of CoPc/ppy/GDE during the electro-polymerization in 0.01 M [BMIM]BF₄ with 4 mM pyrrole and 1 mM CoPc at 100 mV s⁻¹. The potentials were calibrated to the Ag/AgCl scale. **b**, The variation of current density with the number of deposition cycles.

Figure R63. a, The LSV of CoPc/ppy/GDE during the electro-polymerization in an H-cell with CO₂-saturated 0.1 M KHCO₃ electrolyte. The potentials were calibrated to the Ag/AgCl scale. **b**, The variation of current density with the number of deposition cycles.

In summary, we have considered the current density affected by loading amount of the catalyst. Through systematic optimization of synthesis conditions and cost-effective, we investigated the optimal number of deposition cycles for catalyst fabrication to achieve balanced performance and economic feasibility. We have included detailed information on these optimizations in the revised manuscript.

Thank you again for your valuable feedback. We look forward to your further comments and suggestions.

Reviewer #4:

Comment: This manuscript reported the synthesis of CoPc for the efficient conversion of CO₂ to CO product. A photovoltaic-electrolysis device is also constructed and achieved a record solar-to-fuel efficiency of 19.2%. Despite this work provides a complete study, the novelty of the electrocatalytic CO₂ to CO is not enough. Moreover, the electrolysis in alkaline electrolytes under high current density do not provide new insight. Whether this catalyst is suitable for MEA testing under pure water conditions. Thus, the novelty and depth of academic significance does not reach the level of Nature Communications.

Thanks for your comments. The uniqueness and progresses of this work can be summarized as follows.

(1) The importance of CO₂ electrolysis to CO.

Producing CO from CO₂ via electrolysis or reverse water-gas shift route is not only an important pathway for achieving carbon neutrality, but also can be combined with thermocatalysis (such as Fischer-Tropsch synthesis) or electrolysis to manufacture various high-value chemicals such as methanol and olefins¹⁻³. Following are some examples.

Prof. Jiao group have constructed a tandem CO₂ electrolyzer for the generation of multicarbon products at the kilowatt scale; in this tandem electrolysis system, CO₂ is first converted to CO in a CO₂ electrolyzer and then CO is subsequently transformed into C₂₊ products in a CO electrolyzer (Figure R64; Kilowatt-scale tandem CO₂ electrolysis for enhanced acetate and ethylene production, Nature Chemical Engineering, 2024, 1, 421–429).

[FIGURE REDACTED]

Figure R64. Two-step CO₂ electrolysis stack for multicarbon chemical production.

In some cases, the CO₂ and CO electrolysis catalysts could be well arranged (mixed) in the same reactor, leading to enhanced activity and product selectivity. For example, Prof. Wang group have developed a tandem catalyst consisting of CoPc and Zn-N-C to promote CH₄ production in CO₂RR; compared to CoPc or Zn-N-C alone, there is more than 100 times enhancement in CH₄/CO production rate ratio over the CoPc@Zn-N-C tandem catalyst (Figure R65; Enhancing CO₂ electroreduction to methane with a cobalt phthalocyanine and zinc–nitrogen–carbon tandem catalyst,

Angew. Chem. Int. Ed., 2020, 59, 22408–22413).

[FIGURE REDACTED]

Figure R65. CO₂ electroreduction to methane with a cobalt phthalocyanine and zinc–nitrogen–carbon tandem catalyst.

Prof. Chen group have proposed an electrocatalytic–thermocatalytic tandem strategy for carbon nanofiber (CNF) production, which integrates the co-electrolysis of CO₂ and water into syngas (CO and H₂) with a subsequent thermochemical process at relatively mild conditions (370–450 °C, 1 atm), yielding CNF at a high production rate (average 2.5 g_{carbon} g_{metals}⁻¹ h⁻¹) (Figure R66; CO₂ fixation into carbon nanofibres using electrochemical–thermochemical tandem catalysis, *Nature Catalysis*, 2024, 7, 98–109).

[FIGURE REDACTED]

Figure R66. Electrochemical–thermochemical tandem process for producing carbon nanofibers and renewable H₂.

Similarly, Prof. Sun group have developed a tandem electrochemical-CVD system, based on which 52% concentrated CO is produced and is further converted into single layer graphene film (Figure R67; Direct electrosynthesis of 52% concentrated CO on silver’s twin boundary, *Nature Communications*, 2021, 12, 2139).

[FIGURE REDACTED]

Figure R67. The schematic diagram of “two-step” synthesis of graphene from CO₂⁶.

(2) The importance and innovation of covalent grafting of pristine cobalt phthalocyanine on gas diffusion electrodes (GDEs) for CO₂RR.

Molecular catalysts that with atomically defined active sites and tunable electronic structures have demonstrated highly selective CO₂ electrolysis at high current densities in flow reactors. To this end, molecular catalysts are generally coated onto the carbon-based GDEs (to circumvent their limitations of aggregation and poor conductivity), with the majority of the previous works utilizing the noncovalent anchoring strategies. In these strategies, binders like Nafion are usually required, scaffolds such as CNT (carbon nanotube) and graphene are also included to enhance the dispersion and adhesion strength of molecular catalysts through π - π stacking interaction.

However, considering the possible loss of molecular catalysts during operation (especially at high current densities), anchoring them directly onto carbon-based GDEs through strong covalent bonds is a desirable strategy. Presently, direct covalent grafting of CoPc (and other metal phthalocyanine/porphyrin) based molecular catalysts onto carbon-based GDEs would require the use of CoPc with specially designed functional groups (such as amino group) or polymerizable groups (such as carbazole). Until now, due to the inherent chemical inertness of pristine CoPc, direct electrochemical polymerization or covalent grafting of it onto the carbon-based GDEs is challenging. On the other hand, among common covalent bonds that could form between CoPc (or its derivatives) and carbon-based GDEs, C–C bond is more inert and robust for extreme reaction conditions, when comparing to those hydrolysable amide and ester bonds.

In our study, to offer robust C–C bonds between carbon-based GDEs and pristine CoPc, we introduce ppy as the linker which initiates the C–C coupling process between GDEs and ppy, ppy and pristine CoPc thereby connecting them together via C–C bonds. During this covalent grafting process, pristine CoPc without polymerizable groups is applied; meanwhile, no pretreatment of carbon-based GDEs is required, which minimizes the amount of oxygen-containing functional groups (such as carboxyl and hydroxyl groups) thereby guaranteeing sufficient hydrophobicity for efficient CO₂ gas penetration.

Importantly, this covalent grafting strategy allows facile control over catalyst loadings (e.g., through adjusting the polymerization parameters of CV number, potential range), and could be well extended to other pristine metal phthalocyanines or porphyrins.

(3) The advancements of our CoPc/ppy/GDE electrode.

Based on the combined advantages of robust C–C bonds, binder-free interface, highly conductive ppy linker, appropriate catalyst dispersion, electron donation effects from ppy, and active Co single-atom sites, our in situ covalent grafting derived CoPc/ppy/GDE establishes promising activity and stability for CO₂RR. Specifically, CoPc/ppy/GDE electrode exhibits a high current density of -40 mA cm^{-2} at -0.78 V versus RHE in H-type cells, far better than previously reported CoPc-based catalysts (Table R8); an exceptional 200-hour stability at -0.78 V versus RHE is demonstrated on CoPc/ppy/GDE, and at the end of stability test no obvious valence state, structure and coordination environment changes of Co are detected by XPS, HAADF-STEM and XAS. Furthermore, the as-assembled CoFe-Ci@GQDs/NF||CoPc/ppy/GDE electrolyzer achieves a record

device stability of 50 hours at 1000 mA cm⁻² and 120 hours at 500 mA cm⁻², which strongly demonstrate that our CoPc/ppy/GDE electrode is highly resistant to structural collapse/changes under industrially relevant current densities (Table R9 and Figure R68). Based on the CoFe-Ci@GQDs/NF||CoPc/ppy/GDE electrolyzer, a solar-driven CO₂RR device is constructed, which delivers a high solar-to-fuel efficiency of 19.2%.

In addition, we have also tested the availability of our MEA electrolyzer under pure water conditions. As exhibited in Figure R69, the electrolyzer generates severely limited current densities (for example 30 mA cm⁻² at 3.0 V) using pure water, possibly due to its 10-fold higher system resistance compared to that in 1 M KOH (55 Ω versus 0.45 Ω). At present, we tentatively attribute this ohmic loss of our MEA electrolyzer in pure water to the intrinsically low conductivity of pure water and inappropriate MEA architecture. Our future investigations will focus on these issues, and more importantly, exploring the possibility of running our MEA electrolyzer with proton exchange membrane in pure water (aiming to resolve the mass transport limitation induced by salt precipitation and the low carbon utilization efficiency due to carbonate formation).

Thanks again for your comments.

Table R8. Comparison of the CO₂-to-CO conversion of different electrocatalysts in H-type cells.

Catalysts	Electrolyte	FE _{CO} (%)	Potential (V vs. RHE)	J (mA cm ⁻²)	Stability (h)	Ref.
CoPc/ppy/GDE	0.1 M KHCO ₃	96	-0.78	-40	200	This work
EP-CoP	0.5 M KHCO ₃	95	-0.62	-8.5	42	17
CoTPP/CNT	0.5 M KHCO ₃	91	-0.62	-3.2	4	S8
FePGF/CFP	0.1 M KHCO ₃	98.7	-0.54	-1.68	10	S9
Co-PMOF	0.5 M KHCO ₃	98.7	-0.8	-18.3	36	S10
STPyP-Co	0.5 M KHCO ₃	96	-0.62	-6.6	48	S11
Fe-PB	0.5 M KHCO ₃	100	-0.63	~-0.6	24	S12
Co-TTCOF	0.5 M KHCO ₃	91.3	-0.7	-2.02	40	S13
COF-366- (OMe) ₂ - Co@CNT	0.5 M KHCO ₃	93.6	-0.68	~-6	12	S14
COF-367-Co	0.5 M KHCO ₃	91	-0.55	-3.3	24	S15

CoPc-TFPN COF	0.5 M KHCO ₃	99.8	-0.9	-14.1	60	S16
TT-Por(Co)- COF	0.5 M KHCO ₃	91.4	-0.6	~-1.5	10	S17
CoPc-PI-COF-1	0.5 M KHCO ₃	90	-0.7	-10	40	S18
MWCNT-Por- COF-Co	0.5 M KHCO ₃	88	-0.7	-6	50	S19
CoPc/CNT	0.1 M KHCO ₃	92	-0.63	-10	10	S20
CoTMAPc@CN T	0.5 M KHCO ₃	95	-0.62	~-12	12	S21
CoPPc/CNT	0.5 M NaHCO ₃	90	-0.54	-12	24	S22
CoPc-2H2Por	0.5 M KHCO ₃	95	-0.6	-5	70	S23

Table R9. Comparison of the CO₂-to-CO conversion of different electrocatalysts in MEA.

Catalysts	Electrolyte	FE _{CO}	Cell voltage (V)	J (mA cm ⁻²)	Stability (h)	References
CoPc/ppy/GDE	1.0 M KOH	99%	~2.8	500	120	This work
CoPc/ppy/GDE	1.0 M KOH	98%	~3.2	1000	50	This work
Co-CNTs MW	1.0 M KOH	98.3%	~2	100	12	44
CoPc	1.0 M KOH	88%	~2.52	200	6	45
MWNT/ PyPBI/Au	2.0 M KOH	85%	~2.25	158	8	46
Ag/C	1.0 M KOH	83%	~2.75	120	0.417	47
Ag nanoparticles	1.0 M KOH+0.33 M Urea	98%	~2.16	100	10	48
Zn ₂ P ₂ O ₇	1.0 M KOH	93.9%		100	7	49
Au ₂₄	0.1 M KOH	90.0%	~3	100	100	50

TC-CoPc/MWCNTs	0.5 M KHCO ₃	97%	~2.6	50	60	17
CoPc/Mg(OH) ₂	0.1 M KHCO ₃	95%	3.4	100	20	51
β-CoPc/CP	1.0 M KOH	92%	2	100	29.3	52
AG-CoPc/MWCNTs	0.1 M KHCO ₃	80%	3	50	12	26
MD-CoPc/MWCNTs	0.1 M KHCO ₃	80%	3.2	50	40	26

Figure R68. The state-of-art summary of CO₂-to-CO electrocatalysts in MEA devices.

Figure R69. LSV and Faradaic efficiency of CO₂-to-CO conversion in the CoFe-Ci@GQDs/NF||CoPc/ppy/GDE electrolyzers with pure water.

References

- [1] Kim, Y. et al. Integrated CO₂ capture and conversion to form syngas. *Joule* **8**, 3106–3125 (2025).

- [2] Khoshooei, M. A. et al. An active, stable cubic molybdenum carbide catalyst for the high-temperature reverse water-gas shift reaction. *Science* **8**, 3106–3125 (2025).
- [3] Reaction-induced unsaturated Mo oxycarbides afford highly active CO₂ conversion catalysts. doi.org/10.1038/s41929-023-00937-0.
- [4] Hao, S. Y. et al. Improving the operational stability of electrochemical CO₂ reduction reaction via salt precipitation understanding and management. *Nat. Energy* **10**, 266–277 (2025).
- [5] McGregor, J. M. et al. Organic electrolyte cations promote non-aqueous CO₂ reduction by mediating interfacial electric fields. *Nat. Catal.* **8**, 79–91 (2025).
- [6] Yu, X. H. et al. Coverage enhancement accelerates acidic CO₂ electrolysis at ampere-level current with high energy and carbon efficiencies. *Nat. Commun.* **15**, 1711 (2024).
- [7] Tang, X. H. et al. Direct electrosynthesis of 52% concentrated CO on silver's twin boundary. *Nat. Commun.* **12**, 2139 (2021).

Responses to Reviewers' Comments

Reviewer #1:

Comment: After reviewing the manuscript, I acknowledge the authors' efforts in conducting additional experiments and expanding the discussion. Although similar systems have been reported in the literature, this work presents some performance advantages. Given the sound methodology and the relevance of the results, I recommend the manuscript for publication.

Reply to comment: Thanks for your positive feedback! Based on your insightful comments and suggestions, the quality of our manuscript has been significantly improved. Thanks again!

Reviewer #2:

Comment: The authors have revised the ms according to the comments and can be accepted. Still, there are some grammar and words needing further revision, e.g. times or time?

Reply to comment: Thanks for your positive feedback! With your valuable comments and suggestions, the quality of our manuscript has been significantly improved. Based on your suggestion, we have carefully reviewed the entire manuscript to correct the errors associated with grammar, word and phrasing. Thanks again!

Reviewer #2's comments on your response to Reviewer #3's concerns:

Comment: The authors have replied the comment 3 and revised the ms according to the comments. The reply was correct and helpful to the final ms. The present ms can be accepted.

Reply to comment: Thanks for your positive feedback! We are grateful for your time and insightful comments throughout the review process, which have greatly improved the quality of our work. Thanks again!

Reviewer #4:

Comment: Although the authors have addressed some of the previous comments, the overall quality of the manuscript still does not meet the high publication standards of Nature Communications. Therefore, I recommend submitting this work to a more specialized journal. Specific suggestions for improvement include:

Comment 1: While the revised manuscript provides additional data, the fundamental mechanism underlying the enhanced activity and stability of the CoPc/ppy/GDE system remains insufficiently elucidated. For instance: the role of polypyrrole in modulating the electronic structure of CoPc is discussed, but direct experimental evidence to confirm the proposed electron-donation effect is still lacking.

Reply to comment 1:

We sincerely thank the reviewer for raising this important point, which encourages us to find the underlying mechanism using obtained experimental data.

In our study, to offer robust C–C bonds between carbon-based GDEs and pristine CoPc, we introduce ppy as the linker which initiates the C–C coupling process between GDEs and ppy, ppy and pristine CoPc thereby connecting them together via C–C bonds. During this covalent grafting process, pristine CoPc without polymerizable groups is applied; meanwhile, no pretreatment of carbon-based GDEs is required, which minimizes the amount of oxygen-containing functional groups (such as carboxyl and hydroxyl groups) thereby guaranteeing sufficient hydrophobicity for efficient CO₂ gas penetration. As far as we know, such a convenient and efficient approach has not been explored for direct grafting of pristine CoPc (rather than CoPc derivatives) on carbon-based GDEs, the employed ppy not only acts as the highly conductive and robust linker between CoPc and GDEs, but also shows electron donation effects to promote the CO₂RR activity of CoPc centers.

Regarding this electron donation effect from ppy to CoPc, two important evidences have been provided. The first evidence is from XPS analysis. Compared to commercial CoPc (781.4 eV and 796.5 eV), the binding energies in the Co 2p XPS spectra of CoPc/ppy/GDE (780.8 eV and 795.9 eV for Co 2p_{3/2} and Co 2p_{1/2}, respectively) decrease by 0.6 eV, indicating an increased electron density around the Co centers (Figure R1). The second evidence is from XAS analysis. We performed XAS at the Co K-edge on both pristine CoPc and the CoPc/ppy/GDE composite. The results are presented in the Figure R2. The absorption edge of CoPc/ppy/GDE is redshifted compared to pristine CoPc, with weakened 1s→4p_z transition at ~7710 eV; this suggests reduced symmetry of the Co–N₄ coordination in CoPc/ppy/GDE, and the average Co oxidation state locates between metallic Co (0) and CoO (+2). The Co K-edge white line intensity in CoPc/ppy/GDE is lower than that in pristine CoPc. This decrease in white line intensity indicates an increase in electron density at the Co metal center, consistent with electron transfer from the electron-rich ppy

matrix. Fourier-transform extended X-ray absorption fine structure (FT-EXAFS) analyses identify a dominant Co–N coordination peak at ~ 1.5 Å for both CoPc/ppy/GDE and pristine CoPc, indicating Co^{2+} –N as the primary species in them. Wavelet-transforms of EXAFS spectra further confirm this conclusion by revealing prominent intensity maxima at ~ 1.5 Å; meanwhile, a minor intensity maximum is exclusively resolved at ~ 2.7 Å over CoPc/ppy/GDE, which is inferred to arise from the Co^+ –N species (Figure R3). The presence of this Co^+ –N species along with the negatively shifted binding energies of Co 2p peaks together support the electron-donation effect from ppy to CoPc.

Figure R1. Comparison of the Co 2p XPS spectra of CoPc/ppy/GDE and commercial CoPc.

Figure R2. a, XAS spectra at Co K-edge of CoPc/ppy/GDE and the reference Co foil and CoPc samples. b, Co K-edge FT EXAFS spectra in the R space of CoPc/ppy/GDE and reference Co foil and CoPc samples.

Figure R3. a, b WT-EXAFS plots of CoPc samples and CoPc/ppy/GDE.

Furthermore, researchers have claimed that Co^+ species in CoPc formed by one-electron reduction of Co^{2+} , is the active site for CO_2RR ^{1,2}. Through the diagnostic Raman signatures of Co^+ (at ca. 1110 cm^{-1}) and Co^{2+} species (at ca. 1140 cm^{-1}), the identification and quantification of Co^+ active species have been achieved³. Following these reports, we have also monitored the evolution of CoPc/ppy/GDE and the distribution of Co^+ and Co^{2+} species (i.e. the ratio of Co^+ to Co^{2+}) with applied potentials through in situ surface-enhanced Raman spectroscopy (SERS). As shown in Figure R4, at -0.4 V versus RHE, characteristic peaks could be resolved at 1110 and 1140 cm^{-1} on both electrodes of CoPc/ppy/GDE and CoPc/CNT-GDE, which are reported to be associated with the Co^+ and Co^{2+} species, respectively^{4,5}. The signal intensity of Co^+ species (at 1110 cm^{-1}) over CoPc/ppy/GDE becomes prominent at ca. -0.7 V versus RHE, consistent with the redox behaviour of $\text{Co}^{2+}/\text{Co}^+$ in CoPc; beyond this potential, the spectrum reaches a relatively steady state. In comparison, the formed Co^+ species over CoPc/CNT-GDE is significantly lower in amount. Using the Raman peak area, we calculated the ratios of Co^+ to Co^{2+} in CoPc/ppy/GDE and CoPc/CNT-GDE at various applied potentials. The results in Figure R5 show that the formed Co^+ species over CoPc/ppy/GDE are always higher than that of CoPc/CNT-GDE, for which we infer that the electron donation effects from ppy to CoPc is responsible⁶.

Figure R4. a, b, In situ SERS of the CoPc/ppy/GDE and CoPc/CNT-GDE during CO₂RR under applied bias ranging from -0.4 to -1.4 V versus RHE.

Figure R5. The ratio of Co⁺: Co²⁺ during CO₂RR under applied bias ranging from -0.4 to -1.4 V versus RHE in CO₂-saturated 0.1 M KHCO₃.

In the previous study, Co⁺ species in CoPc is reported to be stable up to the potential of -2.2 V versus RHE, as the percentage of Co⁺ species reaches a steady state starting from ca. -1.0 V versus RHE³. Our results under operando conditions in Figure R4 also indicate that no obvious reduction of Co⁺ species occurs up to -1.4 V versus RHE. We have also evaluated the stability of Co⁺ species during CO₂RR using in situ SERS. As shown in Figure R6, with the continuous application of a negative potential of -1.4 V versus RHE, no obvious degradation in the peak intensity of Co⁺ species could be detected on CoPc/ppy/GDE over a reaction period of 30 min; while for CoPc/CNT-GDE, a gradual decrease of Co⁺ species appears, possibly due to the progressive loss of CoPc from CoPc/CNT-GDE electrode. We attribute the enhanced stabilization of Co⁺ on CoPc/ppy/GDE than CoPc/CNT-GDE to the electron donation effects from ppy to CoPc.

Figure R6. Time-dependent in situ SERS of the (a) CoPc/ppy/GDE (b) CoPc/CNT-GDE during CO₂RR from 0 to 30 min at -1.4 V versus RHE in CO₂-saturated 0.1 M KHCO₃.

To gain more comprehensive mechanistic insights into ppy-enhanced electrocatalytic activity, free energy diagrams considering thermodynamics of CO₂-to-CO conversion on CoPc/ppy was constructed using DFT calculations. Bader charge analysis suggests that ppy has a tendency to transfer electrons to CoPc (Figure R7). The ppy acts as a linker to anchor CoPc onto the GDE through covalent bonds, while simultaneously modulating the electronic structure of CoPc for improved catalytic performance.

Figure R7. Schematic diagram of differential charge density of CoPc/ppy and Bader charge analysis. The cyan (yellow) region shows electron loss (gain).

Combined all the above experiments and analyses together, we propose that polypyrrole can modulate the electronic structure of CoPc and the Co⁺ could be the active species in CoPc/ppy/GDE for CO₂RR. Co⁺ species are possibly stabilized by the microenvironment and ligands. We hope these additions have decisively addressed the reviewer's concern again.

References

- [1] Wang, S. F. et al. Manipulating C-C coupling pathway in electrochemical CO₂ reduction for selective ethylene and ethanol production over single-atom alloy catalyst. *Nat. Commun.* **15**, 10247

(2024).

[2] Corbin, N. et al. Heterogeneous molecular catalysts for electrocatalytic CO₂ reduction. *Nano Res.* **12**, 2093–2125 (2019).

[3] Hu, X. M. et al. Enhanced catalytic activity of cobalt porphyrin in CO₂ electroreduction upon immobilization on carbon materials. *Angew. Chem. Int. Ed.* **56**, 6468–6472 (2017).

[4] Ren, S. et al. Catalyst aggregation matters for immobilized molecular CO₂RR electrocatalysts. *J. Am. Chem. Soc.* **145**, 4414–4420 (2023).

[5] Jiang, S. et al. Investigation of cobalt phthalocyanine at the solid/liquid interface by electrochemical tip-enhanced Raman spectroscopy. *J. Phys. Chem. C* **123**, 9852–9859 (2019).

[6] Chen, X. et al. Operando observation of molecular-scale manipulation using electrochemical tip-enhanced Raman spectroscopy. *J. Phys. Chem. C* **123**, 24329–24333 (2018).

Comment 2: The claim that C–C covalent bonds are formed between CoPc and ppy relies heavily on indirect evidence. More rigorous characterization is needed to unambiguously verify the bonding configuration.

Reply to comment 2: Thank you for your valuable comments on our manuscript. We understand your concerns regarding the direct evidence for the formation of covalent bonds between CoPc and the ppy network. We have taken your suggestions seriously and have conducted additional experiments and analyses to address this issue.

First, we utilized in situ Raman spectroscopy to monitor the structural evolution of CoPc during the electro-polymerization on GDE. As shown in Figure R8, the Raman spectra reveal gradually increased peaks that correspond to the characteristic vibrational modes of CoPc. Specifically, the macrocyclic deformation mode at 680 cm⁻¹ exhibits an increase trend with the number of CV polymerization cycles, quantitatively correlating with the progressive deposition of CoPc. We also observed a shift of Co–N vibration at ca. 755 cm⁻¹, which is possibly associated with the interfacial strain-induced changes of Co–N bonds in CoPc during polymerization¹. Collectively, the progressively increased intensities of CoPc-related peaks with the number of CV cycles², provide evidence for the in situ growth and integration of CoPc onto the GDE.

Figure R8. In situ Raman spectra of CoPc/ppy/GDE collected at different CV cycles in 0.1 M [BMIM]BF₄ with 4 mM pyrrole and 1 mM CoPc.

Second, to provide further evidence for the covalent integration of CoPc with ppy on the GDE, CV cycles were systematically performed within under controlled potential windows. As shown in Figure R9, ppy generated from electrochemical polymerization of pyrrole shows only broad and featureless oxidation and reduction waves in the potential range of -1.0 to 1.0 V vs. Ag/AgCl. Comparatively, three redox couples could be resolved for CoPc-modified GDE (CoPc/GDE), with their locations presented in Table R1. These redox couples are probably associated with the redox behaviours of Co centres of CoPc ($\text{Co}^{2+} \leftrightarrow \text{Co}^+ \leftrightarrow \text{Co}^0$). Interestingly, CoPc/ppy/GDE electrode produced in the presence of both pyrrole and CoPc shows similar redox characteristics to that of CoPc/GDE, with however negatively shifted peak positions and largely enhanced current response in the cathodic scan. Based on this observation, we infer that during pyrrole polymerization CoPc is integrated into ppy, with the configuration of CoPc terminated at the ppy chain; electron donation occurs from ppy to CoPc, thus yielding negatively shifted redox peaks relative to bare CoPc.

Figure R9. Comparative CV patterns of ppy/GDE in 0.01 M [BMIM]BF₄ with 4 mM pyrrole, CoPc/GDE in 0.01 M [BMIM]BF₄ with 1 mM CoPc and CoPc/ppy/GDE in [BMIM]BF₄ with 4 mM pyrrole and 1 mM CoPc at 100 mV s⁻¹.

Table R1. The location of the CV peaks

Catalysts	Reduction 1	Reduction 2	Reduction 3
CoPc/ppy/GDE	0.52	-0.04	-0.98
CoPc/GDE	0.61	0.03	-0.88
	Oxidation 1	Oxidation 2	Oxidation 3
CoPc/ppy/GDE	-0.27	0.36	1.18
CoPc/GDE	-0.15	0.48	1.32

Third, we used F-doped tin oxide (FTO) as the conductive substrate to study the grow process of CoPc/ppy, which would exclude the influence of carbon-based substrates. We collected FTIR spectrum of CoPc/ppy/GDE and compared to those of CoPc and ppy/GDE. To avoid the possible contamination of free CoPc (i.e., CoPc adsorbed on GDE or ppy), CoPc/ppy/GDE was thoroughly cleaned by ultrasonication and washed by copious amounts of acetonitrile. As shown in Figure R10, the FTIR spectrum of CoPc/ppy/GDE shows a combined feature of CoPc and ppy/GDE, which indicates that CoPc had been successfully incorporated into ppy/GDE.

Figure R10. FTIR spectra of CoPc/GDE, CoPc/ppy/GDE and ppy/GDE.

Fourth, we have also collected solid-state ^1H NMR spectra of CoPc/ppy, ppy and pristine CoPc (Figure R11). Critically, pristine CoPc exhibits a broad hump in the 6–7 ppm region, while ppy displays its characteristic aromatic proton peaks in the same range. In contrast, the CoPc/ppy composite shows distinct, well-resolved peaks between 5–6 ppm, indicating a significant alteration in its local electronic environment. Importantly, the peak intensity at ca. 7.5 ppm in CoPc/ppy is obviously enhanced than that of ppy, making the signal shape of ppy matrix no longer symmetric. We attribute this phenomenon to protons proximal to a newly formed covalent C–C bond between CoPc and the ppy chain, generated via radical coupling during the electrochemical polymerization process. This covalent grafting changes the chain end structure of ppy, thereby perturbs the electron density and results in a unique absorption.

Fifth, to probe the molecular structure of CoPc/ppy, matrix-assisted laser desorption/ionization time-of-flight mass spectrometry (MALDI-TOF-MS) was employed (Figure R12). The analysis revealed suppression of the characteristic low-mass oligomer peaks of ppy (e.g., the series around m/z 65, 130, and 195). Concurrently, new signals associated with CoPc fragments emerged and became dominant, most notably an intense peak at m/z 128, which is attributed to phthalonitrile ($\text{C}_8\text{H}_4\text{N}_2^+$), along with a peak at m/z 59 corresponding to the cobalt ion.

Figure R11. Solid-state ^1H NMR spectra of CoPc/ppy, CoPc and ppy.

Figure R12. MALDI-TOF-MS spectra of CoPc/ppy, CoPc and ppy.

Sixth, CoPc/ppy/GDE electrode exhibits an exceptional 200 hours stability during chronoamperometry measurements at -0.78 V versus RHE, while the control sample of CoPc/CNT-GDE shows only a considerably shorter operational lifespan of 12 hours. After 200 hours stability test, no obvious valence state, structure and coordination environment changes were detected over CoPc/ppy/GDE by XPS, HAADF-STEM and XAS. Furthermore, the as-assembled CoFe-Ci@GQDs/NF||CoPc/ppy/GDE electrolyzer achieves record device stability of 50 hours at 1000 mA cm^{-2} and 120 hours at 500 mA cm^{-2} .

All these experimental observations point to the formation of covalent bonds between CoPc and the ppy network, based on which improved activity (due to electron transfer from ppy to CoPc) and robust operational stability are simultaneously achieved. We hope that these additional analyses would address your concerns and strengthen the scientific validity of our work.

References

- [1] Szybowicz, M. et al. Micro-Raman spectroscopic investigations of cobaltphthalocyanine thin films deposited on quartz and diamond substrates. *Cryst. Res. Technol.* **45**, 1265–1271 (2010).
- [2] Chen, Y. T. et al. Charge transfer and electromagnetic enhancement processes revealed in the SERS and TERS of a CoPc thin film. *Nanophotonics* **8**, 1533–1546 (2019).

Comment 3: The manuscript emphasizes the novelty of covalent grafting for pristine CoPc, but the performance metrics do not significantly surpass recent reports using functionalized molecular catalysts. The stability data is commendable but could be further contextualized against industrial benchmarks.

Reply to comment 3: We thank the reviewer for this insightful comment, which allows us to better articulate the significance and novelty of our work. We agree that the field of molecular CO₂RR catalysis is advancing rapidly, with many high performing systems being reported. The value of our work demonstrates a highly effective, scalable, and generalizable strategy for integrating molecular catalysts onto electrodes, which addresses critical stability and scalability issues that often plague molecular-based systems.

(1) Novelty in simplicity and generalizability:

The reviewer notes that functionalized molecular catalysts can achieve high performance. However, the synthesis of such specially designed molecules (e.g., with pyrene or ethynyl anchoring groups) often involves multi-step, low-yield organic synthesis requiring specialized expertise, which poses a significant barrier to scale-up and cost reduction (Figure 13). In our study, to offer robust C–C bonds between carbon-based GDEs and pristine CoPc, we introduce ppy as the linker which initiates the C–C coupling process between GDEs and ppy, ppy and pristine CoPc thereby connecting them together via C–C bonds. During this covalent grafting process, pristine CoPc without polymerizable groups is applied; meanwhile, no pretreatment of carbon-based GDEs is required, which minimizes the amount of oxygen-containing functional groups (such as carboxyl and hydroxyl groups) thereby guaranteeing sufficient hydrophobicity for efficient CO₂ gas penetration. As far as we know, such a convenient and efficient approach has not been explored for direct grafting of pristine CoPc (rather than CoPc derivatives) on carbon-based GDEs, the employed ppy not only acts as the highly conductive and robust linker between CoPc and GDEs, but also shows electron donation effects to promote the CO₂RR activity of CoPc centers. Importantly, this covalent grafting strategy allows facile control over catalyst loadings (e.g., through adjusting the polymerization parameters of CV number, potential range), and could be well extended to other pristine metal phthalocyanines or porphyrins (Figure R14).

	100 mg \$9		100 mg \$375
Co(II) meso-Tetraphenylporphine		Co(II) meso-Tetra (3,5-dihydroxyphenyl) Porphine Chloride	
	100 mg \$91		100 mg \$180
Co(III) Protoporphyrin IX chloride		Co(III) meso-Tetra (4-carboxyphenyl) Porphine Chloride	
	10 mg \$175		100 mg \$279
Co(III) Etioporphyrin I chloride		Co(III) meso-Tetra (N-methyl-4-pyridyl) Porphine pentachloride	
	100 mg \$210		100 mg \$404
Co(III) meso-Tetra (4-hydroxyphenyl) Porphine Chloride		Co(II) meso-Tetra (Pentafluorophenyl) Porphine	

Figure R13. Prices of pristine CoPc and functionalized molecular catalysts.

Figure R14. a, LSV curves of CoPc/ppy/GDE, FePc/ppy/GDE, CuPc/ppy/GDE and NiPc/ppy/GDE during CO₂RR in CO₂-saturated 0.1 M KHCO₃ with an H-type cell. **b,** Faradaic efficiency and current density of CoPc/ppy/GDE, FePc/ppy/GDE, CuPc/ppy/GDE and NiPc/ppy/GDE during CO₂RR in CO₂-saturated 0.1 M KHCO₃ with an H-type cell at -0.7 V versus RHE.

(2) Contextualizing stability against industrial benchmarks:

We thank the reviewer for this positive feedback and valuable suggestion. We fully agree that contextualizing stability performance relative to emerging industrial benchmarks is essential for evaluating the practical potential of our technology. The performance of CoPc/ppy/GDE not only

rival those of advanced pre-functionalized systems but also demonstrate exceptional industrial relevance and scalability. The as-assembled CoFe-C_i@GQDs/NF||CoPc/ppy/GDE electrolyzer achieves a record device stability of 120 hours at 500 mA cm⁻² and 50 hours at 1000 mA cm⁻² (Figure R15), which strongly demonstrate that our CoPc/ppy/GDE electrode is highly resistant to structural collapse/changes under industrially relevant current densities. To contextualize this performance, we provide a comprehensive multi-parameter comparison with several recently reported high-performing MEA devices based on CoPc or its derivatives, using a radar chart that quantitatively evaluates key metrics such as operating current density, stability, CO Faradaic efficiency, and cell voltage (Figure R16, Table R2). Crucially, the performance remains consistent when the device is scaled to a 9 cm² electrode and operates continuously for 20 hours at a total current of 1 A (Figure R17). Furthermore, we demonstrated a hectowatt-level (180 W) stack operating at 1 A cm⁻², providing concrete evidence of the practical viability and scalability of our approach (Figure R18). Coupled with an unbiased solar-to-CO efficiency of 19.2%, these results underscore that our method using pristine catalysts delivers industrially-relevant performance while offering superior simplicity and universality.

Figure R15. Radar chart comparing the performance metrics of different molecular catalysts in a MEA.

Table R2. Comparison of the CO₂-to-CO conversion of different electrocatalysts in MEA.

Catalysts	FE _{CO} (%)	Cell voltage (V)	J (mA cm ⁻²)	Stability (h)
CoPc/ppy/GDE	99	~2.8	500	120
Co-CNTs MW	98.3	~2	100	12

CoPc	88	~2.52	200	6
MWNT/PyPBI/Au	85	~2.25	158	8
TC-CoPc/MWCNTs	97	~2.6	50	60
CoPc/Mg(OH) ₂	95	3.4	100	20
β-CoPc/CP	92%	2	100	29.3
AG-CoPc/MWCNTs	80	3	50	12
MD-CoPc/MWCNTs	80	3.2	50	40

Figure R16. Long-term stability test at 1 A cm^{-2} with corresponding Faradaic efficiency of the electrolyzer in 1.0 M KOH .

Figure R17. a, LSV and Faradaic efficiency of $\text{CoFe-C}_i\text{@GQDs/NF||CoPc/ppy/GDE}$ electrolyzers with electrode area of 9 cm^2 during CO_2RR in 1.0 M KOH . b, Stability test at 1 A with corresponding Faradaic efficiency of the electrolyzer in 1.0 M KOH .

Figure R18. Stability and power of the electrolyzer at a current density of 1 A cm^{-2} .

In summary, we present a simplified and robust integration strategy that enables a cheap, pristine catalyst to perform on par with or exceed the durability of more complex, customized systems. We believe this approach, which prioritizes scalability and stability alongside high performance, is a valuable and novel addition to the field.

Comment 4: The post-reaction characterization suggests structural retention, but the origin of eventual performance decay is not investigated. Long-term stability tests under intermittent operation would strengthen the practical relevance.

Reply to comment 4: We are grateful to the reviewer for this exceptionally valuable suggestion. Investigating the degradation mechanisms and testing stability under practical, intermittent operating conditions are crucial steps toward assessing the practical potential of any electrocatalyst. We have conducted these recommended experiments, and the results have strengthened its practical relevance.

(1) Investigating the origin of performance decay.

To probe the origin of the eventual activity decay observed in our long-term test, we performed a detailed post-mortem analysis of the CoPc/ppy/GDE electrode after 50 hours of continuous operation at 500 mA cm^{-2} . We have conducted a post-mortem analysis of the cell components and explored the reasons for the gradual performance loss. The post-test images of the GDE and the flow field reveal substantial blockage of the critical gas and liquid transport pathways by these insoluble salts (Figure R19). This phenomenon, often referred to as salt precipitation, is a well-known challenge in CO_2 electrolysis operating in aqueous alkaline electrolytes, particularly at high current densities and extended durations. The precipitation physically blocks the triple-phase boundaries, severely hindering the mass transport of CO_2 to the active sites and leading to increased

polarization and eventual performance decay.

Figure R19. Photograph of the electrolysis cell and electrodes.

The structural and chemical retention of the CoPc/ppy/GDE catalyst itself, as confirmed by pre- and post-reaction spectroscopic analyses (XAS, XPS, and HAADF-STEM), indicates that the eventual performance decay is caused by a system-level issue of salt management, rather than by intrinsic catalyst deactivation (Figure R20–22). We thank the reviewer again for prompting this important investigation, which significantly strengthens the practical relevance of our study.

Figure R20. **a**, XAS spectra at Co K-edge of CoPc/ppy/GDE and **b**, Co K-edge FT EXAFS spectra in the R space of CoPc/ppy/GDE of initial state and after 200 hours in CO₂-saturated 0.1 M KHCO₃ with an H-type cell at -0.75 V versus RHE.

Figure R21. Comparison of the Co 2p XPS spectra of CoPc/ppy/GDE of initial state and after 200 hours in CO₂-saturated 0.1 M KHCO₃ with an H-type cell at -0.75 V versus RHE.

Figure R22. a, TEM image of post-reacted CoPc/ppy/GDE after 200 hours. **b,** HAADF-STEM image of post-reacted CoPc/ppy/GDE after 200 hours.

(2) Long-term stability under intermittent operation:

Following the reviewer's excellent suggestion, we conducted a new stability test under intermittent operation to simulate the intermittent nature of renewable electricity sources. The experiment consisted of 50 cycles. The results are presented in the Figure R23. Remarkably, the catalyst quickly recovers its activity after each idle period, maintaining a stable $FE_{CO} > 80\%$ throughout all 50 cycles. This demonstrates the outstanding robustness and resilience of the CoPc/ppy/GDE catalyst. The covalent linkage prevents detachment or reconstruction during the frequent potential swings and idle times, which is a common failure mode for non-covalently attached catalysts.

Figure R23. Long-term stability tests under intermittent operation of 100 mA cm^{-2} of the electrolyzer in 1.0 M KOH.

We believe these additions, prompted by the reviewer's comment, have transformed a simple stability demonstration into a mechanistic investigation of electrode longevity, enhancing the impact and significance of our work.

Comment 5: The DFT analysis of CO_2 reduction energetics is oversimplified. Key aspects are overlooked: How does the ppy linker influence the local pH or CO_2 concentration at the catalyst surface? Are the proposed active sites (Co^+) stable under alkaline conditions, given prior reports of CoPc demetallation at high pH?

Reply to comment 5: We thank the reviewer for raising these important questions, which allow us to clarify the scope and conclusions of our computational study. In response to the query regarding the potential influence of the ppy linker on the local chemical environment, we modelled the effect of varying local proton and hydroxide concentrations by introducing H^+ and OH^- into the solvation sphere of the Co center in both the CoPc and CoPc/ppy models.

As shown in Figure R24, we computed the average adsorption energy of H^+ for CoPc and CoPc/ppy with varying numbers of protons. The results indicate that the trend in adsorption energy as a function of H^+ number is nearly identical for both systems, with minimal differences observed at each specific value. This suggests that the presence of the ppy linker does not significantly alter the local proton affinity around the Co site. Similarly, Figure R25 presents the corresponding analysis for OH^- . Figures R25a and R25b show representative optimized structures of CoPc and CoPc/ppy with different numbers of OH^- , with the resulting adsorption energies. Again, the trends are similar between CoPc and CoPc/ppy, and the energy differences are negligible. These results indicate that the ppy matrix does not promote the accumulation of OH^- near the catalytic center. Therefore, the DFT calculations demonstrate that the local pH environment around the Co site remains largely unaffected by the presence of ppy. Consequently, the enhancing effect of the ppy modifier is likely electronic and structural in origin, rather than mediated through changes in the local concentration of protons or CO_2 . Moreover, since each individual CoPc unit possesses only one CO_2 adsorption site, the ppy matrix is unlikely to influence the local CO_2 concentration at the

catalytic center.

Figure R24. Adsorption configurations **a**, **b** and **(c)** corresponding adsorption energies of CoPc and CoPc/ppy under different acidities.

Figure R25. Adsorption configurations **a**, **b** and **(c)** corresponding adsorption energies of CoPc and CoPc/ppy under different alkalinities.

Regarding the stability of the proposed Co^+ species under alkaline conditions, we fully appreciate the reviewer's concern based on known demetallation pathways of molecular CoPc. Our computational stability assessments included evaluation of the coordination strength of the Co center in the presence of explicit hydroxide ions. Figures R26a and R26b show the anchoring of Co in the phthalocyanine macrocycle without OH^- , while Figures R26c and R26d illustrate the structures with coordinated OH^- . The calculated binding energies of Co in the macrocyclic framework are exceptionally high, indicating strong stabilization. The presence of OH^- cause a minor change in the binding energy (on the order of 0.4 eV), leading us to conclude that alkaline conditions do not significantly compromise the stability of the Co site.

Figure R26. The binding energy of Co of CoPc/ppy under alkaline conditions.

Our approach offers a thermodynamically grounded assessment that the proposed mechanism and active site are viable. We hope this addresses the reviewer's concern regarding the potential oversimplification of the DFT analysis.

Responses to Reviewer' Comments

Reviewer #4:

Comment: I thank the authors for their thorough revisions in response to the feedback. They have addressed the majority of the concerns raised. I have no further issues and believe the manuscript is now suitable for publication.

Reply to comment: Thanks for your positive feedback! Based on your insightful comments and suggestions, the quality of our manuscript has been significantly improved. Thanks again!